# Live tumor imaging shows macrophage induction and TMEM-mediated enrichment of cancer stem cells during metastatic dissemination

Ved P. Sharma [1,2], Binwu Tang[3], Yarong Wang[1,2,4], Camille L. Duran [1], George S. Karagiannis[1,4], Emily A. Xue[1], David Entenberg [1,2,4], Lucia Borriello[1], Anouchka Coste[1,5], Robert J. Eddy[1], Gina Kim [1], Xianjun Ye [1], Joan G. Jones[1,4,6,7], Eli Grunblatt[8], Nathan Agi[8], Sweta Roy[8], Gargi Bandyopadhyaya[8], Esther Adler[9], Chinmay R. Surve[1,4], Dominic Esposito [10], Sumanta Goswami[1,8], Jeffrey E. Segall[1,2], Wenjun Guo [11,12], John S. Condeelis [1,2,4,5✉], Lalage M. Wakefield[3✉] & Maja H. Oktay[1,2,4,6✉]

Cancer stem cells (CSCs) play an important role during metastasis, but the dynamic behavior and induction mechanisms of CSCs are not well understood. Here, we employ high-resolution intravital microscopy using a CSC biosensor to directly observe CSCs in live mice with mammary tumors. CSCs display the slow-migratory, invadopod-rich phenotype that is the hallmark of disseminating tumor cells. CSCs are enriched near macrophages, particularly near macrophage-containing intravasation sites called Tumor Microenvironment of Metastasis (TMEM) doorways. Substantial enrichment of CSCs occurs on association with TMEM doorways, contributing to the finding that CSCs represent >60% of circulating tumor cells. Mechanistically, stemness is induced in non-stem cancer cells upon their direct contact with macrophages via Notch-Jagged signaling. In breast cancers from patients, the density of TMEM doorways correlates with the proportion of cancer cells expressing stem cell markers, indicating that in human breast cancer TMEM doorways are not only cancer cell intravasation portals but also CSC programming sites.

[1] Department of Anatomy and Structural Biology, Albert Einstein College of Medicine, Bronx, NY, USA. [2] Gruss-Lipper Biophotonics Center, Albert Einstein College of Medicine, Bronx, NY, USA. [3] Laboratory of Cancer Biology and Genetics, National Cancer Institute, Bethesda, MD, USA. [4] Integrated Imaging Program, Albert Einstein College of Medicine, Bronx, NY, USA. [5] Department of Surgery, Albert Einstein College of Medicine, Bronx, NY, USA. [6] Department of Pathology, Albert Einstein College of Medicine, Bronx, NY, USA. [7] Department of Epidemiology and Population Health, Albert Einstein College of Medicine, Bronx, NY, USA. [8] Department of Biology, Yeshiva University, New York, NY, USA. [9] Department of Pathology, NYU Langone Medical Center, New York, NY, USA. [10] Protein Expression Laboratory, Frederick National Laboratory for Cancer Research, Frederick, MD, USA. [11] Department of Cell Biology, Albert Einstein College of Medicine, Bronx, NY, USA. [12] Ruth L. and David S. Gottesman Institute for Stem Cell and Regenerative Medicine Research, Albert Einstein College of Medicine, Bronx, NY, USA. ✉email: john.condeelis@einsteinmed.org; wakefiel@dce41.nci.nih.gov; moktay@montefiore.org

Metastasis, which accounts for >90% of cancer-related deaths, is a multi-step process involving cancer cell intravasation into the bloodstream, dissemination to distant sites, and seeding in new organs[1]. A conceptual advance in understanding of this complex process came from realization that breast tumors are hierarchically organized with respect to individual cell proliferative potential and differentiation status[2]. At the apex of the hierarchy are the cancer stem cells (CSCs). CSCs have an enhanced ability to self-renew, which makes them uniquely capable of initiating and sustaining primary and metastatic tumor growth[3]. Consistent with this concept, a high proportion of CSCs in primary tumors is associated with poor prognosis and increased metastatic relapse[4]. Importantly, CSCs are intrinsically more therapy resistant than their more differentiated progeny[5]. Thus, a better understanding of the in vivo mechanisms that regulate CSCs is essential for the development of more effective anti-cancer therapies.

Cancer progression is profoundly influenced by complex dynamic and reciprocal interactions between the tumor cells and other components of the tumor microenvironment[6,7]. Depending on the tissue, many different cell types, as well as extracellular matrix, secreted factors, and other microenvironmental components, can contribute to the niches that nurture and sustain the CSC phenotype and influence subsequent CSC fate[8,9]. Furthermore, although cell differentiation trajectories are largely unidirectional in normal development, it is now clear that cellular hierarchies are much more plastic in the context of tissue injury and cancer, and that some microenvironmental signals can induce non-stem cancer cells to acquire a stem-like phenotype[10–13].

Macrophages are emerging as major cellular factors in the tumor ecosystem that can influence the stem phenotype and cancer progression. Previous studies have found that macrophages can enhance breast tumorigenesis[14], are associated with prometastatic changes during tumor progression including increased metastatic seeding[15,16], and promote tumor cell intravasation upon direct contact with tumor cells[17–19]. Furthermore, macrophages were shown to contribute to a niche that can support and maintain a breast CSC (BCSC) phenotype through heterotypic CD90/EphA4 signaling between macrophage and tumor cell[14]. Cell contact-dependent regulation of stemness is a feature of many tissues[10,14] and this process is frequently mediated by activation of the Notch pathway. In mammary gland development, Notch-dependent heterotypic signaling between resident tissue macrophages and mammary stem cells supports survival and function of the normal mammary stem cell[20]. Notch signaling has been implicated in many microenvironmental processes associated with cancer progression such as vascular remodeling, immunosuppression, epithelial to mesenchymal transition (EMT), as well as maintenance of the CSC pool[21]. In vitro modeling recently showed that Notch/Jagged signaling in breast cancer cells can stabilize a hybrid epithelial/mesenchymal state and expand the CSC population[22]. However, much remains to be understood about how and under what circumstances these processes shape CSC properties and hence metastatic dissemination and progression.

The complex interplay of so many potential contributors to the CSC niche makes in vivo observation particularly important as a starting point for understanding and mechanistically dissecting CSC regulation. Existing strategies for understanding CSC biology in vivo have suffered from a number of limitations. Identification of CSCs in digested tumors by flow cytometry for cell surface markers results in loss of both spatial and dynamic information[3]. Lineage tracing approaches retain spatial context and have yielded useful insights such as the multipotent vs. unipotent nature of the mammary stem cell during development[5]

and the origin and fate of CSCs in the intestine[23,24]. However, cell-fate trajectories are generally inferred from sequential static images, and cause and effect relationships as well as reversible phenomena cannot be readily observed.

A complementary approach that preserves both positional and dynamic information is to use intravital imaging coupled with fluorescent reporters that are transiently switched on in specific phenotypic states. To this end, we used an extensively validated lentiviral-based fluorescent reporter that is activated by binding of the stem cell master transcription factors Sox2 and Oct4, or their paralogs, to an artificial enhancer element (SORE6)[25]. This reporter specifically identifies tumor cells with the expected properties of CSC[25]. As the fluorescent protein is destabilized, the acquisition and loss of the stem-like phenotype can be monitored with good kinetic resolution and high selectivity, which is critical to capture the plasticity of CSCs[9]. Coupled with multiphoton microscopy in live animals, this sensor allowed us to address dynamic properties of the CSC in real time in the living tumors.

Here we use intravital imaging at single cell resolution and longitudinally during tumor progression to show the phenotype of individual CSCs in vivo during dissemination and seeding. We use complementary in vitro approaches for detailed molecular mechanistic analysis of the induction of CSC and show that the induction of stemness in cancer cells occurs upon their contact with macrophages via Notch-Jagged signaling. Via this multimodal approach, we identify the mechanism that links the induction of stemness to form CSCs, intravasation, and dissemination, and investigate the extraordinary enrichment of CSCs during the early steps of metastasic progression. Furthermore, we explore whether similar relationships between tumor cell dissemination and density of CSCs occur in breast cancer patients.

## Results

**The SORE6>GFP sensor identifies breast cancer cells with properties of CSCs.** To characterize the CSC population in vivo in the living animal and in situ in tissue, we used our previously validated "SORE6" CSC biosensor[25], with minor modifications to enable detection in formalin-fixed paraffin-embedded (FFPE) tissue (see Supplementary Fig. S1a and "Methods"). In this sensor ("SORE6>GFP"), six repeats of a composite SOX2/OCT4 response element coupled to a minimal cytomegalovirus promoter (minCMV) drives expression of a destabilized fluorescent protein following activation by the CSC master transcription factors, SOX2 and OCT4, or their paralogs. A parallel construct ("minCMV>GFP") lacking the SOX2/OCT4 response elements was used as a control for fluorescence-activated cell sorting (FACS) gating and setting microscopy detection thresholds. SORE6+ cells were defined as cells containing the SORE6>GFP sensor, whose green fluorescent protein (GFP) expression exceeded the threshold set using the control construct. Conversely, SORE6− cells contained the SORE6>GFP sensor but did not activate it above threshold levels due to the absence of the SOX2/OCT4 transcription factor activity. Using an in vivo limiting dilution assay, we confirmed that SORE6+ cells are significantly enriched (7.8-fold) for tumor-initiating activity in the MDA-MB-231 subline used here (Supplementary Fig. S1b). This level of enrichment is similar to the enrichment (9.5-fold) that we previously found using the original sensor in MDA-MB-231 cells and confirmed by 5 passages of serial transplantation[25]. It is also comparable to the enrichment found by others (8.5-fold) using a different (integrin-based) sorting strategy to identify CSCs in the MDA-MB-231 model[26]. Moreover, we demonstrated greatly enhanced chemoresistance (Supplementary Fig. S1c) and tumorsphere-forming ability

(Supplementary Fig. S1d) in SORE6+ cells, properties associated with the stem phenotype[5], as well as increased expression of canonical stem cell transcription factors (Supplementary Fig. S1e). Importantly, we also found that SORE6+ (compared to SORE6−) cells express significantly higher levels of the transcription factor Snail1, which is involved in EMT. This indicates that SORE6+ cells have also activated an embryonic program critical for invasion and metastatic dissemination[27–29]. Knockdown of endogenous Oct4 caused a 95% reduction in the number of SORE6+ cells, confirming that expression of the sensor is absolutely dependent on the presence of this stem cell transcription factor (Supplementary Fig. S1f). Furthermore, we evaluated the expression of relevant cell surface and functional markers that conventionally identify BCSCs[2,30–32]. We found a highly statistically significant overlap between SORE6+ cells and CD133+ cells (Supplementary Fig. S1g), and between SORE6+ cells and ALDH+ cells as assessed by the Aldefluor assay (Supplementary Fig. S1h). CD44 is expressed on >95% of MDA-MB-231 cells and is not a good CSC marker in this model[33]. Thus, by multiple independent criteria, we have shown that the SORE6>GFP sensor identifies a cell population that is enriched for CSC properties in this model.

**BCSCs are a minority population in vivo in primary tumors.** To address CSC representation in primary tumors, orthotopic xenograft breast tumors were generated from MDA-MB-231 cells expressing the SORE6>GFP CSC sensor and a constitutive tdTomato volume marker of all tumor cells. Using fixed-frozen tumor imaging, we found that the SORE6+ cells (i.e., GFP+ cells) comprised a minority population in primary tumors compared to all tumor cells (Fig. 1a). FACS analysis of primary tumors also showed that the CSC population was a minority population of cancer cells (Supplementary Fig. S1i). We also investigated the representation of SORE6+ CSCs in FFPE xenograft tissues. As FFPE destroys the fluorescent signal (destabilized form of copepod GFP (dsCopGFP) and tdTomato), we immunostained for the FLAG tag on the SORE6 biosensor (see Supplementary Fig. S2a, b for FLAG antibody characterization in vitro and in vivo, respectively). In severe combined immunodeficient (SCID) mouse hosts, tumor tissue expressing the SORE6 construct showed a small population of FLAG+ single CSCs distributed throughout the tumor tissue (Fig. 1b), which were not seen in tumor tissue expressing the minCMV>GFP control construct. Similar results were seen in an independent experiment in nude mouse hosts (Supplementary Fig. S2c).

To visualize SORE6+ CSCs in living tissue in mice, we imaged primary tumors expressing the SORE6 construct using intravital imaging through implantable mammary imaging windows (see "Methods" section for details). The tumor expressing control vector (minCMV>GFP) was used to determine the background GFP signal (Supplementary Fig. S3). Live intravital imaging confirmed the presence of GFP+ CSCs in tumors with tumor cells harboring the SORE6>GFP sensor, compared to tumors harboring minCMV>GFP control (Fig. 1c). We quantified SORE6+ CSCs in fixed and live tissue using all three different methodologies (fixed-frozen, FFPE, and live intravital imaging) described above and found that in vivo SORE6+ CSCs are on average between 0.7–2.5% of the total tumor cells in the primary tumor, constituting a minority population (Fig. 1d), as predicted by the CSC hypothesis[3].

**BCSCs exhibit the slow-migratory phenotype and contain invadopodia.** A major advantage of the live-imaging approach is that it allows visualization of dynamic features of the CSC

population. To investigate the migratory phenotypes of CSC in the live primary tumor, SORE6+ CSCs were imaged in vivo using time-lapse intravital multiphoton microscopy. Tumor cells in vivo are heterogeneous with respect to their motility, with the majority of tumor cells being non-motile[34–36]. In the MDA-MB-231 cell xenografts, we observed two distinct migratory phenotypes within the motile tumor cell population, based on locomotion speed of tumor cell; the fast (migrating rapidly toward blood vessels) and slow (perivascular, invadopodium-rich, invasive) phenotypes, as documented previously[17,35,36]. Among motile cells, we found that the high-speed migration phenotype was more characteristic of SORE6− (non-CSCs) cells, whereas SORE6+ CSCs showed the slow-migrating phenotype (Fig. 2a, Supplementary Fig. S4a, and Supplementary Movies 1 and 2). We quantified single tumor cell speeds and found that compared to non-CSCs, which moved at high speeds of ~1 μm/min, CSCs moved approximately five times slower (~0.2 μm/min) (Fig. 2b).

In the in vitro setting, invadopodia (invasive protrusions) form only on the ventral surface of the cell, as that is where the cell contacts the extracellular matrix (ECM). In contrast, in the three-dimensional (3D) tumor microenvironment in vivo, cells are surrounded by ECM and invadopodia are seen extending in all directions[36–39]. To investigate whether CSCs make invadopodia, we imaged SORE6+ CSCs at high resolution in vivo. We observed that CSCs make oscillatory cellular protrusions directed toward ECM fibers (Fig. 2c and Supplementary Movie 3) and blood vessels (Fig. 2d and Supplementary Movies 4 and 5). In addition, when the SORE6+ cells are far away from the blood vessel, they also make protrusions (Supplementary Movie 6 and Supplementary Fig. S4b), similar to the previously described oscillatory invadopodia associated with the slow-migratory phenotype of disseminating tumor cells[36,40,41]. Kymograph analysis and the line profile through the protrusions showed the highly dynamic oscillatory nature of invasive invadopodial protrusions (Fig. 2e). In contrast, non-CSCs, whether close to blood vessels or away from them, did not show oscillatory invadopodial protrusions (Fig. 2f, Supplementary Fig. S4c, d, and Supplementary Movies 7 and 8). Quantification of these protrusions showed that compared to non-CSCs, CSCs display on average six protrusions per hour (i.e., protrusion period of ~10 min) (Fig. 2g), in agreement with previously published values[37,40]

To determine whether the CSC protrusions seen in vivo are indeed invadopodia with the canonical cortactin-Tks5 core and ECM-degrading activity[36,37,42,43], fixed tissue sections were stained with FLAG antibody to identify CSCs, together with cortactin and Tks5 antibodies. We found that both cortactin and Tks5 colocalized to protrusions on SORE6+ CSCs (Fig. 2h, i), indicating these CSC protrusions are invadopodia. A few SORE6− non-CSCs also showed cortactin and Tks5 colocalized invadopodia protrusions (Supplementary Fig. S4j). We counted the number of invadopodia in CSC and non-CSC cell areas using both an automated and manual method (see "Methods"), and found that compared to non-CSCs, CSCs contain significantly higher numbers of invadopodia (Fig. 2j and Supplementary Fig. S4f). To check whether CSC invadopodia have ECM-degrading activity, we utilized a cleaved collagen antibody which detects degraded collagen in vivo[38]. Triple immunofluorescence (IF) staining with cortactin, cleaved collagen, and FLAG antibodies showed colocalization of cortactin positive protrusions in stem cells with degraded collagen in vivo (Fig. 2k, l). Quantification of degraded collagen showed that CSCs have higher degradation activity than non-stem cells (Fig. 2m and Supplementary Fig. S4g).

To further assess the matrix degradation potential of stem cells vs. non-stem cells, we performed an in vitro invadopodium-dependent degradation assay[37]. SORE6+ CSCs or SORE6− non-CSCs were plated separately on fluorescent gelatin overnight and

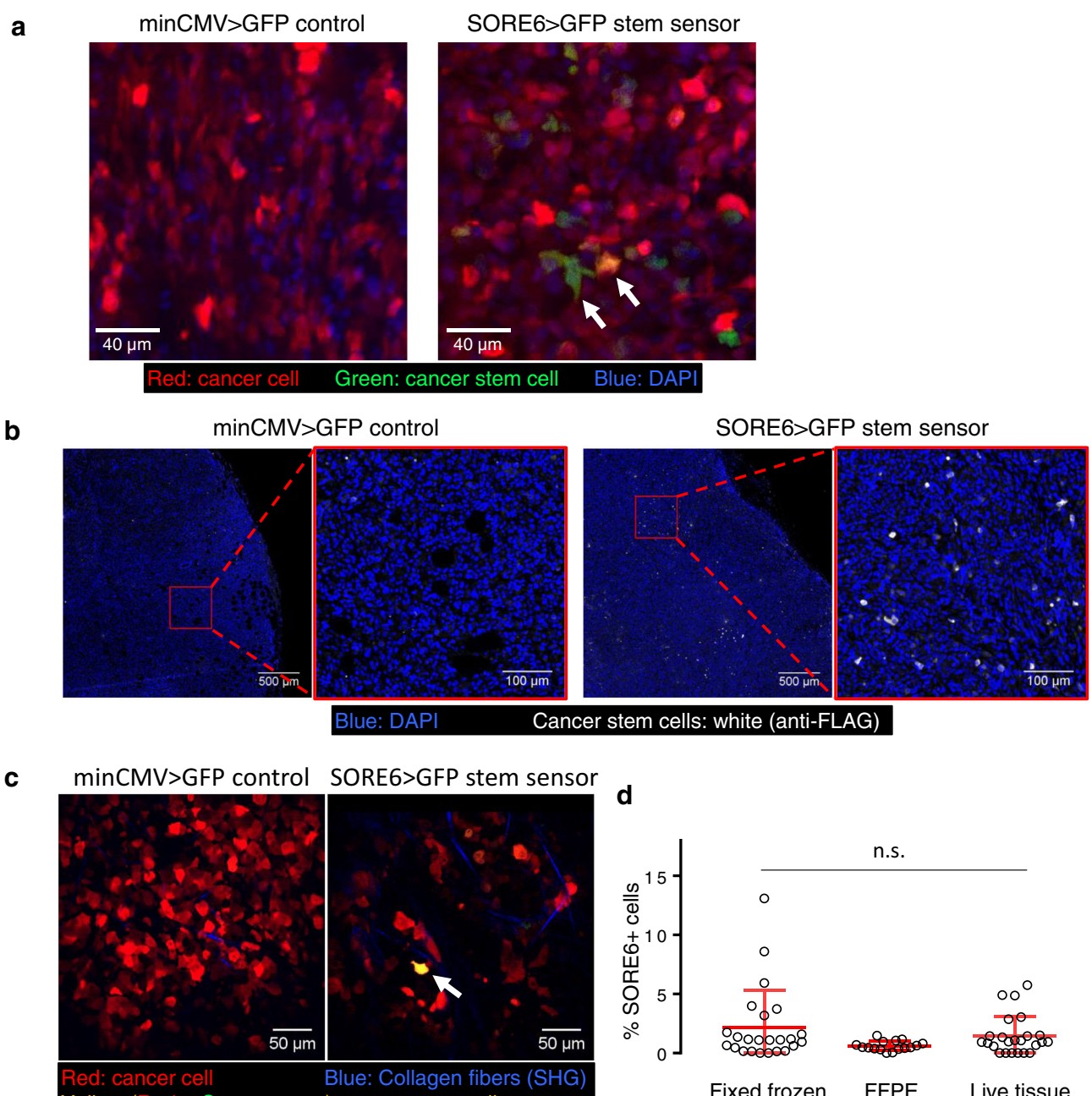

**Fig. 1 Breast carcinoma stem cells are a minority population in the primary tumor in vivo. a** Representative images of fixed-frozen tumor tissues. Left panel: tdTomato-minCMV>GFP vector control tumor tissue; right panel: tdTomato-SORE6>GFP reporter MDA-MB-231 tumor tissue. Arrows point to two examples of GFP+/SORE6+ stem cells, which appear green to yellow due to varying levels of tdTomato (volume marker) intensity in the cell. **b** In situ immunofluorescence staining for the FLAG tag (white) on the GFP reporter in FFPE tumor tissues identifies SORE6+ stem cells as a minority population in fixed tissues in SCID host. Two left panels: tdTomato MDA-MB-231 minCMV>GFP vector control tumor tissue; two right panels: tdTomato MDA-MB-231 SORE6>GFP reporter tumor tissue. Nuclei were stained with DAPI (blue). **c** Intravital multiphoton microscopy identifies SORE6+ stem cells as a minority population (white arrow) in living mouse. Left panel: tdTomato MDA-MB-231 minCMV>GFP vector control tumor tissue; right panel: tdTomato MDA-MB-231 SORE6>GFP reporter tumor tissue. SHG: second harmonic generation. **d** Quantification of % SORE6+ stem cells in breast tissue using the three different methodologies described in **a–c**. $n = 24$ fields of $330 \times 330\ \mu m^2$ from 3 mice for fixed-frozen tissues; $n = 16$ fields of $500 \times 500\ \mu m^2$ from 3 mice for FFPE tissues; $n = 24$ fields of $340 \times 340\ \mu m^2$ from 3 mice for live tissue. Scatter plot showing mean ± SD, one-way ANOVA n.s. $p = 0.09$.

stained for invadopodial markers cortactin and Tks5. We found cortactin- and Tks5-positive invadopodia actively degrading the underlying matrix (Supplementary Fig. S4h). Degradation area quantification showed that CSCs have higher matrix degrading activity compared to the non-CSCs (Supplementary Fig. S4i). Our in vivo and in vitro results together indicate that CSCs have increased numbers of invadopodia with extracellular matrix

degradation activity, a key property needed for the transendothelial migration step of intravasation during the dissemination of tumor cells in the metastatic cascade[42].

**Macrophages regulate stemness in breast cancer cells**. As macrophages have been strongly implicated in promoting metastatic progression[14–19], we investigated the relative spatial

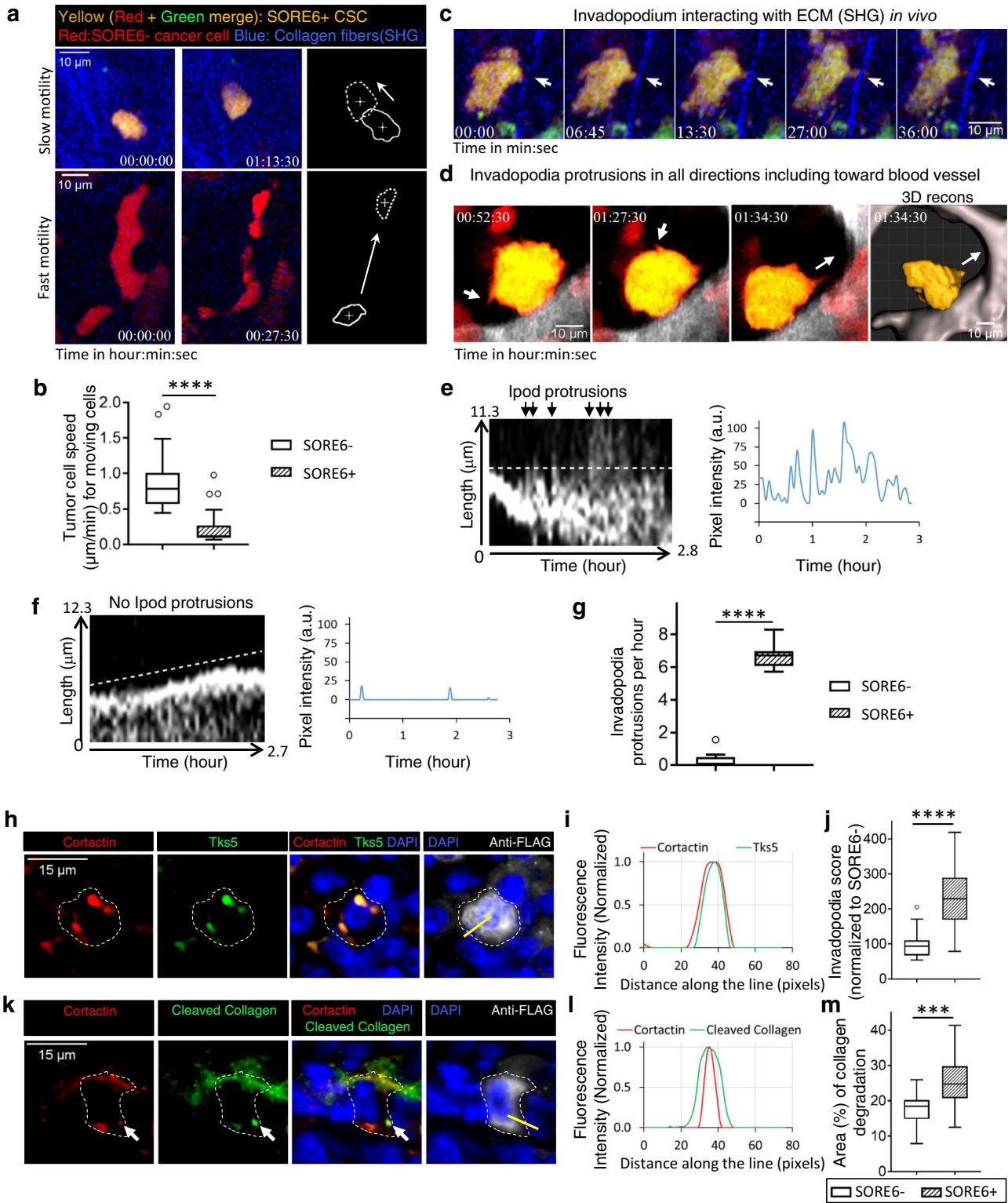

distribution of CSCs and macrophages by intravital imaging of SORE6+ CSCs in primary mammary tumors in mice with (cyan fluorescent protein (CFP)-expressing macrophages (MacBlue host mice) (Fig. 3a). We found in live tumors that 60–70% of the CSCs are in direct physical contact with a macrophage (Fig. 3c). We confirmed our findings in fixed-frozen tumor tissue, where we identified macrophages with an anti-Iba1 antibody (Fig. 3b) and again found that there are more than two times more CSCs in direct physical contact with a macrophage than not

(Supplementary Fig. S5a). In contrast, only about 30% non-CSCs were found to be in contact with a macrophage (Fig. 3c).

To evaluate whether this interaction between macrophages and CSCs was causally associated with stemness in vivo, we depleted macrophages in tumor tissues with clodronate treatment[35]. In vivo intravital imaging with micro-cartography-based localization[44] was performed to image the same tumor fields over 2 days in mice bearing primary mammary tumors expressing the SORE6 sensor with daily clodronate treatment. Clodronate

**Fig. 2 Breast carcinoma stem cells exhibit the slow-migratory, invadopodium-rich phenotype in vivo. a** Examples of slow motility of CSC (top panel, Supplementary Movie 1) and fast motility of non-CSC (bottom panel, Supplementary Movie 2). Rightmost panels show cell outlines in first (solid) and last (dotted) frames of the movies with crosses marking cell centroids. White arrows point in the direction of cell movement. **b** Quantification of cell speeds for non-CSCs (SORE6−) and CSCs (SORE6+). $n = 25$ SORE6−, 32 SORE6+ cells in 3 mice; unpaired two-sided Mann–Whitney test, ****$p < 0.0001$. **c** SORE6+ CSC (yellow) showing dynamic invadopodial protrusion (white arrows) interacting with ECM fiber (blue) in vivo (Supplementary Movie 3). **d** SORE6+ CSC (yellow) showing dynamic invadopodial protrusions (white arrows) coming out in all directions, including toward the blood vessel shown in gray (third panel). The rightmost panel (from Supplementary Movie 5) shows 3D reconstruction of CSC invadopodium directed toward the blood vessel (white arrow). **e** Kymograph showing the periodicity of invadopodial protrusions (black arrows) from a SORE6+ CSC. Right panel shows the intensity profile through the kymograph (white dotted line). **f** Kymograph of a SORE6− non-CSC leading edge showing no periodic invadopodial protrusions. Right panel shows the intensity profile through the kymograph (white dotted line). **g** Quantification of invadopodia protrusions per hour in SORE6− ($n = 13$ cells from 3 mice) and SORE6+ ($n = 10$ cells from 3 mice) cells; unpaired two-sided Mann–Whitney test, ****$p < 0.0001$. **h** In situ immunofluorescence staining shows that CSCs (anti-FLAG) contain invadopodial protrusions as identified with invadopodial core markers Cortactin and Tks5. White dotted line marks the boundary of the CSC. **i** Cortactin and Tks5 fluorescence intensity plots along the yellow line shown in **h**. **j** Invadopodia score quantification in non-CSCs and SORE6+ CSCs using an automated scoring method. $n = 20$ SORE6 primary tumor fields at ×40 magnification from four mice; two-sided Wilcoxon test, ****$p < 0.0001$. **k** In situ immunofluorescence staining shows that CSC (anti-FLAG) invadopodia have degradation activity associated with them. White arrows indicate the colocalization of freshly cleaved collagen with invadopodia. White dotted line marks the boundary of the CSC. **l** Cortactin and Cleaved collagen fluorescence intensity plots along the yellow line shown in **k**. **m** Quantification of collagen degradation area in non-CSC and CSC areas using an automated scoring method. $n = 16$ SORE6 primary tumor fields at ×40 magnification from four mice; paired two-tailed Student's $t$-test, ***$p < 0.0001$. **b**, **g**, **j**, **m** The boxes indicate 25th–75th percentile interquartile range (IQR) and central line indicates the median. Top and bottom whiskers were plotted using Tukey's method and extend to 75th percentile + 1.5 × IQR and 25th percentile − 1.5 × IQR, respectively. Points below and above the whiskers are drawn as individual dots.

treatment led to significant macrophage depletion in tumor tissue over 2 days (Fig. 3d) and this was associated with significant reduction in CSCs in vivo during the same time period (Fig. 3e and Supplementary Fig. S5b), suggesting the macrophages contribute to the CSC niche. Treatment of MDA-MB-231 cells with clodronate in vitro showed no effect on the number of CSCs or general cancer cell viability and proliferation, indicating that the clodronate-induced reduction in CSCs in vivo was not due to direct effects on the tumor cells or on the sensor (Supplementary Fig. S5c).

Furthermore, we extended the role of macrophages in stemness regulation beyond the MDA-MB-231 human xenograft model to an autochthonous breast cancer model that fully recapitulates the entire cancer progression process, by using the MMTV-PyMT transgenic mouse ("PyMT")[45]. As the lentiviral SORE6 stemness sensor cannot be used directly in this transgenic model, we sought a surrogate marker of stemness. Met-1 cells, which are derived from a PyMT tumor[46], were transduced with the SORE6 and minCMV control sensors, and FACS analysis showed that the SORE6 sensor identifies a minority stem cell population (~8%) in these cultures (Supplementary Fig. S5d), similar to results obtained with MDA-MB-231 cells (Supplementary Fig. S1i). Although Sox2 mRNA was below the limit of detection (Ct value > 35) in these cells, expression of the stem cell transcription factors Sox9 (a Sox2 paralog), Oct4, and Nanog were significantly upregulated in SORE6+ Met-1 cells when compared with SORE6− Met-1 cells (Supplementary Fig. S5e), similar to results obtained in MDA-MB-231 cells (Supplementary Fig. S1e). As Sox9 has previously been implicated in the maintenance of the mammary stem cell state[47,48] and was consistently associated with stem cell features in a panel of breast cancer cell lines[49], we proceeded to use Sox9 expression as a surrogate marker of stemness in the PyMT model. PyMT mice bearing primary tumors were treated with either control phosphate-buffered saline (PBS) or clodronate liposomes, and macrophage (Iba1 immuno-histochemistry (IHC), brown staining, Supplementary Fig. S5f) coverage area was quantified. We found ~50% decrease in macrophage density after clodronate treatment (Fig. 3g), similar to results obtained in MDA-MB-231 primary tumor model (Fig. 3d). To check the effect of clodronate treatment on Sox9Hi stem cells (see "Methods" for Sox9Hi definition), PyMT tissues were stained with Sox9 antibody (Fig. 3f) and Sox9Hi tissue

coverage area was quantified. We found significant reductions (~80%) in Sox9Hi cells after clodronate treatment (Fig. 3h), similar to results obtained in MDA-MB-231 primary tumor model (Fig. 3e). Thus, in two independent models, we observed a reduction in macrophages to be associated with a concomitant reduction in CSCs.

**Macrophage contact induces stemness in breast cancer cells.** The functional link between CSCs and macrophages observed above could result from macrophage-regulated expansion of a pre-existing CSC population, or it could reflect induction de novo of the CSC state in non-CSCs. To distinguish these possibilities, we stained control and clodronate-treated PyMT primary tumor tissues with Ki67 (proliferation marker) and found that macrophage depletion did not affect CSC proliferation in vivo (Supplementary Fig. S6a), indicating that the primary mechanism of macrophage-mediated tumor cell stemness does not involve proliferative expansion of pre-existing CSCs.

To investigate contributing mechanisms further, we performed intravital time-lapse imaging of tumors to observe the response of individual tumor cells to macrophage interactions. Using the SORE6 biosensor, we found examples of non-stem tumor cells converting to stem cells upon macrophage contact in vivo (Fig. 4a upper panels and Supplementary Movies 10 and 11), with the stemness biosensor signal rising significantly above background by ~0.5 h after the first contact with a macrophage (Fig. 4a) and, on average, by 1.5 h for the average of the tumor cell population (Fig. 4b). In contrast, tumor cells that were not contacted by a macrophage did not show stemness induction (Fig. 4a, lower panels).

To confirm and mechanistically dissect these in vivo observations we performed in vitro co-culture experiments. Tumor cells with the SORE6>GFP sensor and macrophages (unlabeled) were co-cultured and imaged live. Similar to our in vivo experiments, we observed induction of the stem phenotype (GFP positivity) in non-stem cells upon macrophage contact (Supplementary Fig. S6b and Supplementary Movie 12). The kinetics of stemness induction in vitro (Supplementary Fig. S6c) were similar to those observed in vivo (Fig. 4b). Only tumor cells that made direct physical contact with macrophages were induced to become SORE6+ (Supplementary Fig. S6b). Moreover, we found that prior co-culture with macrophages for 24 h increased the tumorsphere-initiating activity of the tumor cells (Fig. 4c),

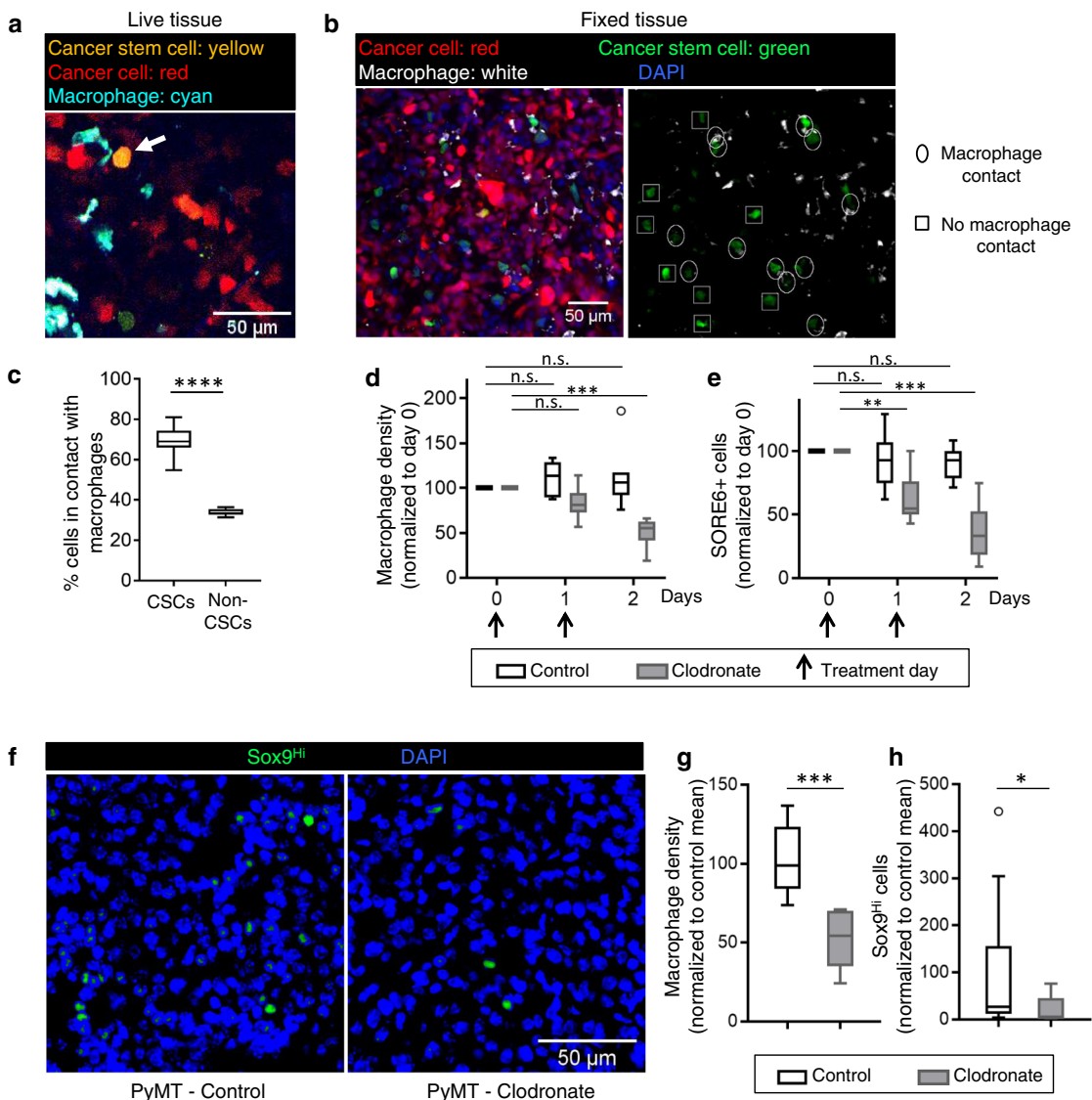

**Fig. 3 Macrophages regulate stemness in breast cancer cells. a** Intravital microscopy image of a CSC (arrow) in contact with a macrophage in primary tumor in a living mouse. **b** Left panel: representative images of fixed-frozen primary tumor tissue showing cancer cell (tdTomato, red), CSC (SORE6+, green), macrophage (Iba1, white), and nuclei (DAPI, blue). Right panel: same as left but lacking tdTomato and DAPI channels, showing CSCs in contact with macrophages (oval), or not in contact with macrophage (square). **c** Quantification of % CSCs and % non-CSCs in direct contact with macrophages or not in live tissue. $n = 42$ CSCs from 17 fields of $340 \times 340\ \mu m^2$ in 4 mice, $n = 602$ non-CSCs from 11 fields of $340 \times 340\ \mu m^2$ in 3 mice; two-sided Fisher's exact test, ****$p < 0.0001$. **d** Quantification of changes in macrophage density in live mammary tumor over time after control PBS liposome or clodronate liposome treatment. Vertical arrows indicate the treatment days. $n = 7$ fields of $512 \times 512\ \mu m^2$ each for control (3 mice) and clodronate (3 mice) treatments. Two-sided Tukey's multiple comparisons test; $p$-values for control: day0 vs. day1 = 0.44, day0 vs. day2 = 0.67; and for clodronate: day0 vs. day1 = 0.09, day0 vs. day2(clodronate) = 0.0005. **e** Quantification of changes in SORE6+ cells in live mammary tumor over time after control PBS liposome or clodronate liposome treatment. $n = 7$ fields of $512 \times 512\ \mu m^2$ each for control (3 mice) and clodronate (3 mice) treatments. Two-sided Tukey's multiple comparisons test; $p$-values for control: day0 vs. day1 = 0.76, day0 vs. day2 = 0.16; and for clodronate: day0 vs. day1 = 0.007, day0 vs. day2 = 0.0007. **f** Images of PyMT primary tumor tissues treated with either PBS liposomes (control) or clodronate liposomes and stained with Sox9 antibody (green) and DAPI (blue). Image shows Sox9Hi cells. **g** Quantification of changes in macrophage density in fixed PyMT mammary tumor after PBS (control) or clodronate liposomes treatment every 2 days for 2 weeks. $n = 11$ (control) and 6 (clodronate) fields of $2$–$3\ mm^2$ tumor tissue from 5 mice (control) and 4 mice (clodronate); two-sided Mann–Whitney test, ***$p = 0.0002$. **h** Quantification of changes in Sox9Hi cells in fixed PyMT mammary tumor after PBS (control) or clodronate liposomes treatment every 2 days for 2 weeks. $n = 17$ (control) and 11 (clodronate) fields of $1000 \times 500\ \mu m^2$ tumor tissue from 5 mice (control) and 4 mice (clodronate); Mann–Whitney two-sided test, *$p = 0.0221$. **c–e**, **g**, **h** The boxes indicate 25th–75th percentile interquartile range (IQR) and central line indicates the median. Top and bottom whiskers were plotted using Tukey's method and extend to 75th percentile + 1.5 × IQR and 25th percentile − 1.5 × IQR, respectively. Points below and above the whiskers are drawn as individual dots.

indicating that the induced stem phenotype is both durable and functional.

To determine whether the induction of stemness was specific to macrophages, we co-cultured tumor cells with either macrophages or endothelial cells. In vitro, the baseline SORE6 positivity for MDA-MB-231 cells varied from ~15% to 40%. In co-cultures, there was consistently a 40% increase in the SORE6+ CSCs over baseline when MDA-MB-231 tumor cells were co-cultured overnight with BAC1.2F5 macrophages compared to tumor cells alone (Fig. 4d). This effect was

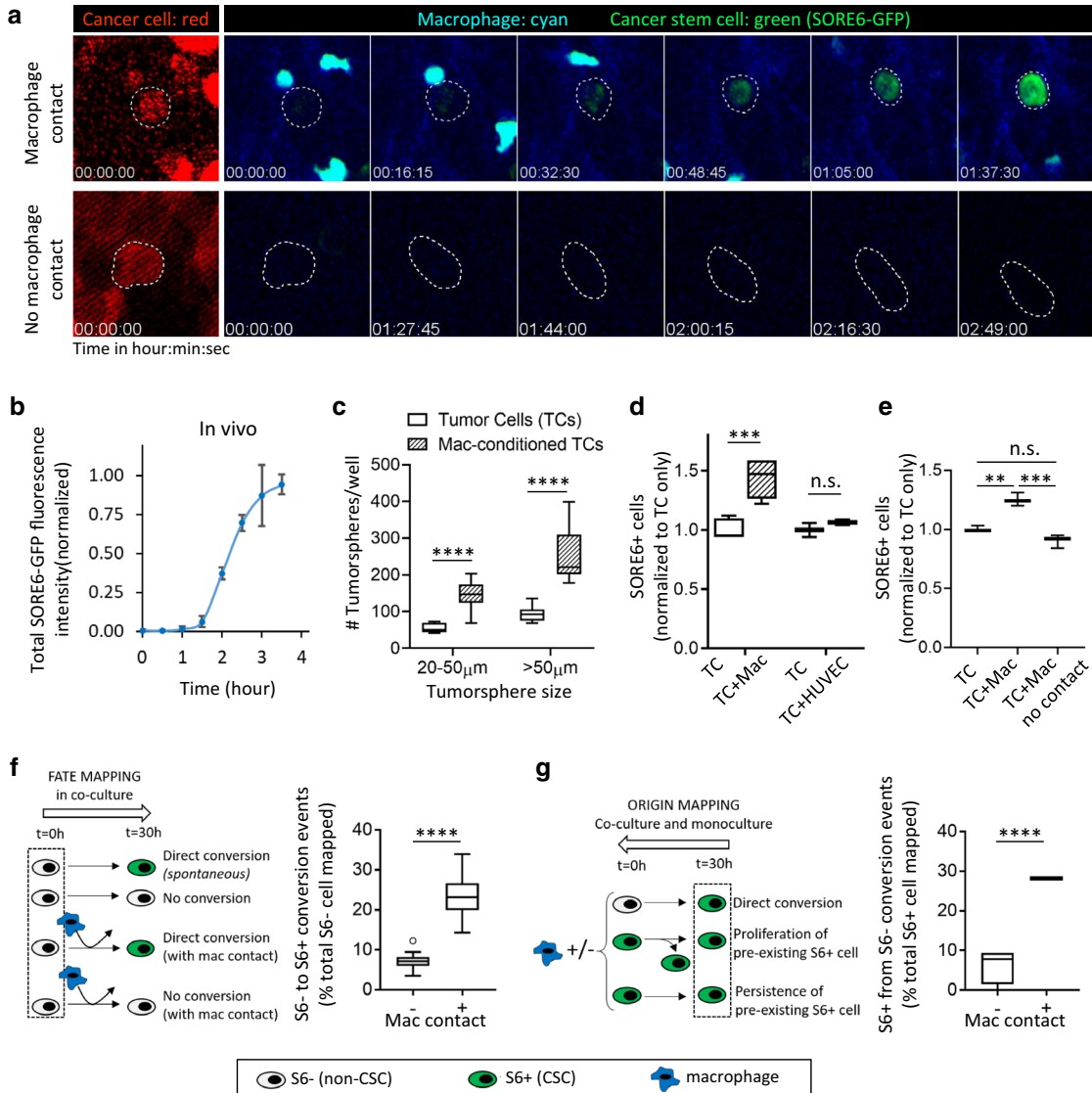

**Fig. 4 Macrophage contact induces stemness in breast cancer cells. a** Upper panels show SORE6>GFP induction (green) in vivo in a tumor cell after macrophage contact (Supplementary Movie 10). Lower panels show lack of stemness induction (no SORE6-GFP increase) in a tumor cell, which is not contacted by a macrophage. White dotted lines show tumor cell outlines, drawn based on the tdTomato volume marker (left panel). **b** Quantification of total SORE6-GFP fluorescence intensity during stemness induction in tumor cells in vivo. Time 0 corresponds to the time when macrophage contacts the tumor cell. $n > 200$ cells analyzed from 10 mice, to identify tumor cells contacting macrophages and showing induced stemness, each data point plotted as mean ± SD. Data normalized to SORE6-GFP value before macrophage contact to set the 0 background and max SORE6-GFP value as 1 after macrophage contact. **c** Tumorsphere-forming activity of MDA-MB-231 cells with or without prior co-culture with BAC cells for 24 h. Multiple unpaired two-sided $t$-tests for each size class, $n = 6$ (TCs), $n = 12$ (MAC-conditioned TCs), ****$p < 0.0001$. **d** In vitro co-culture assay of tumor cells with or without macrophages or endothelial cells. Tumor cells were co-cultured with macrophages or endothelial cells overnight before the measurements were made. TC: MDA-MB-231 tumor cells, Mac: BAC1.2F5 macrophages. $n = 120$ fields (TC + Mac) and 40 fields (TC + HUVEC) at ×20 magnification; unpaired two-tailed Student's $t$-test, ***$p = 0.0001$, n.s. $p = 0.4226$. **e** MDA-MB-231-LM2 cells (TC) were co-cultured with BAC1.2F5 cells (Mac) for 36 h without or with a 3 μm pore insert to separate the cell populations and assessed for the number of SORE6+ cells by flow cytometry. Two-sided Tukey's multiple comparisons test, $n = 3$; ***$p$-values = 0.0013, n.s. = 0.0919, ***$p = 0.0002$. **f** Left: schematic showing the TC + Mac co-culture experimental setup and quantification procedure for the fate mapping of non-CSCs (S6−) in time-lapse movies. Right: quantification of SORE6 induction without or with macrophage contact in TC + Mac co-culture assay. Unpaired two-tailed Student's $t$-test, $n = 113$ S6− cells from 28 fields of $330 \times 330$ μm$^2$, ****$p < 0.0001$. **g** Left: schematic showing the TC alone or TC + Mac co-culture experimental setup and quantification procedure for the origin mapping of CSCs (S6+) in time-lapse movies. Right: quantification of SORE6 induction in TC alone (−) or TC + Mac (+) co-culture assay, respectively. Unpaired two-tailed Student's $t$-test, $n = 4$ wells/condition for a total of 128 cells traced/condition, ****$p < 0.0001$. **c–g** The boxes indicate 25th–75th percentile interquartile range (IQR) and central line indicates the median. Top and bottom whiskers were plotted using Tukey's method and extend to 75th percentile + 1.5 × IQR and 25th percentile − 1.5 × IQR, respectively. Points below and above the whiskers are drawn as individual dots.

specific to macrophages, as we did not see any increase in SORE6+ CSCs when tumor cells were co-cultured with human umbilical vein endothelial (HUVEC) cells, the other cell type most frequently in contact with disseminating tumor cells at perivascular regions of the tumor[17] (Fig. 4d). The macrophage co-culture assay was further repeated in two additional breast tumor models: MDA-MB-231-LM2 (baseline %SORE6 positivity $21.2 \pm 0.5\%$) and 4T1 (baseline %SORE6 positivity $3.3 \pm 0.4\%$). We found ~20% and 40% increase in SORE6+ CSCs in these two models resulting from direct contact with macrophages (Fig. 4e and Supplementary Fig. S6d), respectively. In addition, we used human M2-like macrophages in the co-culture assay with MDA-MB-231 cells and found ~25% increase in SORE6+ CSCs (Supplementary Fig. S6e). The increase in CSCs in macrophage/tumor cell co-cultures was not seen if macrophages and tumor cells were separated by a 3 μm pore-size filter (Fig. 4e), confirming that direct cell contact, not conditioned medium, is required for induction of stemness. Taken together, these results indicate a general role for macrophage contact in inducing stemness de novo.

To rigorously address the quantitative contribution of this direct induction of stemness in non-stem cancer cells upon macrophage contact, we performed cell-fate and cell-origin mapping experiments from video microscopy movies of the co-cultures in vitro. Assessing the fate of randomly selected SORE6− non-CSCs in macrophage/tumor cell co-cultures, we showed that SORE6− cells that underwent direct contact with macrophages in the time period had a >4× increase in frequency of conversion to SORE6+ CSCs compared with SORE6− cells that did not contact macrophages, with 23% of the non-stem cancer cells that contacted a macrophage undergoing conversion to a stem cell phenotype in the time-frame analyzed (Fig. 4f). In a complementary and independent experiment, we mapped the origin of SORE6+ CSCs that were present at the end of the culture period in either tumor cell/macrophage co-cultures or cultures of tumor cells alone. We similarly found that the macrophage co-cultures showed a >4× increase in the direct non-stem to stem conversion frequency (Fig. 4g).

Consistent with the finding above that changes in macrophage numbers are not associated with changes in the proliferation of CSCs in PyMT tumors (Supplementary Fig. S6a), we found that macrophage contact in vitro did not change the proliferation of the CSCs, indicating that macrophages do not induce the expansion of pre-existing CSCs (Supplementary Fig. S6f). Thus, from our in vitro and in vivo experiments, we conclude that contact with macrophages increases the CSC population through direct de novo induction of the stem phenotype in non-stem cells, without any impact on the proliferation of pre-existing stem cells.

**Notch1-Jagged signaling regulates macrophage contact-induced stemness in breast cancer cells.** The Notch signaling pathway was recently shown to be involved in mediating cross-talk between macrophages and the normal mammary stem cell during mammary gland development[20]. Furthermore, Notch/Jagged1 signaling can promote CSC traits in homotypic cultures of breast cancer cells[22]. Given studies showing Jagged1 (a Notch ligand) expression in macrophages[50,51], and Notch1-mediated induction of Mena[INV] expression, which is required for intravasation in association with macrophages at tumor microenvironment of metastasis (TMEM)[19], we investigated whether Notch signaling plays a role during the stemness induction by macrophages that we observed in mammary tumors in vivo. Pharmacologic inhibition of Notch signaling with either N-[N-(3,5-Difluorophenacetyl)-L-alanyl]-S-phenylglycine-t-butyl ester (DAPT), a γ-secretase inhibitor that blocks Notch

signaling[52], or a more specific Notch transcription factor complex inhibitor, SAHM1[53], led to decreases in SORE6+ CSCs in cancer cell-macrophage co-cultures (Fig. 5a, b). To address possible off-target effects of DAPT inhibition[54], we knocked down Notch1 protein in cancer cells using Notch1 small interfering RNA (siRNA) (Fig. 5c) and co-cultured them with macrophages. We found that macrophage contact is unable to induce stemness in Notch1 signaling defective cancer cells (Fig. 5d). Moreover, the reduction in stemness is not due to the reduced motility of cancer cells upon Notch1 inhibition and subsequent reduced interaction with the macrophages, as Notch1 inhibition in MDA-MB-231 cancer cells does not affect their motility[19]. We did not see any difference in Notch1 expression in SORE6+ cells compared to the SORE6− cells (Supplementary Fig. S7a), indicating that macrophages do not induce stemness in non-CSC by increasing expression of Notch1 but rather that macrophages use Notch1-mediated signaling to induce stemness in non-CSCs. As there are many Notch ligands—Jag1, Jag2, DLL1, DLL3, and DLL4[54]—we wanted to identify which Notch ligand(s) is/are involved in stemness induction in cancer cells. We evaluated the macrophages (BAC1.2F5), which we used in our co-culture experiments, for Notch ligand expression and found that these macrophages primarily express Jag1 and Jag2, with an order of magnitude lower amounts of DLL1, DLL3, and DLL4 (Supplementary Fig. S7b). We found a significant increase in SORE6+ CSCs after Notch signaling activation with Jagged1 but not with scrambled peptide (Fig. 5e). Conversely, we used Jag1KO BAC1.2F5 macrophages[51] in tumor cell-macrophage co-culture assay and found that Jag1 deletion from macrophages leads to complete inhibition of CSC induction caused by macrophages (Fig. 5f). Next, we evaluated the roles of both Jag1 and Jag2 in CSC induction by inhibiting Notch signaling with Jag1 and Jag2 blocking antibodies in the tumor cell-macrophage co-culture assay. We found that blocking either Jag1 or Jag2, or Jag1 and Jag2 together leads to complete inhibition of CSC induction caused by macrophages (Fig. 5g), suggesting that DLL ligands play no role in macrophage contact-mediated CSC induction. Finally, we inhibited Notch signaling in vivo by treating PyMT mice with DAPT. DAPT activity in vivo was confirmed by goblet cell hyperplasia in the intestinal crypts (Supplementary Fig. S7c)[55–57], a decrease in nuclear Notch1 intracellular domain (NICD) levels (Supplementary Fig. S7d, e) and a decrease in the expression of Notch1 target protein Hes1 (Supplementary Fig. S7f, g). We then investigated the effect of DAPT on Sox9[Hi] CSCs in PyMT tumors. Notch inhibition led to a five-fold decrease in CSCs in DAPT-treated mice compared to the vehicle control-treated mice (Fig. 5h), while not affecting the percentage of macrophages in the tumor (Fig. 5i).

These results, together with the clodronate results in Fig. 3g, h, confirm that stemness induction is dependent on physical contact involving Notch1-Jagged1/2 signaling between cancer cells and macrophages. However, further experiments are required to identify specific Notch ligands in the in vivo setting.

**BCSCs preferentially associate with and intravasate at TMEM doorways.** The cancer cells capable of forming invadopodia are crucial for the function of intravasation portals called TMEM through which cancer cells enter the blood vessels and systemically disseminate[15,17,44]. TMEM is a triple cell complex composed of a Mena[Hi] tumor cell, a macrophage and a blood vessel endothelial cell, all in direct physical contact on the blood vessel surface. TMEM doorways are the only known portals for tumor cell intravasation in mammary tumors[17,58]. The density of TMEM doorways is a clinically validated prognostic marker of metastatic recurrence in breast cancer patients[59–61]. As CSCs

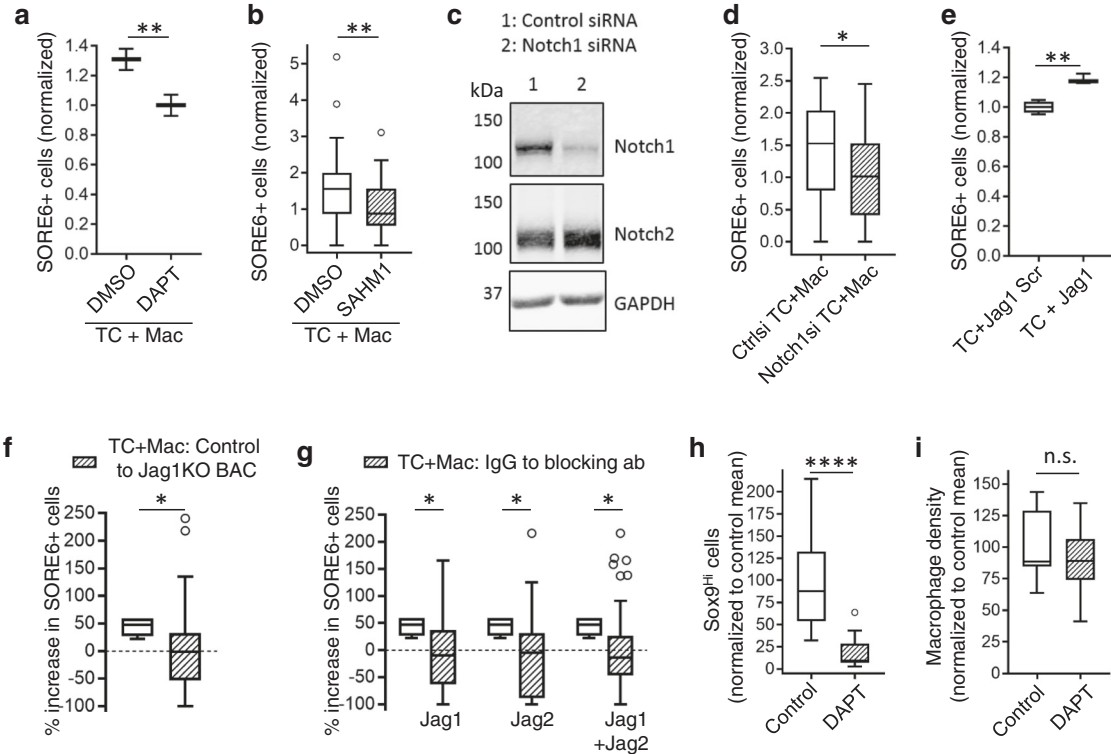

**Fig. 5 Notch1-Jagged signaling regulates macrophage contact-induced stemness in breast cancer cells. a** Quantification of SORE6+ cells in TC + Mac co-cultures treated with DMSO or DAPT. $n = 60$ fields at ×20 magnification; data normalized to the DAPT mean; unpaired two-tailed Student's $t$-test, **$p = 0.0061$. **b** Quantification of SORE6+ cells in TC + Mac co-cultures treated with DMSO or 20 μM SAHM1. $n = 48$ (DMSO), 47 (SAHM1) fields at ×20 magnification; data normalized to the SAHM1 mean; two-sided Mann–Whitney test, **$p = 0.0025$. **c** Western blotting showing Notch1 KD in Notch1 siRNA-treated SORE6 MDA-MB-231 cells after 48 h. It is noteworthy that Notch2 levels are unaffected by Notch1 siRNA. Source data are provided as a Source Data file. **d** Quantification of SORE6+ cells in TC + Mac co-culture assay. Tumor cells were treated with either control siRNA or Notch1 siRNA for 36 h and co-cultured with macrophages overnight. Measurements were made 48 h post siRNA transfection. $n = 32$ fields of $330 × 330 \mu m^2$; data normalized to the Notch1si condition mean; unpaired two-tailed Student's $t$-test, *$p = 0.0232$. **e** Quantification of SORE6+ cells in tumor cells (MDA-MB-231-expressing tdTomato and with SORE6>GFP) treated with either Jagged1 scramble peptide or functional Jagged1 peptide. $n = 80$ fields at ×20 magnification; data normalized to the TC + Jag1 scramble mean; unpaired two-tailed Student's $t$-test, **$p = 0.0012$. **f** Change in % SORE6+ cells after co-culture with either wild-type BACs compared to tumor cells alone (40% increase, data from Fig. 4d, unshaded box) or Jag1KO BACs compared to wild-type BACs (no change, shaded box). $n = 50$ (wild-type BACs) and 53 (Jag1KO BAC) fields at ×20 magnification; two-sided Mann–Whitney test, *$p = 0.033$. **g** Change in % SORE6+ cells after co-culture with either wild-type BACs compared to tumor cells alone (40% increase, data from Fig. 4d, unshaded boxes) or Jag1/Jag2 antibody treatments compared to IgG-treated sample (no change, shaded boxes). $n = 54$ (IgG), 57 (Jag1), 56 (Jag2), and 55 (Jag1 + Jag2) fields at ×20 magnification; two-sided Mann–Whitney test, *$p$-values: Jag1 = 0.035, Jag2 = 0.013, Jag1 + Jag2 = 0.02. **h** Quantification of changes in Sox9$^{Hi}$ cells in fixed PyMT mammary tumor after PBS (control) or DAPT treatment. $n = 5$ and 7 mice for control and DAPT conditions; two 2–6 mm$^2$ tumor tissue areas were analyzed in each mouse; unpaired two-tailed Student's $t$-test, ****$p < 0.0001$. **i** Quantification of changes in macrophage density in fixed PyMT mammary tumor after PBS (control) or DAPT treatment. $n = 5$ and 7 mice for control and DAPT conditions; two 2–6 mm$^2$ tumor tissue areas were analyzed in each mouse; unpaired two-tailed Student's $t$-test, n.s. $p = 0.41$. In all quantifications, boxes indicate 25th–75th percentile interquartile range (IQR) and central line indicates the median. Top and bottom whiskers were plotted using Tukey's method and extend to 75th percentile + 1.5 × IQR and 25th percentile − 1.5 × IQR, respectively. Points below and above the whiskers are drawn as individual dots.

form invadopodia, which are required for TMEM function and transendothelial migration, macrophage contact induces stemness in cancer cells, and the areas around TMEM are perivascular macrophage-rich compartments[17,62], we investigated whether CSCs are enriched at and around TMEM doorways using intravital imaging. We quantified the location of SORE6+ stem cells with respect to TMEM doorways or blood vessels away from TMEM doorways by performing four-color time-lapse intravital imaging of mammary tumors expressing SORE6>GFP in a Rag2KO/MacBlue host, where macrophages are labeled in CFP[17]. We found CSC enrichment in perivascular regions near TMEM (Fig. 6a).

To further investigate stem cell associations with TMEM doorways, we performed triple-IHC for TMEM markers (Mena, Iba1, and CD31) in fixed tumors. TMEM are identified as three cells in direct and stable physical contact by triple staining of

TC (Mena$^{Hi}$), macrophage (Iba1), and blood vessel (CD31). A sequential tissue section was stained with FLAG antibody to identify SORE6+ CSCs. TMEM IHC and FLAG IF images were then aligned and superimposed. As suggested by imaging of live tissue (Fig. 6a and Supplementary Movie 13), we found CSCs in association with TMEM doorways (Fig. 6b), with ~38% of TMEM doorways being assembled with a CSC (Supplementary Fig. S8a).

TMEM IHC and FLAG IF images were further analyzed to determine the percentage of CSCs relative to TMEM doorways and blood vessels without TMEM doorways. Approximately 50% of blood vessels contain at least one TMEM and 50% are free of TMEM. An enrichment of CSCs was observed in the tumor areas close to TMEM doorways (Fig. 6c, d) resulting in about seven times higher percentage of CSCs compared to >300 μm away from TMEM doorways (Fig. 6d). We also determined the

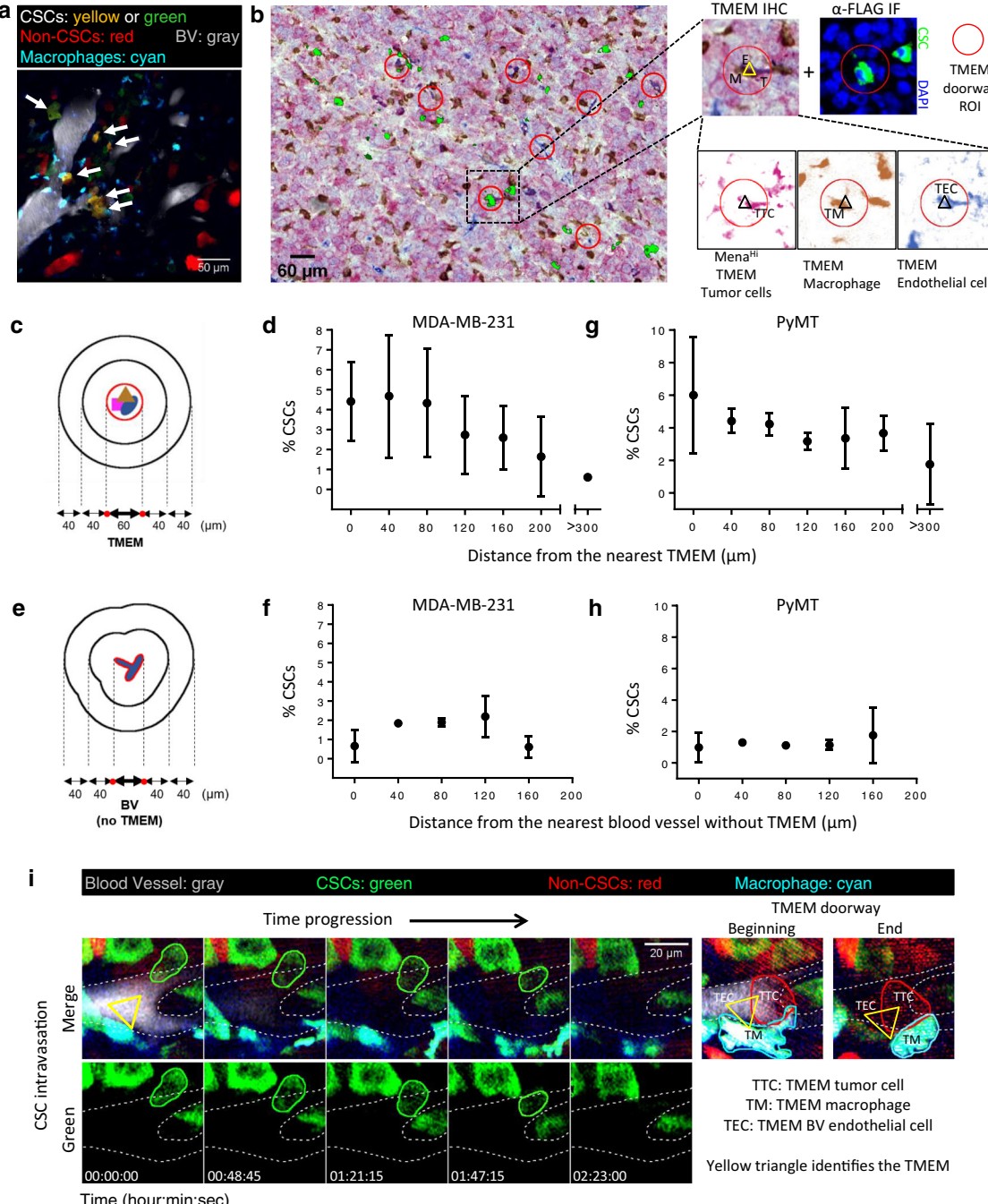

percentage of CSCs relative to blood vessels without TMEM doorways and did not see any enrichment (Fig. 6e, f), indicating that CSCs specifically cluster around TMEM doorways. We repeated our analysis in another mammary tumor type (PyMT), with Sox9 marking the CSCs. Again, we found CSC enrichment (3.4×) around TMEM doorways compared to the average seen throughout the tumor at sites not near TMEM (Fig. 6g). As seen in the MDA-MB-231 tumors, in PyMT tumors we also did not observe any CSC enrichment around blood vessels not containing any TMEM (Fig. 6h). These results suggest that the induction of CSCs occurs in association with the macrophages proximal to TMEM doorways[17,62], leading to increased CSCs in association with each TMEM doorway. This raises the key question of whether TMEM doorways are not only niches for CSCs but also intravasation portals for seeding of metastatic CSCs.

To determine whether the CSCs that are associated with TMEM doorways actually intravasate at TMEM, we used intravital imaging as described previously to directly observe tumor cell intravasation in primary tumors[17]. We observed intravasation of CSC, identified by expression of the SORE6+ biosensor (Fig. 6i and Supplementary Movie 14). Furthermore, high-resolution imaging of mice with CFP macrophages showed that this intravasation occurred at TMEM doorways (Fig. 6i right panels show labeled TMEM doorway). As previous work showed that TMEM doorways are the only sites for cancer cell intravasation in mammary tumors[17,58], CSC intravasation in areas away from TMEM doorways was not evaluated.

**CSCs are highly enriched in circulation and at their first arrival in the lung, where they seed new micro-metastases**. Consistent

**Fig. 6 Breast carcinoma stem cells preferentially associate with and intravasate at TMEM doorways. a** Still image from Supplementary Movie 13 of SORE6>GFP xenograft MDA-MB-231 tumor in Rag2KO mice, showing CSCs (yellow: tdTomato volume marker + SORE6-GFP or green: SORE6-GFP only) enriched in perivascular regions and in contact with perivascular macrophages (cyan). Blood vessels shown in gray. **b** A composite image of TMEM staining (pink: Mena^Hi tumor cells, brown: macrophages, blue: endothelial cells) and FLAG+ stem cells (green). TMEM doorway ROI are highlighted with red circles. Zoomed images show an example of FLAG+ stem cell (α-FLAG IF panel) being associated with the TMEM doorway (TMEM IHC panel) when panels are aligned. Triple-stained TMEM IHC image is further deconvolved into individual channels and shows that TMEM (marked with triangle) is composed of a TMEM tumor cell (TTC) (Mena^Hi), TMEM macrophage (TM) (Iba1), and an TMEM endothelial cell (TEC) (endomucin). Image is a representative of similar results found in $n = 4$ mice. **c** The schematic shows how the region containing TMEM within the section is assigned a boundary surrounded by concentric contours radiating out from the TMEM in 40 µm intervals. **d** Percentage CSCs distribution from the nearest TMEM in fixed MDA-MB-231 mammary tumor tissue. $n = 4$ fields of $2.2 \times 2.2$ mm$^2$ from 4 mice; data plotted as mean ± SD. **e** Schematic shows how the region containing a blood vessel without TMEM within the section is assigned a boundary surrounded by concentric contours radiating out from the blood vessel in 40 µm intervals. **f** Percentage CSCs distribution from the nearest blood vessel without TMEM in fixed MDA-MB-231 mammary tumor tissue. $n = 4$ fields of $2.2 \times 2.2$ mm$^2$ from four mice; data plotted as mean ± SD. **g** Percentage CSCs (anti-Sox9 stained) distribution from the nearest TMEM in fixed PyMT mammary tumor tissue. $n = 8$ fields of 2–3 mm$^2$ from 4 mice; data plotted as mean ± SD. **h** Percentage CSCs (anti-Sox9 stained) distribution from the nearest blood vessel without TMEM in fixed PyMT mammary tumor tissue. $n = 11$ fields of 2–3 mm$^2$ from 5 mice; data plotted as mean ± SD. **i** Panels from Supplementary Movie 14 showing intravasation of SORE6+ CSC (outlined in green) into blood vessels (gray, outlined by white dotted lines) in the primary tumor. Right panels (taken from the same X–Y field but adjacent Z-planes, as the TMEM doorway is above the point of intravasation of tumor cells as described previously[17]), show that intravasation occurs at TMEM. It is noteworthy that the TMEM structure is stable and is seen in the beginning of the time course till the end. It is noteworthy that TMEM tumor cell (TTC) is different from the intravasating CSC shown in left panels and does not intravasate as consistent with previous work[17].

with our observation of intravasation of CSCs at TMEM doorways, we found CSCs in the circulation (Fig. 7a). Moreover, there was a large enrichment of CSCs in the circulating tumor cell (CTC) population, where CSCs were over 60% of CTCs. In comparison, CSCs at TMEM doorways waiting to intravasate comprised only ~4–5% of the tumor cell population at that location (Fig. 7b, d). To control for the possibility that the enrichment of CSCs in the CTC population is due to selective death of non-CSCs or conversion of non-CSCs to CSCs in the circulation, we injected a mixed population of MDA-MB-231 CSCs and non-CSCs via the tail vein and then excised the lungs 6 h later for quantification of CSCs and non-CSCs by imaging. No difference was seen in the proportion of CSCs before and after their passage through the circulation (Supplementary Fig. S8b), confirming that the enrichment of CSCs in the CTC population arises at the intravasation step.

One possible explanation for the big increase in the number of CSCs in the circulation compared with around the intravasation doorways is that CSCs form invadopodia (Fig. 2) and are, therefore, much more efficient at intravasation than non-CSCs from the same primary tumor. Previous work has shown that the assembly of invadopodia and invadopodium-dependent transendothelial migration during intravasation at TMEM doorways depends on Mena^INV expression[18], which is induced by macrophage-driven Notch1 signaling in breast tumors in vivo[19]. To investigate the inter-relationship between Mena^INV+ cells and CSCs, we first investigated the preferential enrichment of Mena^INV in tumor cells associated with TMEM doorways as compared to elsewhere in the primary tumor. As shown in Fig. 7c, there is a 2.4-fold increase in Mena^INV expressing tumor cells in the proximity of TMEM doorways compared to the areas away from TMEM, indicating enhanced intravasation potential of cancer cells around TMEM. As CSCs are also enriched at TMEM doorways, we wanted to determine whether CSCs at the primary tumor also express Mena^INV. We co-stained primary tumor tissue with FLAG (to identify SORE6+ CSCs) and Mena^INV antibodies, and found Mena^INV expression in more than 55% of CSCs (Fig. 7d–e). The analysis of Mena^INV expression in non-CSCs was beyond the scope of this study. The data support enhanced intravasation activity of these double-positive (CSC + Mena^INV) tumor cells at the TMEM doorway, consistent with the enrichment of stem cells seen in circulation (Fig. 7b).

To address the relative efficiency of circulating CSCs and non-CSCs in metastasizing to distant organs such as the lung, we performed high-resolution intravital imaging of live lungs using a permanent lung imaging window to visualize the arrival of single CTCs. Using the permanent lung window and its associated micro-cartography technique to return to the same imaging field over time[44], multiple fields were imaged on two consecutive days. We saw arrival of both CSCs and non-CSCs at the lung (Fig. 7f), with the majority (>70%) of cancer cells being CSCs (Fig. 7g), similar to their enrichment numbers seen in circulation (Fig. 7b). Taken together, these results indicate that the tumor cell population becomes progressively more enriched for CSCs after passage through TMEM and during the dissemination and seeding of metastasis (Fig. 7b, g).

Next, we investigated whether the newly arriving tumor cells establish metastases in the lung, and we analyzed relative proportions of CSCs and non-CSCs in the metastases as they grew in size, using fixed-frozen lung sections to identify SORE6+ CSCs relative to all tumor cells. Lung micro-metastases composed of a few tumor cells to hundreds of tumor cells were observed (Fig. 7h). SORE6+ CSCs were quantified in small (≤10 tumor cells), medium (11–300 tumor cells), and large (>300 tumor cells) lung micro-metastases. Small lung micro-metastases were greatly enriched in stem cells (~84% of total tumor cells) (Fig. 7i), suggesting that early lesions are primarily seeded by CSCs. This conclusion is consistent with our observations that the majority (>70%) of tumor cells arriving in the lung are CSCs (Fig. 7g) and our previously published demonstration that SORE6+ cells (CSCs) initiate metastasis with ~8-fold higher efficiency than SORE6− cells (non-CSCs)[25]. As the lung metastasis size increased, the proportion of stem cells in the metastases decreased, with only 24% SORE6+ CSCs in metastases with more than 300 cells (Fig. 7h–i). An independent study, using flow cytometry rather than imaging, showed a similar progressive decrease in the relative CSC representation with time in tumor cells recovered from metastasis-bearing lungs[63]. Both datasets are consistent with the expectations of the CSC hypothesis that CSCs will generate more differentiated non-CSC progeny[64]. However, we cannot exclude the possibility that the non-CSCs have a proliferative advantage at the metastatic site, or that a microenvironmental stimulus supporting the CSC phenotype is lost in larger lesions.

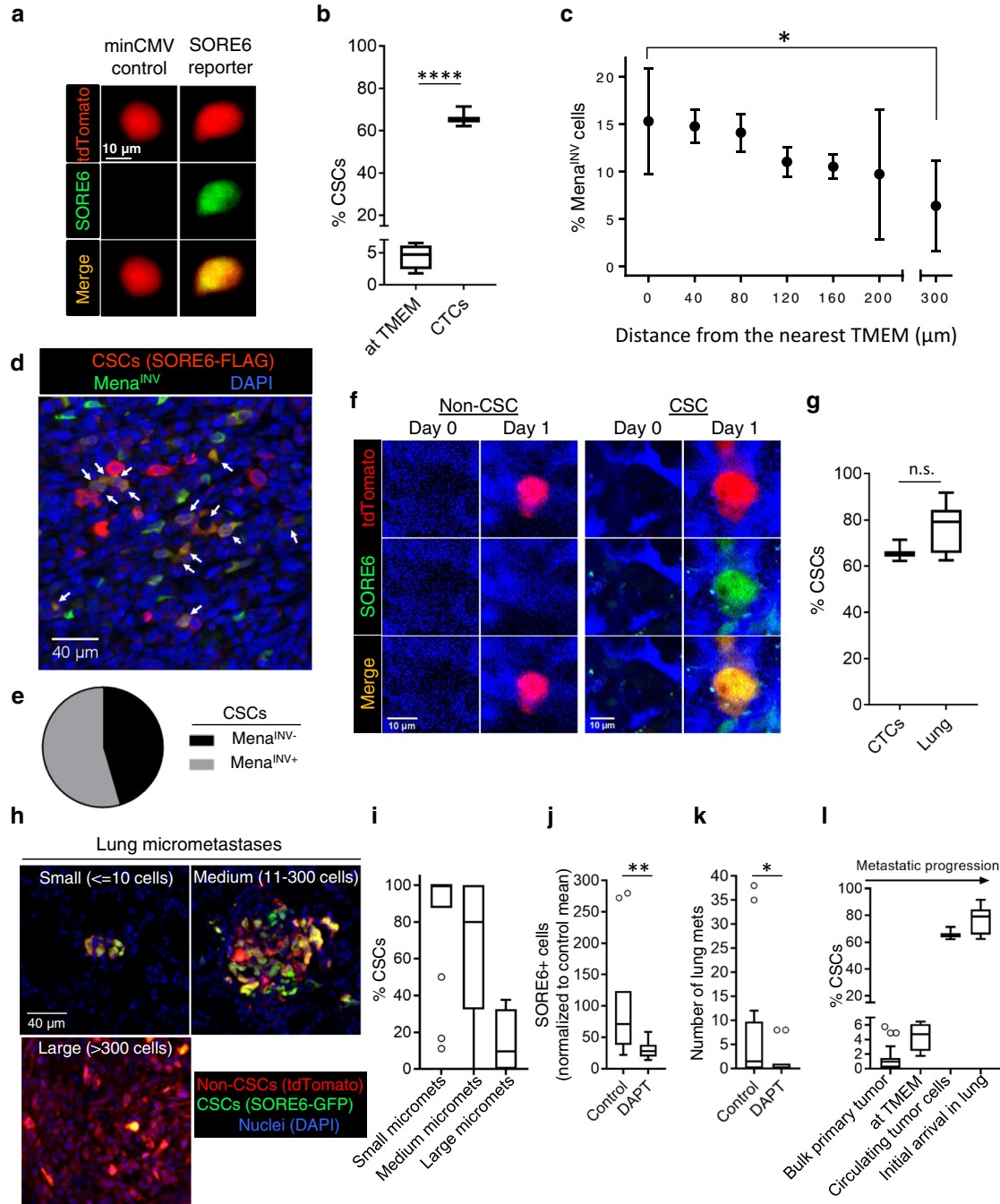

Given our earlier observations that macrophages can induce a stem phenotype through activation of the Notch pathway, we investigated whether macrophage-induced Notch signaling plays any role in the metastatic spread of CSCs. We treated SORE6>GFP MDA-MB-231-bearing SCID mice with vehicle control or DAPT and stained lungs extracted from these mice with FLAG antibody to identify SORE6+ CSCs. We found a threefold decrease in CSCs in the lung of DAPT-treated mice compared to the control-treated mice (Fig. 7j), suggesting a casual role of macrophage-induced Notch signaling in CSC metastatic potential. Importantly, inhibition of Notch signaling led to reduced metastatic burden in the lungs (Fig. 7k and Supplementary Fig. S8c).

Overall, our data show that during the course of dissemination of tumor cells from the primary site, CSCs become progressively enriched in the tumor cell population as they approach the TMEM doorway, intravasate, circulate, and arrive at the lung (Fig. 7l). Association with and passage through the TMEM doorway potentially generates the greatest enrichment in CSCs (~60-fold), which could be explained by the product of a combination of CSC enrichment at TMEM (7-fold, Fig. 5d), Mena$^{INV}$ enrichment in tumor cells at TMEM (2.4-fold, Fig. 6c) and a previously documented 3.5-fold increase in intravasation potential of tumor cells expressing Mena$^{INV}$[18,19]. These previous studies in addition to our current study could account for the enrichment of CSCs in the CTC population. On arrival in the lung, CSCs represent more than 75% of the disseminated tumor cell population, greatly enriched compared with their representation in the bulk primary tumor of ~1%. As the CSCs are similarly enriched in the smallest micrometastatic lesions in the lung, the data strongly suggest that CSCs are the main drivers of the early steps of metastatic dissemination, with the generation and

**Fig. 7 CSCs are highly enriched in circulation and at their first arrival in the lung, where they seed new micro-metastases. a** CSCs are found in CTCs collected from the mice imaged in Fig. 6i. Images of stem cell (top panels—tdTomato red, middle panels—SORE6-GFP, green) collected from blood. **b** Quantification of % CSCs in the circulation (n = 3 mice). For comparison, % CSCs (SORE6-GFP) at TMEM (Fig. 6d) is plotted. Unpaired two-tailed Student's t-test, ****p < 0.0001. **c** Percentage Mena$^{INV}$ cell distribution from the nearest TMEM in fixed MDA-MB-231 mammary tumor tissue. n = 6 fields of 1200 × 650 μm$^2$ from 3 mice having MDA-MB-231 primary tumors stained with antibodies for Mena$^{INV}$ and TMEM doorways; data plotted as mean ± SD, two-sided Mann–Whitney test, *p = 0.038. **d** In situ immunofluorescence staining of MDA-MB-231 FFPE primary tumor tissue with anti-FLAG antibody (red, to identify SORE6-FLAG CSC) and Mena$^{INV}$ antibody (green). Double-positive (stem + Mena$^{INV}$) cells are seen in yellow (white arrows). **e** Quantification of fraction of double-positive Mena$^{INV}$ + stem cells compared to CSCs that are Mena$^{INV}$ negative plotted as a pie chart. n = 7 fields of 1200 × 650 μm$^2$ from MDA-MB-231 FFPE primary tumor tissue. **f** Representative images of single non-CSC (tdTomato red) and CSC (SORE6+ green, and yellow in merged channel) cells arriving in the lung spontaneously from the primary tumor, imaged with intravital microscopy at the lung site. Note the absence of cancer cells on day0 and presence of cancer cells on day1 in the same field of view. Lung vasculature is shown in blue. **g** Percentage of CSCs among CTCs and in the newly arriving tumor cells in the lungs. Note significant enrichment in CSCs at both sites compared to TMEM doorways (**b**); n = 18 stem cells, six non-stem cells in five mice, unpaired two-tailed Student's t-test, n.s. p = 0.11. **h** Representative images of fixed-frozen lung tissue show metastases at different stages of progression. Small, medium, and large micro-metastases are defined as micro-metastases containing ≤ 10 cells, 11–300 cells, and >300 cells, respectively. **i** Quantification of % CSCs in lung metastases at different stages of progression as shown in Fig. 6h. Note a progressive loss of CSCs as metastases enlarge. n = 70 micro-metastases analyzed in three mice. **j** Quantification of SORE6+ cells in the lungs of control or DAPT-treated SCID mice bearing SORE6>GFP MDA-MB-231 tumors. n = 5 and 6 mice for control and DAPT conditions, respectively; two 1.8 × 1.8 mm$^2$ lung areas were analyzed in each mouse; two-sided Mann–Whitney test, **p = 0.002. **k** Quantification of histologically detectable lung metastases in control and DAPT-treated SCID mice bearing SORE6>GFP MDA-MB-231 tumors. n = 7 and 6 mice for control and DAPT conditions, respectively; two sections 50 μm apart were analyzed in each mouse; two-sided Mann–Whitney test, *p = 0.0198. **l** The % CSCs for the bulk primary tumor (Fig. 1d, FFPE), at TMEM (Fig. 6d), in circulation (**b**) and initial arrival in the lung (**g**) are plotted and show progressive enrichment in CSCs up to the point of initial metastatic seeding in the lung. **b**, **g**, **i–l** The boxes indicate 25th–75th percentile interquartile range (IQR) and central line indicates the median. Top and bottom whiskers were plotted using Tukey's method and extend to 75th percentile + 1.5 × IQR and 25th percentile − 1.5 × IQR, respectively. Points below and above the whiskers are drawn as individual dots.

expansion of non-CSC progeny driving the later steps of metastatic colonization and growth.

**The density of TMEM doorways correlates with the proportion of CSCs in breast cancer excisions from patients**. As we found that, in the pre-clinical models of breast cancer, areas around TMEM are enriched for CSCs, and that macrophages can induce a stem phenotype, we hypothesized that human breast cancers with high TMEM doorway density would also have a high proportion of CSCs. If high TMEM doorway density is accompanied by a high density of CSCs, this would provide an additional mechanistic insight into the prognostic power of TMEM doorway score for metastatic outcome in patients[59–61] beyond the function of TMEM doorways as a cancer cell intravasation doorway. To investigate the relationship between TMEM doorway density and the proportion of CSCs in breast cancers in patients, we used 49 breast cancer excisions from patients who had various breast cancer subtypes (Fig. 8a) and a special tissue collection approach described in Fig. 8b. Briefly, we collected cancer cells by fine needle aspiration (FNA) from breast cancer excisions as described under "Methods." This approach has several benefits. It allowed us to sample large portions of tumors since the aspiration was performed from at least three distinct tumor areas with five to ten needle passes for each tumor area. Furthermore, the FNA sample contained mostly single cells that facilitated the flow cytometry analysis of the expression of CD44+/24− on the cell surface, a commonly used marker combination associated with stemness, which is difficult to analyze in fixed tissues[2]. Lastly, this approach was not associated with appreciable tissue destruction as assessed by histopathology, which allowed us to use the same tumors upon formalin fixation for the analysis of TMEM doorway score. Thus, upon completion of FNA, the very same breast cancer excisions were FFPE and the representative tissue sections were used for TMEM doorway density analysis. We then correlated the density of TMEM doorways with the proportion of cancer cells expressing stem cell markers standardly used for detection of CSCs in breast cancers from patients. We wanted to take into consideration the plasticity that BCSCs exhibit as they transit between

proliferative, epithelial-like (E), and a quiescent, invasive, mesenchymal-like state. To accomplish that, we used CD44+/CD24−, which marks BCSC in the mesenchymal-like state and aldehyde dehydrogenase 1 (ALDH1), which marks BCSC in the epithelial-like state[30]. As an additional marker of stem phenotype, we used CD133, because it is the most frequently used maker of stemness in solid cancers from patients[31,32] (Fig. 8d). We used a variety of methods for detecting stem cell markers to evaluate the results in a technique independent way (Fig. 8c–e). With all methods, we observed an extraordinarily strong positive correlation between TMEM doorway density and the proportion of cancer cells expressing stem cell markers (CD44+/24−, CD133, and ALDH1) (Fig. 8c, d).

In addition to the percentage of cells expressing ALDH1 and CD133 as determined by in situ hybridization, we also wanted to evaluate the total amount of ALDH1 and CD133. Therefore, we determined the amount of mRNA transcript for ALDH1 and CD133 by quantitative reverse-transcription PCR (qRT-PCR). We then correlated the amount of ALDH1 and CD133 mRNA measured by qRT-PCR (normalized to glyceraldehyde 3-phosphate dehydrogenase (GAPDH)) with TMEM doorway density and again obtained significant correlation (Fig. 8e).

As the SORE6 biosensor cannot be expressed in human samples, to identify CSCs in situ, we used Sox9 as a surrogate stemness marker in fixed tissue. We stained FFPE human tissues with Sox9 antibody and correlated Sox9 expression (mean fluorescence intensity) in each tissue with its corresponding TMEM doorway density, and found significant correlation between Sox9 expression and TMEM doorway density in human tissues (Fig. 8f). To check spatial distribution of stem cells with respect to TMEM doorways in human tumors, we stained sequential tissue sections for TMEM doorways using a humanized version of triple-IHC (pan-Mena, CD68, and CD31 antibodies) and for stem cells using Sox9 antibody. After aligning TMEM doorway and Sox9 sections, we found significant enrichment of Sox9$^{Hi}$ cells close to TMEM doorway-rich areas (Fig. 8g). Quantification showed threefold enrichment of Sox9$^{Hi}$ cells in TMEM doorway-rich areas compared to that in TMEM doorway-deficient areas in human tissues (Fig. 8h).

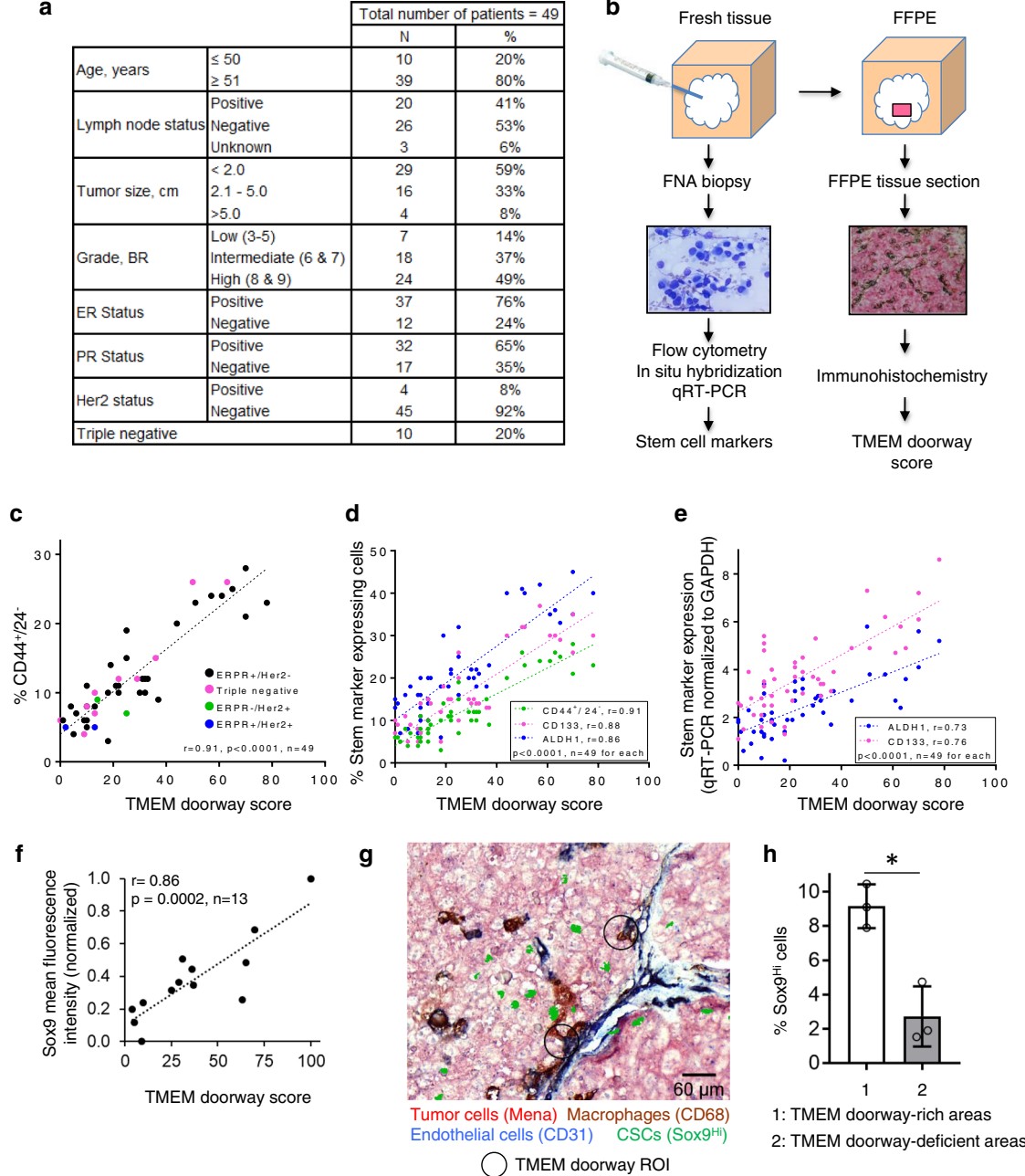

**Fig. 8 Correlation of TMEM density with the proportion of tumor cells expressing stem cell markers in breast cancer tissues from patients. a** Clinical and pathological characteristic of the 49 breast cancer cases included in the analysis. **b** A schematic of tissue collection procedure for the correlation of TMEM doorway score with the proportion of cancer cells expressing stem cell markers. **c** TMEM correlation with stem cell marker CD44+/24− in 49 invasive ductal carcinomas. Cells were collected by FNA and analyzed by flow cytometry for % cells expressing CD44+/CD24−. TMEM score from the same tumor tissue was correlated with the % CD44+/24− cells by Pearson's correlation. **d** Correlation of TMEM with the percentage of breast cancer cells expressing stem cell markers CD44+/24−, ALDH1, and CD133 in 49 invasive ductal carcinomas from patients. Cells were collected by FNA and analyzed for CD133 and ALDH1 by in situ hybridization. CD44+/24− data are plotted from **c**. TMEM score from the same tumor tissue was correlated with the percentage of cancer cells expressing stem cell markers using Pearson's correlation. **e** Correlation of TMEM with the percentage of breast cancer cells expressing stem cell markers ALDH1 and CD133 in 48 invasive ductal carcinomas from patients. Cells were collected by FNA and analyzed for CD133 and ALDH1 by qRT-PCR. TMEM score from the same tumor tissue was correlated with the percentage of cancer cells expressing stem cell markers using Pearson's correlation. **f** Sox9 mean fluorescence intensity in FFPE of the above human tissues ($n = 13$) were correlated with the TMEM score for each human tissue. Pearson's correlation $r = 0.86$, $p = 0.0002$, data normalized to the min and max Sox9 mean fluorescence intensity values. **g** A composite image of TMEM staining (pink: Mena+ tumor cells, brown: CD68+ macrophages, blue: CD31+ endothelial cells) and Sox9Hi stem cells (green), showing enrichment of Sox9Hi cells close to TMEM areas. Two TMEM doorway ROIs are highlighted with black circles. **h** Quantification of % Sox9Hi cells in TMEM-rich and TMEM-deficient areas. $n = 3$ human tissues, data plotted as mean ± SD, paired two-tailed Student's $t$-test, *$p = 0.0336$.

Overall, these results indicate that TMEM doorways in patients may not only be sites for cancer cell intravasation but also microenvironments enriched for cancer cells with stem properties, in agreement with our animal studies.

## Discussion

A major limitation in studying CSC biology has been a heavy reliance on assays that disrupt tumor microenvironments, thus preventing characterization of CSCs in situ in their specialized niches. Here we have used a validated CSC fluorescent reporter system in combination with intravital high-resolution multiphoton microscopy to gain important new insights into the induction, dynamic behaviors, and fates of CSCs in vivo in breast cancer primary tumors, and during the process of seeding of lung metastases. In addition, we identified specific CSC-enriched niches in the tumor microenvironment and verified that the same niches exist in human breast cancers obtained from patients. These insights may have impact on patient care.

As expected[3], we found that CSCs comprise a minority (~1%) of cancer cells in the primary tumor. In vivo time-lapse imaging revealed that these CSCs demonstrate a slow-migratory, invasive invadopodium-rich phenotype, which is the hallmark of disseminating tumor cells[36]. In contrast, the non-CSCs show a fast-migrating phenotype. Our previous work showed that migratory tumor cells use aligned collagen fibers for high-speed migration toward blood vessels[7,34,65–67]. We have observed that fast migratory non-CSCs slow down as they approach the blood vessel probably due to lower collagen fiber alignment and density near blood vessels. Thus, tumor cell speed is expected to switch from fast to slow as cells approach blood vessels[36]. Previous work has also shown that the slow-migratory invadopodium-rich, invasive phenotype is also present in the transendothelial migration competent cancer cells expressing high levels of Mena[INV][18,19,36]. Given the disseminating phenotype of CSCs and Mena[INV]-expressing cells, we evaluated their distributions within the tumor microenvironment in relationship to TMEM doorways, which mediate tumor cell intravasation and dissemination[7,42,68], and are prognostic for distant metastatic recurrence[59–61,68]. Interestingly, both CSCs and Mena[INV]-expressing cells are preferentially located close to the TMEM doorways. Furthermore, association with and passage through the TMEM doorways contributed to the 60-fold enrichment of CSCs among CTCs and 70-fold upon arrival to the lungs, compared with their relative representation in the bulk primary tumor. Thus, our data suggest that TMEM doorways promote the intravasation of invasive CSCs from the primary tumor. Our data further indicated that these CSCs initiate metastasis at the distant site, thereby intimately linking stemness and dissemination. This concept is consistent with our observations that early metastatic lesions in the lung have a very high CSC representation (~80%), a finding that has also been made by others using orthogonal techniques[69]. As only 50% of SORE6+ cells showed Mena[INV] expression in the primary tumor, future work is necessary to identify the precise role of Mena[INV] expression in SORE6+ intravasation. A detailed study looking at Mena[INV] expression in SORE6+ vs. SORE6− cells in the primary tumor, CTCs, and lung is required.

Recently, it was shown by live imaging that small clusters of tumor cells can be seen at intravasation sites and in the circulation, and that they result from the aggregation of single CSCs, instead of from collective migration and vascular invasion of non-stem tumor cells[70]. Further, the aggregation seemed to be mediated by intercellular homophilic interactions of CD44, a classic CSC marker that is strongly associated with TMEM density in breast cancer patients. These findings are consistent with the enhanced metastatic seeding activity of the CSCs reported here and may explain the reports of enhanced metastatic seeding activity of tumor cell clusters[71] if clusters are primarily composed of CSCs[72].

As self-renewal, differentiation potential, and phenotypic plasticity of both normal and CSCs are regulated by input from the local microenvironment[8], we used the stem cell reporter to directly observe the effect of the tumor microenvironment on cancer cell plasticity in vivo in real time. We found that CSCs compared to non-CSCs are more frequently present close to or in direct contact (60–70%) with intra-tumoral macrophages. Most importantly, we observed a novel contact-dependent de novo induction of the CSC phenotype in non-CSC tumor cells when they directly touched macrophages, both in vitro and in vivo. Even though we found no effect of clodronate on the number of CSCs or general cancer cell viability and proliferation in vitro, further work is required to evaluate any indirect effect of clodronate-mediated macrophage depletion on SORE6+ cells in vivo. Also, we have no evidence from the short-term (30 h) fate mapping experiments in vitro that the SORE6+ cells die following loss of macrophage contact, suggesting that clodronate-mediated CSC reduction seen in vivo results primarily from blockade of de novo CSC generation by macrophages, but it is conceivable that additional mechanisms (such as death of pre-existing CSCs) may also contribute in vivo. The data suggest a model in which macrophages generate a cell contact-dependent inductive signal that can promote phenotypic plasticity and re-acquisition of stem properties in more differentiated tumor cells.

Interaction between macrophages and post-EMT stem-like cells, characterized by high expression of CD90, has previously been studied in the HMLER model of human breast cancer and in human breast cancer specimens[14]. In that study, it was proposed that tumor-associated macrophages constitute a supportive niche by enabling pre-existing CSCs to maintain their residence in the stem cell state, through Ephrin-dependent expression of the cytokines IL6 and IL8[14]. In addition, a juxtacrine signaling interaction between macrophages and tumor cells has implicated the LSECtin–BTN3A3 axis in CSC promotion leading to enhanced tumor growth[73].

In a significant advance, our data show that intra-tumoral macrophages may not merely support survival and expansion of pre-existing CSCs leading to increased tumor growth, but can also actively induce stemness in non-stem cancer cells, resulting in a population of CSCs that is linked to systemic dissemination. This induction of stemness operates via a molecular pathway that is distinct from the Ephrin-dependent maintenance of stemness and the LSECtin–BTN3A3 axis supporting tumor growth, and involves macrophage-tumor cell contact-dependent Notch signaling. Notch is one of the core signaling pathways that has been implicated in regulation of both normal and CSCs in many organ systems[74] and in tumor cell intravasation activity[19]. High expression of Notch1 and Jag1 is associated with poor overall survival (OS) in breast cancer, suggesting the importance of this signaling axis in human breast cancer[75]. Our results connect the de novo induction of cancer cell stemness by contact with macrophages to the dramatic enrichment of CSCs that we see in association with and passage of tumor cells through the TMEM doorway. As CSCs accumulate around TMEM doorways, a macrophage-enriched environment that can also contribute to CSC induction[17,67], these newly induced CSCs are likely to disseminate more efficiently than CSCs positioned away from TMEM doorways.

The observation that macrophage-driven induction of stemness in tumor cells is associated with intravasation and dissemination through TMEM doorways is arguably the worst possible scenario from a clinical perspective, likely leading to poor patient survival. This linkage is further supported by earlier

findings that macrophages induce invadopodium assembly and maturation in breast tumor cells in vitro and in vivo via a Notch signaling pathway by inducing Mena[INV] expression[19,42,76]. These results indicate that Notch signaling is used in common to induce both stemness and the Mena[INV]-dependent invasive invadopodium-rich tumor cell phenotype. Mena[INV]-expressing tumor cells have greatly enhanced chemotaxis toward both macrophages and blood vessels[7,67,77], and Mena[INV] expression induces invadopodium assembly[76], generating an enhanced transendothelial migration phenotype that is essential for TMEM-mediated dissemination[18]. If indeed both the CSC state and Mena[INV] expression are induced at TMEM, this would explain the dramatic enrichment of CSC levels among CTCs. This novel finding is consistent with a previous literature where the presence of CSCs was detected in CTCs in mouse models and breast cancer patients[72,78].

Previous studies showed that depletion of macrophages, knockout of the vascular endothelial growth factor gene in macrophages[17,62], or inhibition of TMEM by genetic and pharmacologic strategies[15,58], all lead to depletion of CTCs and inhibition of metastasis, indicating that cancer cell intravasation occurs only at TMEM[15,17]. This is in accordance with the data from breast cancer patients, which indicate that TMEM density, independently of blood vessel density, is associated with tumor cell dissemination and metastasis[59–61]. The enrichment of CSCs at TMEM doorways in human tumors demonstrated here supports prior work and the conclusion that TMEM doorways may be involved in the dissemination of CSCs in humans.

Given the high percentage of CSCs among CTCs, it is not surprising that the presence of CTCs in breast cancer patients correlates with worse disease-free survival (DFS) and OS. For example, a multi-institutional analysis of individual CTC data from more than 3000 patients with non-metastatic breast cancer showed that the presence of CTCs is an independent predictor of poor DFS, distant DFS, breast cancer-specific survival, and OS[79]. Likewise, the detection of CTCs before the onset of pre-operative chemotherapy is an independent prognostic indicator of worse DFS and OS[80]. In fact, meta-analysis of data from individual patients treated with pre-operative chemotherapy collected from 21 studies showed that the hazard ratio of death proportionally increases with the number of CTCs detected before the onset of chemotherapy[81]. In addition to the mere presence of CTCs, the gene expression pattern of CTCs also affects metastatic formation. In accordance to our data showing that SORE6+ cells (CSCs) are the source of early metastatic foci, the expression of gene combinations involving stem programs such as Notch and CD44 confers higher metastatic potential[82,83]. Likewise, the expression of CSC markers in CTC in patients with several epithelial cancer types correlates with occurrence of metastasis and decreased patient survival[84,85]. Interestingly, in patients with metastatic breast cancer the occurrence of CTC expressing not only stem markers, but also markers consistent with partial EMT was shown to be an independent factor for prediction of increased relapse[86] supporting the notion that both stem and EMT/plasticity programs are involved in metastasis in patients. Our data showing that SORE6+, compared to SORE6− cells express significantly more EMT transcription factor Snail1 are in accordance with these clinical findings, which link patient outcome with activation of both stem and EMT programs in CTCs.

We found a striking correlation between the density of TMEM doorways and the proportion of cancer cells expressing CSC markers in breast cancer samples from patients. These data indicate that TMEM doorways are not only portals for cancer cell dissemination to distant sites but may also represent microenvironmental niches for the induction of the stem cell program in human breast cancer cells. These data further

explain the prognostic power of TMEM for metastatic disease[59–61], given the association of stem and EMT program with metastatic dissemination[84,85,87]. Indeed, TMEM density correlates with the proportion of CSCs expressing markers associated with both proliferative, epithelial-like (ALDH1), and quiescent mesenchymal-like (CD44+/24−) states[30]. This transition of CSCs between the epithelial- and mesenchymal-like states closely resembles the EMT program, which in itself is associated with the acquisition of stem cell properties[87]. Thus, the biology of TMEM is strongly linked with stem and EMT programs, as well as with cancer dissemination. These data further support the use of TMEM score as a prognosticator of distant recurrence of breast cancer[59,61,68].

Beyond contributing to increased understanding of the fundamental biological mechanisms underlying metastatic dissemination, our findings have potential implications for the prognosis of distant recurrence in breast cancer patients with localized disease, as well as for the prognosis of disease progression and treatment of breast cancer patients with metastases. In terms of prognostication, the association of CSCs with TMEM doorways, resulting in increased CSC-mediated seeding of metastases, provides additional mechanistic insight into the prognostic power of TMEM score. As TMEM is currently the only clinically validated marker of cancer cell dissemination, it may provide complementary information to current prognostic markers that measure proliferative potential such as OncotypeDX recurrence score (RS). For example, high TMEM score in patients with low OncotypeDX RS could be used to tailor treatment as described in Sparano et al.[61]. Further, as TMEM score correlates strongly with the percentage of CSCs in tumors as we show here, TMEM score may be used as an indicator of chemoresistance. It is known that CSCs are intrinsically resistant to many therapies[5] and CSC-dependent chemoresistance is believed to be a major cause of metastatic relapse[88]. Our observations that tumor-associated macrophages induce the CSC program indicate that chemoresistance may be further increased in situations that promote macrophage influx into tumors. We and others have previously shown that chemotherapy induces macrophage influx[15,89,90] and increases the density and the activity of TMEM doorways[15,91,92]. Consequently, increased macrophage density induces the expression of Mena[INV] in tumor cells[15], which enhances the transendothelial migration of cancer cells via promoting the assembly and activity of invadopodia[15,76]. Thus, chemotherapy may be impacting two macrophage-associated steps in the dissemination of tumor cells: their stemness and their invasiveness.

Similarly, chemotherapy has previously been associated with enrichment of CSCs in human breast cancer patients[93]. To treat metastatic relapse, the inter-related phenomena of chemotherapy-mediated induction of CSCs and dissemination need to be addressed. Recent studies have identified drugs that block the recruitment of macrophages to tumors in response to chemotherapy and block TMEM function[58]. As TMEM doorways are found in metastatic foci in the lymph nodes[94] and in the lungs[44], the same mechanism of dissemination may perpetuate metastases even after the removal of the primary tumor. Thus, it would not be too late to inhibit TMEM function after the removal of the primary tumor, because in some patients, cancer cells might have already disseminated from the primary site and formed clinically undetectable micro-metastases, which can be a source of further metastatic cancer cell dissemination via TMEM doorways. Interestingly, the enumeration of CTCs is prognostic even in patients with metastatic breast cancer[95,96]. Our current findings suggest that drugs targeting macrophages might have the important additional benefit of disrupting the induction and dissemination of CSCs[97]. Thus, many vexing problems associated

with chemoresistance and metastasis may be addressed through this one cellular target.

## Methods

**Cell culture.** The MDA-MB-231 human breast cancer cell line was obtained from ATCC and the identity of the line was re-confirmed by Short Tandem Repeat (STR) profiling (Laragen Corp.), after expansion and passaging. The human breast cancer stem reporter cell line, tdtomato MDA-MB-231 SORE6>GFP, and the vector control cell line, tdtomato MDA-MB-231 minCMV>GFP, were maintained in 10% fetal bovine serum (FBS) in Dulbecco's modified Eagle medium (DMEM). The MDA-MB-231-LM2 subline[98] was obtained from Dr. Joan Massague, Sloan Kettering Institute; the Met-1 cell line derived from an MMTV-PyVT mouse mammary tumor[46] was obtained from Dr. Alexander Borowsky, University of California, Davis; the 4T1 metastatic mammary cancer cell line was obtained from Dr. Fred Miller, Karmanos Institute, Detroit. All were maintained in DMEM, 10% FBS. The BAC1.2F5 macrophage cell line was maintained in 10% FBS in α-MEM with 3000 unit/ml CSF-1. Cryopreserved human primary monocyte-derived macrophages that had been isolated from peripheral blood monocytes and polarized to an M2 phenotype by 4–5 days culture with 10% FBS + 50 ng/ml CSF-1 + 10 ng/ml IL4 were obtained from StemExpress, Folsom, CA (Cat# PBMAC001.5C), and were thawed and used upon receipt. HUVECs (Lonza, Walkersville, MD, USA Lot # 0000396930) were grown in EGM-2 SingleQuot Kit media (Lonza) and used at passages 4–6. All cells were maintained at 37 °C in a 5% CO$_2$ incubator and were shown to be mycoplasma-free (Sigma LookOut Mycoplasma PCR detection kit, Cat# MO0035-1KT).

**Design of CSC reporter.** The original SORE6+ CSC biosensor consisting of six repeats of a composite SOX2/OCT4 binding element coupled to a minimal CMV promoter driving the expression of a dsCopGFP has been described previously[25]. A second-generation version of this reporter was generated (readily available from the authors upon request) for these studies. As the dsCopGFP fluorescence is lost on formalin fixation, the new biosensor has an N-terminal 3× FLAG epitope tag on the dsCopGFP to enable immunohistochemical detection in FFPE tissue and the puromycin drug selection cassette has been replaced with an expression cassette for a C-terminally truncated CD19 marker to enable rapid FACS selection of transduced cells. New constructs were made by Gateway multisite recombinational cloning. The SORE6 enhancer, minimal CMV promoter, tagged destabilized fluorescent protein and SV40-driven truncated CD19 selection marker elements were separately cloned and sequence-verified before Gateway assembly into pDest-412, a lentiviral destination vector based on the pCDF lentiviral backbone to generate the stem cell biosensor (SORE6>GFP). A parallel construct ("minCMV") lacking the SORE6 enhancer element was generated as a nonspecific background control for FACS gating and fluorescent image thresholding in every experiment. The dsCopGFP protein has a half-life of 1–2 h, which is similar to the 1.5 h half-life reported for the OCT4 protein in P19 mouse teratocarcinoma cells[99]. As a small reduction in OCT4 protein levels in sufficient to alter the balance between self-renewal and differentiation[100], the SORE6 reporter kinetics are tuned to parallel closely the dynamics of the stem cell phenotype. The second-generation SORE sensor behaves identically to the first-generation sensor that was extensively validated by us in multiple breast cancer models[25] and used by others[101–107].

**siRNA knockdown of stem cell transcription factors.** siGENOME SMARTpool siRNAs were purchased from Dharmacon as follows: OCT4/POU5F1 (Cat# M-019591-03-0005); SOX2 (Cat# M-011778-00-0005); SOX9 (M-021507-00-0005); Non-Targeting siRNA Control Pool #1 (Cat# D-001206-13-05). MDA-MB231 cells transduced with the SORE6>GFP reporter were transfected with 20 nM siRNAs using Lipofectamine RNAiMAX Reagent (Invitrogen #13778075) in OptiMEM$^{TM}$ Reduced Serum Medium (Thermo Fisher Scientific #31985062). The % SORE6+ cells in the transfected cultures was assessed after 3 days by flow cytometry, using cells transduced with the minCMV>GFP as a gating control.

**Tumorsphere assays.** MDA-MB-231 cells transduced with the SORE6>GFP stem cell reporter were FACS-sorted into SORE6+ and SORE6− populations using cells transduced with minCMV>GFP construct as a gating control. Five thousand cells/well were seeded into Costar 24-well Ultralow attachment plates (Corning Cat#3473) in DMEM/10% FBS. After 10 days, tumorspheres were quantified from images captured using an EVOS FL Auto2 Imaging system using ImageJ, to size the tumorspheres for binning by diameter. To address the functionality of SORE6+ cells generated by co-culture with macrophages, 20,000 MDA-MB-231 cells were cultured with or without 40,000 BAC cells in 24-well plates for 24 h in BAC culture medium. After 24 h, tumor cells were collected by brief trypsinization (30 s), which selectively removes tumor cells, whereas the BAC cells remain adherent. The collected tumor cells were seeded at 1500 cells/well in Ultralow attachment plates in DMEM/10% FBS and assessed for tumorsphere formation after 10 days as above.

**Taxol treatment.** Unsorted MDA-MB-231 cells containing the SORE6>GFP reporter were seeded into 24-well plates and grown for 24 h. Cells were then treated

with dimethyl sulfoxide (DMSO) vehicle or the indicated concentrations of Paclitaxel (time $t = 0$ h) and imaged every 6 h using the Incucyte S5 Live-Cell imaging system. SORE6-positive cells were quantifed as GFP+ cells and SORE6-negative cells were quantitated as GFP− cells within the same culture, using a parallel culture with the minCMV>GFP construct to set the threshold for positivity.

**qRT-PCR for stem cell transcription factors in MDA-MB-231 and Met-1 cells.** RNA was prepared from freshly sorted cells using the Trizol Reagent (Ambion) and cDNA was synthesized using SuperScriptTM III First-Strand Synthesis System (Thermo Fisher Scientific) as per the manufacturer's instructions. qRT-PCR was performed with Brilliant II Ultra-Fast SYBR® Green QPCR Master Mix (Agilent) using a Bio-Rad CFX96 Real-Time Detection System. Fold gene expression was calculated using the delta CT method formula: $2^{(-\Delta Ct)}$. Peptidylprolyl Isomerase A was used as the reference transcript for normalization. Sequences for all the primers used are listed in Supplementary Table 1.

**Animal models.** All procedures were conducted in accordance with the National Institutes of Health regulation concerning the care and use of experimental animals and with the approval of Albert Einstein College of Medicine Animal Care and Use Committee. The stem reporter cells tdTomato MDA-MB-231 SORE6>GFP or the reporter control cells tdTomato MDA-MB-231 minCMV>GFP were injected into the mammary fat pad of SCID mice (NCI) as previously described[35]. Enhanced cyan fluorescent protein (ECFP) macrophage/Rag2-knockout mice (Rag2KO Macblue) were generated from the crossing B6.129S6-Rag2$^{tm1Fwa}$ N12(Rag2—Model RAGN12, Taconic) with Tg(Csf1r*-GAL4/VP16,UAS-ECFP)$^{1Hume/J}$ (Stock No: 026051, the Jackson Laboratory). Tumor cells were orthotopically injected with 50% Matrigel (BD Biosciences, cat # 354234) into the fourth mammary fat pad of Rag2KO MacBlue mice. For drug treatment (Clodronate, DAPT) experiments, animals were randomly assigned to the drug-treated or vehicle-treated groups. All subsequent analyses were performed on randomized samples in a user-blinded manner.

**In vivo limiting dilution assay for stem cell reporter validation.** To validate the newly modified version of the reporter, MDA-MB-231 cells expressing the tdTomato volume marker were transduced with the new SORE6>dsCopGFP construct. In parallel, cells were transduced with the matched minCMV>dsCopGFP as a gating control. Transduced cells were first selected by FACS sorting for the CD19 selectable marker, to ensure the presence of the reporter in all cells. CD19+ cell populations that were either SORE6+ (CSCs) or SORE6− (non-CSCs) were generated by FACS sorting for GFP. SORE6+ and SORE6− populations were then orthotopically implanted into the #2 and #7 mammary fat pads of 6-week-old female SCID/NCr mice in 0.1 ml 50% Matrigel. Tumor formation was monitored by regular palpation and caliper measurements, and confirmed at necropsy. CSC frequency was calculated from tumor incidence at day 98 post inoculation, using the ELDA software tool (http://bioinf.wehi.edu.au/software/elda/). It should be noted that this assay is useful for assessing the relative tumor-initiating ability (and hence CSC representation) of two cell populations, but that the absolute CSC frequencies calculated by this approach strongly under-represent the true absolute frequencies. This is because the efficiency of experimental tumor initiation is strongly affected by multiple parameters, such as degree of immunodeficiency, mouse strain background, site of tumor cell implantation, and others, as documented in the literature[108].

**FACS and flow cytometry.** For cultured cells, cells were trypsinized, pelleted, and resuspended in 2% bovine serum albumin (BSA)/PBS as a single cell suspension for FACS sorting. Tissues from tdTomato MDA-MB-231 SORE6>GFP or tdTomato MDA-MB-231 minCMV>GFP primary tumors were washed once with PBS, diced, suspended in PBS, passed through a 70 μm cell strainer, and centrifuged down to cell pellets. Pellets were resuspended in PBS, filtered with a 40 μm cell strainer, spun down, resuspended in 200 μL of 2% BSA/PBS, and stained with Antigen presenting cell (APC)-lineage antibody cocktail (BD Pharmingen, APC mouse lineage antibody cocktail, Cat# 51-9003632) at 5 μL antibody/million cells for 30 min at room temperature. After staining, the cells were washed twice with PBS and resuspended in 2% FBS/PBS for FACS sorting. Comparing with isotype control, APC-negative cells are the tumor cells for analysis. When sorting the SORE6+ (GFP+) and SORE6− (GFP−) cells from tdTomato MDA-MB-231 SORE6>GFP tissue, tdTomato MDA-MB-231 minCMV>GFP was always used as the GFP gating control. Cell surface marker expression in the SORE6+ and SORE6− compartments of MDA-MB-231 cells cultured in vitro or recovered from primary tumors was assessed by flow cytometry following staining with Brilliant Violet 421$^{TM}$ anti-human CD133 (Biolegend Cat#372807), used at 5 μl antibody per million cells. Flow cytometry of stained cells was performed on a BD LSRFortessa SORP1 (BD Biosciences) using collection software FACS Diva version 8 and data were analyzed using FlowJo v10.6.2 software. MDA-MB-231 cells transduced with the minCMV>GFP control construct and stained with isotype control (Biolegend cat #400157) were used to set the thresholds. To assess the relationship between SORE6 positivity and ALDH positivity, MDA-MB-231 cells transduced with SORE6>GFP but lacking the tdTomato volume marker were incubated with

AldeRedTM 558A (AldeRedTM ALDH Detection Assay, EMD Millipore Cat# SCR150) for 1 h at 37 °C. The AldeRedTM 588-A oxidation product was detected in the red channel (610 nm) by flow cytometry. Treatment of MDA-MB-231 cells transduced with control minCMV>GFP construct with AldeRedTM 588-A plus the ALDH inhibitor DEAB established the thresholds for AldeRed and SORE6>GFP positivity.

**Fixed-frozen tissue immunofluorescence staining and imaging.** Primary tumors and the inflated lungs from tdTomato MDA-MB-231 SORE6>GFP or tdTomato MDA-MB-231 minCMV>GFP xenograft mice were collected. The tissues were fixed in 5% neutral buffered formalin (NBF) and 20% sucrose in 4 °C for 2 days before being embedded and frozen in Optimal cutting temperature (OCT) compound. The fixed-frozen sections were brought in room temperature for 30 min, treated with −20 °C cold acetone for 10 min, washed with PBS, outlined with PAP pen, stained with DAPI (4′,6-diamidino-2-phenylindole) for 10 min, washed with PBS, and mounted on a coverslip for imaging. In the case of immunofluorescent staining, before the DAPI staining was performed, each section was blocked with 1% BSA, 10% FBS in PBS for 1 h at room temperature, incubated with anti-Iba1 antibody at 1 : 200 (Wako) in blocking buffer for 1 h, washed with PBS three times, incubated with secondary antibody (1 : 200) for 1 h, and washed two times as validated previously[58]. The dried stained slides were imaged with the DeltaVision microscope. tdTomato MDA-231 minCMV>GFP tissue slides were used for thresholding the images. Images were analyzed in ImageJ.

**In vivo time-lapse intravital imaging and analysis.** A mammary imaging window was implanted into mice with an MDA-MB-231 primary tumor expressing tdTomato + SORE6>GFP or tdTomato + minCMV>GFP, and intravital multiphoton imaging was performed on a custom-built two-laser multiphoton microscope as described earlier[109,110]. Mice were anesthetized with 1–2% isofluorane and kept at physiological temperatures on the microscope stage with the built-in environmental heat enclosure and vitals were monitored with pulse oximeter (PhysioSuite, Kent Scientific). All images were captured using a ×25, 1.05 NA water-immersion objective.

For SORE6>GFP stem biosensor imaging, GFP background was set using tumors expressing the minCMV>GFP control construct (Supplementary Fig. S3). All subsequent SORE6>GFP stem biosensor imaging was performed at this laser power and GFP gain setting. Far-red Q dots (Thermo Fisher Scientific, cat# Q21061MP) were used to visualize the blood vasculature. Long time-lapse z-stack imaging was performed by imaging multiple fields (512 × 512 μm² or 341 × 341 μm²) with a 5 μm z-step size.

Motility analysis (Fig. 2b) in the intravital movies was performed as described before[35]. The majority of tumor cells in the field were stationary and not included in the analysis. Only motile tumor cells, consistent with ref. [35], Supplementary Fig S2, were analyzed for speed calculations. Individual 16-bit TIFF images were assembled into a CZT hyperstack using a custom ImageJ macro. Temporal drift in multiple-channel time-lapse hyperstacks was corrected using HyperStackReg plugin[111]. For single cells SORE6− and SORE6+ cell motility analysis, cells were manually tracked using TrackMate plugin[112]. In addition, we measured tissue drift in our intravital movies and found that tissue drift after post-acquisition drift correction is minimal (~0.02 μm/min, Supplementary Fig. S4e and Supplementary Movie 9) and does not contribute to the cell migration on the time scale of the movies.

**Macrophage depletion using Clodronate.** A chronic mammary imaging window was implanted into Rag2KO Macblue mice with an MDA-MB-231 primary tumor expressing tdTomato + SORE6>GFP, and intravital multiphoton imaging of multiple 512 × 512 μm² fields were performed on three consecutive days (day0, day1, and day2) using an in vivo micro-cartography technique[44] on a custom-built two-laser multiphoton microscope as described above. Mice were injected intraperitoneally with 200 μl of either control PBS liposomes or Clodronate liposomes (Encapsula Nano Sciences, SKU# CLD-8901) on day0 and day1 after intravital imaging. The 3D objects counter plugin in Fiji was used to quantify macrophage voxels in the 3D hyperstacks. Macrophage density was calculated as the voxels covered by macrophages divided by total voxels in the field. SORE6+ cells were manually counted in each 3D hyperstack. In PyMT model, mice were injected intraperitoneally with 200 μl of either control PBS liposomes or Clodronate liposomes (concentration 18.4 mM) every 2 days for 2 weeks. Mammary tumors from PBS liposomes or Clodronate liposome-treated mice were extracted. FFPE tumor sections were stained with Iba1 IHC and macrophage density was calculated by the area covered by macrophages (Iba1 IHC, brown staining, Supplementary Fig. S5f) divided by the total area of the field. For Sox9^Hi quantification, FFPE tumor sections were stained with Sox9 antibody at 1 : 100 dilution (EMD Millipore, cat# ab5535) and Sox9Hi tumor cell population was defined by choosing ~top 5% of the tumor cells with the highest Sox9 fluorescence signal in the control tissues, which corresponded to a fluorescence intensity threshold range of 80–255. Same threshold was applied to the clodronate tissue. Both control and clodronate tissues were imaged at the same excitation intensity and exposure times on 3D Histech Pannoramic P250 digital whole slide scanner. To assess potential effects of clodronate directly on tumor cells, unsorted MDA-MB-231 cells containing the

SORE6>GFP sensor were treated with 25 or 100 μM clodronate for 48 h and SORE6+ cells in the population were assessed at endpoint by flow cytometry as above.

**In vivo CSC identification in fixed tissue with anti-FLAG IF staining.** Primary MDA-MB-231 xenograft tumors from SCID and nude mice, expressing tdTomato + SORE6>GFP or tdTomato + minCMV>GFP, were excised, FFPE, and 5 μm sections were cut using a cryostat. For IF staining for the FLAG tag on the dsCopGFP, sections were deparaffinized, followed by antigen retrieval in 1 mM EDTA pH 8 for 20 min in a conventional steamer and 20 min incubation at room temperature. Tissues were washed 3 × 5 min and incubated in blocking buffer (1% BSA + 5% goat serum) for 1.5–2 h at the room temperature, followed by rat anti-FLAG at 1 : 1000 (BioLegend, cat# 637301) incubation overnight at 4 °C and secondary antibody goat anti-rat Alexa 647 for 1 h at room temperature. The FLAG channel in the minCMV>GFP tissue was used to set the background in SORE6>GFP tissue, to identify single stem cells. For in vivo characterization of FLAG antibody (Supplementary Fig. S2b), SORE6>GFP sensor expressing MDA-MB-231 primary tumor FFPE sections were treated with 1× citrate pH 6.0 buffer (PerkinElmer, cat# AR6001) and stained with chicken anti-GFP antibody at 1 : 100 (Novus, cat# NB100-1614) and rat anti-FLAG antibody at 1 : 200 dilutions.

**In vivo invadopodia and degraded collagen IF staining, imaging, and quantification.** In live tissue (Fig. 2c–e), invadopodia are identified morphologically as thin oscillatory protrusions with 8–10 min cycle[36,40]. For invadopodia identification in fixed tissue, primary MDA-MB-231 xenograft tumors were excised from SCID mice expressing tdTomato + SORE6>GFP or tdTomato + minCMV>GFP. FFPE tissues were deparaffinized, followed by antigen retrieval in 1 mM EDTA pH 8 for 20 min in a conventional steamer. Tissues were incubated in blocking buffer (1% BSA + 5% goat serum) for 2 h at the room temperature, followed by primary antibodies overnight at 4 °C and secondary antibodies for 1 h at room temperature. In fixed tissue (Fig. 2h), invadopodia are identified molecularly by Cortactin + Tks5 colocalized dots[36,43]. Cortactin + Tks5 colocalized puncta within the cell boundary marks invadopodia in these cells, consistent with published literature[42,43,113]. We did not use phalloidin staining, as F-actin is present in other cellular structures (e.g., filopodia, focal adhesions, etc.) and is not a specific marker of invadopodia. Degraded collagen was detected with C1,2C (Col 2 3/4C_short) rabbit antibody at 1 : 100 dilution (Ibex Pharmaceuticals)[38]. Degraded collagen staining colocalizing with cortactin, one of the invadopodia marker, identifies mature invadopodia (i.e., actively degrading shown by white arrow in Fig. 2k). The position of the cleaved collagen does not always colocalize with invadopodia, This is a well-known issue with cleaved collagen[38]. The reason is that cleaved collagen signal represents the current and historical buildup of degraded collagen as different invadopodia at different times degrade different areas of ECM and move on. Stem cells were identified with rat anti-FLAG antibody as described above.

The slides were imaged on a 3D Histech Pannoramic P250 digital whole slide scanner, using a ×40, 0.95 NA air-objective lens. Digital whole slide scan tissue images were opened in CaseViewer v2.1. FLAG channel intensity in minCMV>GFP tissue was used to set the background in SORE6>GFP tissue to identify single stem cells and five to six random regions of interest (ROIs) at ×40 from non-necrotic areas were saved. Images were opened in ImageJ and a custom ImageJ macro was used for dividing each image into stem cell and non-stem cell areas. Cell areas were defined as the circular 36 μm diameter ROI around each single stem cell (3× diameter of a single cell, ~12 μm to capture all the extending invadopodia protrusions and their collagen-degrading activity from each cell). Invadopodia were identified as cortactin and Tks5 containing puncta and normalized to the size of stem and non-stem cell areas. Accordingly, collagen degradation was calculated as the Col3/4-positive area (%) in stem and non-stem cell areas as described above. In addition, a higher resolution analysis, to assign the origin of the cortactin and Tks5 containing puncta signal to SORE6+ or SORE6− cells, was done by reanalyzing random fields in Fig. 2j, m by drawing the boundary of FLAG+ cells manually (as shown in Fig. 2h, k) to define stem cell and non-stem cell areas. Re-analyzed results (Supplementary Fig. S4f, g) led to the same conclusions as that obtained using the automated 36 μm ROI method (Fig. 2j, m), i.e., compared to non-stem cells, stem cells have significantly more invadopodia and higher degradation activity in vivo.

**CSC distance analysis relative to TMEM, and blood vessels without TMEM, in fixed tissue in vivo.** Sequential sections from primary MDA-MB-231 xenograft tumors from SCID mice, expressing tdTomato + SORE6>GFP or tdTomato + minCMV>GFP, were stained for TMEM IHC (mouse anti-Mena at 1 : 1000, rabbit anti-Iba1 at 1 : 5000, and rat anti-Endomucin at 1 : 50)[15,60] and anti-FLAG to identify SORE6+ stem cells. TMEMs were identified manually by a pathologist. Stem cells were identified by thresholding the anti-FLAG image based on the threshold value obtained from the anti-FLAG-stained minCMV>GFP control tumor section. TMEM IHC and FLAG images were aligned in ImageJ using Landmark Correspondences plugin. For scoring purposes, each TMEM doorway was visualized as 60 μm diameter circle and distance of all CSCs were measured to its nearest TMEM circle using a custom-written ImageJ macro (provided as a supplementary software). Cells that were inside or touching the TMEM boundary

were counted in data point 0 μm in Fig. 6d, g, with subsequent cell counts measured within each 40 μm interval area contour moving away from the TMEM. For non-stem cell distance with the nearest TMEM in the field, DAPI channel was thresholded followed by watershed algorithm to find non-stem cells. Distance of each non-stem cell to its nearest TMEM was calculated using the same ImageJ macro as above. The distance of CSCs to TMEM was calculated in reference to TMEM, not the entire length of the blood vessel that contains TMEM (TMEM and blood vessel without TMEM schematics shown in Fig. 6c, e). To exclude the possibility of CSCs being near TMEM by chance, CSC distance histograms (Fig. 6d, f–h) were normalized by distances of all tumor cells in the field (DAPI staining) to nearest TMEM (Fig. 6d, g) or blood vessel without TMEM (Fig. 6f, h). Similarly, stem cell distance analysis with nearest blood vessel without TMEM was done by thresholding the blood vessel channel and removing blood vessels that contained TMEM. For scoring purposes, each blood vessel is visualized as an ROI delineating the vessel boundary and distance of all CSCs were measured to its nearest blood vessel using the ImageJ macro as above. Cells that are touching the vessel are counted in data point 0 μm in Fig. 6f, h, with subsequent cell counts measured within each 40 μm interval area contour moving away from the vessel. All distance histograms were analyzed and plotted in Excel.

For MenaINV staining, MDA-MB-231 FFPE primary tumor tissues were stained with chicken anti-MenaINV antibody[19] at 1:200 dilution.

**In vitro tumor cell-macrophage co-culture assay**. In vitro, the baseline SORE6 positivity for the MDA-MB-231 subline used in the current study varied from ~15% to 40%. Tumor cells (tdtomato MDA-MB-231 SORE6>GFP) were plated alone or with macrophages (BAC1.2F5) or HUVEC (Lonza, cat# CC-2517) at a 1:5 ratio (5000 tumor cells and 25,000 macrophages or HUVEC) in a 35 mm glass-bottom dish (Ibidi, cat# 81156) and allowed to adhere overnight. Next day, cell media was replaced with imaging media (L-15 + 10% FBS) and cells were imaged live on an epi-fluorescence DeltaVision microscope (GE) with a ×20 objective and CoolSNAP HQ2 CCD camera. For live imaging, tumor cells were plated overnight. Next day, media was replaced with imaging media, followed by macrophage addition at the beginning of live imaging. To identify stem cells, control tdtomato MDA-MB-231 minCMV>GFP cells plated under identical conditions were used to set the background GFP level. For Notch1 inhibition, 10 μM DAPT (Sigma, cat# D5942) or 20 μM SAHM1 (Sigma, cat# 491002), or vehicle control DMSO was added to the co-culture dish. For Notch activation, 50 μM Jagged1 (AnaSpec, cat# AS-61298) and its control, Jagged1 scramble (AnaSpec, cat# AS-64239) was added in solution to the co-culture dish. Cells were incubated with peptides overnight before the measurements were made. For Notch signaling blocking antibodies experiment, TC + Mac co-cultures were treated with 20 μM IgG (BioLegend, clone: HTK888, cat# 400901) or 20 μM Jag1 (BioLegend, clone: HMJ1-29], cat# 130902), or 20 μM Jag2 (BioLegend, cat# 131001) or 20 μM Jag1 + 20 μM Jag2 in solution overnight. To confirm the co-culture observations independently in different tumor models using different analysis procedures, 10,000 4T1 cells transduced with SORE6>dsEGFP (baseline %SORE6 positivity 3.3 ± 0.4%) were co-cultured with 50,000 BAC1 cells in BAC medium (α-MEM, 10% FBS, 10 ng/ml CSF-1) for 4 days and then trypsinized for flow cytometry analysis to identify the SORE6+ cells as % total tumor cells using the minCMV>dsEGFP as a gating control. BAC1 cells were gated out using the F4/80 macrophage marker. Similarly, MDA-MB-231-LM2 cells expressing a GFP volume marker were transduced with a SORE6>dsmCherry stem cell sensor (baseline %SORE6 positivity 21.2 ± 0.5%). Next, 10,000 tumor cells were co-cultured with 20,000 BAC cells in BAC medium for 36 h and then analyzed by FACS to determine SORE6+ CSCs (mCherry+ cells) as % total tumor cells (identified by GFP volume marker). To assess the requirement for direct contact, BAC cells were seeded into Millicell Hanging Cell Inserts (Millipore Sigma, 3 μm pore-size PET membrane), with tumor cells in the well below, so that direct cell contact was prevented but secreted factors could be freely exchanged between the two cell types.

**Microarray for Notch ligands**. For analysis of gene expression of Bac 1.2F5 cells, RNA was isolated using a Qiagen RNeasy mini kit. The RNA was quality controlled using an Agilent Bioanalyzer and then provided to the Einstein Genomics Core. The core performed labeling, hybridization, scanning, and RNA normalization using an Affymetrix Mouse Gene 2.0 ST microarray. The data discussed in this publication have been deposited in NCBI's Gene Expression Omnibus and are accessible through GEO Series accession number GSE185443.

**Notch1 signaling inhibition in vitro using siRNA**. AllStars Neg. Control siRNA (1027281) was from Qiagen and On-target plus human Notch1 siRNA smart-pool was from Dharmacon (Thermo Scientific, cat# L-007771-00-0005). A total of 1.5 × 10^6 tumor cells (tdtomato MDA-MB-231 SORE6>GFP) were transfected with 10 μl of 20 μM siRNA stock solution using Nucleofector Kit V from Lonza (cat# VCA-1003) for 48 h, as described previously[19]. Notch1 knockdown was confirmed using western blotting with the following antibodies: rabbit anti-Notch1 at 1:1000 (Cell Signaling, cat# 3608), rabbit anti-Notch2 at 1:1000 (Cell Signaling, cat# 5732), and mouse anti-GAPDH at 1:10,000 (Abcam, cat# ab8245). At 36 h post siRNA transfection, tumor cells were plated with

macrophages (1:5 ratio) overnight. Next day, cells were imaged live on Delta-Vision microscope as described above.

**Notch signaling inhibition in vivo using DAPT**. DAPT (Sigma-Aldrich Cat# D5942) was reconstituted in 100% ethanol to a stock concentration of 20 mg/ml, then further diluted in corn oil to a final concentration of 2 mg/ml. Eight-week-old PyMT mice bearing palpable tumors and separate cohort of SCID mice with MDA-MB-231 xenograft tumors expressing tdTomato + SORE6>GFP or tdTomato + minCMV>GFP were given daily intraperitoneal injections of 10 mg/kg DAPT or vehicle control (1:10 ethanol in corn oil) for 14 days. On day 15, the primary tumors, lungs, and duodenums were collected from the mice and fixed in 10% formalin. Mice were weighed on day1 and day 15, to determine that no significant weight loss was suffered due to the DAPT treatment. Duodenums were stained using the Periodic acid-Schiff staining.

In PyMT model, primary FFPE tumor sections were stained with Iba1 antibody and macrophage density was calculated by the area covered by macrophages divided by the total area of the field. For Sox9 quantification, FFPE tumor sections were stained with Sox9 antibody at 1:100 and Sox9Hi tissue coverage area was quantified. For NICD1 and Hes1 staining, FFPE tumor sections from PyMT mice bearing SORE6>GFP and treated with either control or DAPT were stained with either Cleaved Notch1 (Val1744) (D3B8) at 1:200 (Cell Signaling, cat# 4147) or Hes1 antibody at 1:100 (Millipore, cat# AB5702). In MDA-MB-231 model, FFPE lung sections were stained with FLAG antibody to identify SORE6+ CSCs.

**Time-lapse imaging of in vitro tumor cell-macrophage co-culture**. In vitro time-lapse movies of co-cultured cells were taken with a DeltaVision microscope at 2 min intervals for 16 h, at ×20 magnification. To avoid phenol red auto-fluorescence, the cells were plated into L-15 culturing media. Fixed culture plates were also imaged at ×20; these images were used to determine percentage of SORE6+ cells by GFP intensity, thresholded based on the background GFP expression in vector control cell, tdtomato MDA-MB-231 minCMV>GFP. For cell-fate mapping in co-culture experiment shown in Fig. 4f, 25,000 tumor cells (MDA-MB-231 SORE6>GFP) were plated in BAC culture media in 35 mm glass-bottom dish (Ibidi, cat# 81156) and allowed to adhere overnight. Next day, media was changed to L-15 + 10% FBS + CSF-1 media and cells were imaged live every 10 min using an epi-fluorescence DeltaVision microscope (GE) with a ×20 objective and CoolSNAP HQ2 CCD camera. After 30 min of imaging, 125,000 BAC cells were added to tumor cells in the dish on the imaging stage and imaging was continued for 30 h. Time-lapse movies were analyzed by identifying SORE6− cells in the frames before macrophage addition, based on the GFP threshold set by MDA-MB-231 mCMV>GFP fluorescence. SORE6− cells were manually tracked frame-by-frame for macrophage touch or no touch in the phase channel time lapse and the corresponding GFP fluorescence increase or no change in the GFP channel time lapse.

For cell-origin mapping in co-culture experiments as shown in Fig. 4g, MDA-MB-231 tumor cells were FACS-sorted to give a starting population that contained ~2% SORE6+ cells so as to simplify the mapping process. Next, 20,000 tumor cells were co-cultured with or without 30,000 BAC cells in 24-well plates in BAC medium and live imaged every hour starting at 6 h after plating, using an Incucyte S3 Live-Cell Analysis System at ×10 magnification (Essen BioScience). Cells that were SORE6+ at t = 30 h were origin-traced back to their starting state at t = 6 h, to determine whether they originated from pre-existing SORE6+ cells or from direct induction of a stem phenotype in SORE6− cells. The fraction of SORE6+ cells at 30 h that originated from SORE6− cells at 6 h was assessed by origin mapping 32 SORE6+ cells/well from four wells/condition, for a total of 128 SORE6+ cells/condition.

**Gelatin degradation assay and in vitro immunofluorescent staining**. Invado-podia matrix degradation assay was performed as described previously[37]. SORE6+ and SORE6− cells were FACS-sorted and plated on fluorescent Alexa-405-labeled gelatin matrix-coated plates overnight. Cells were fixed with 3.7% paraformalde-hyde for 15 min and permeabilized with 0.1% Triton X-100 for 5 min. Cells were blocked in 1% BSA + 5% FBS for 1 h, incubated with primary antibodies; anti-cortactin (1:400) and anti-Tks5 (1:100 dilution) for 1 h, followed by incubation with appropriate secondary Alexa antibodies (1:400) for 1 h. Cells were images on DeltaVision widefield fluorescence microscope. Images were processed in ImageJ.

**Intravasation assay (CTC analysis)**. About 1 ml of blood was drawn from the right heart ventricle of anesthetized mice bearing an ~1.5 cm diameter primary tumor. Blood was immediately fixed in 5% NBF at room temperature for 15 min, then washed with 10 ml PBS 3 times by centrifugation at 200 × g for 5 min. The pellet was resuspended in PBS/0.1% Triton X for 1 min, washed once, and stained with DAPI. The final pellet was resuspended in 300 μl PBS, dropped onto 35 mm glass-bottom μ-Dishes (Ibidi, cat #81156), and left in 4 °C overnight before imaging on an epi-fluorescence DeltaVision microscope (GE). Quantification of SORE6+ CSCs was done in ImageJ. CTCs from minCMV>GFP control reporter tumor mice were used for setting the GFP threshold.

**Live lung imaging**. The stem reporter cells tdTomato MDA-MB-231 SORE6>GFP or the reporter control cells tdTomato MDA-MB-231 minCMV>GFP were injected into the mammary fat pad of SCID mice (NCI). In ~6 weeks, SCID mice bearing orthotopic MDA-MB-231 SORE6+ tumors of size ~1 cm in diameter, underwent placement of permanent lung imaging window as previously described[44]. Large volume high-resolution intravital (LVHR-IVI) imaging[109] was performed through the window 24 h post implantation (Day0), to capture the architecture of lung vasculature. Vasculature was labeled with an intravenous injection of 50 μL of a fluorescently tagged dextran (50 μg/mL, 10kD Cascade Blue Dextan). Twenty-four hours later (Day1), LVHR-IVI imaging was repeated to capture disseminated tumor cells that arrived in the lung vasculature. SORE6− and SORE6+ cells were distinguished by the presence or absence of the green GFP signal in addition to the red tdTomato cell volume marker present in all tumor cells.

**Ex vivo lung processing**. SCID mice, bearing 1–1.5 cm orthotopic MDA-MB-231 SORE6+ tumors, were killed and lungs were inflated with 5% formalin injected into trachea. Lungs were removed and incubated in 5% formalin/20% sucrose solution overnight at 4 °C. Next day, lungs were embedded in OCT compound and 7 μm-thick sections were cut. Slides containing lung sections were equilibrated at room temperature for 10 min, permeabilized in cold acetone for 10 min, and washed once with PBS. Slides were then mounted with mounting media containing DAPI (Vector Laboratories, Burlingame, CA). Slides were imaged on P250 scanner and images were acquired in DAPI, fluorescein isothiocyanate (FITC) (SORE6-GFP), and TRITC (tdTomato) channels.

**Quantification of micrometastatic foci in the lung**. After the termination of the DAPT treatment scheme and animal sacrifice, lungs were collected and immersed in 10% formalin. The tissues were processed for histological examination by embedding in paraffin. Two consecutive sections cut at 50 μm intervals were obtained for each lung and stained for hematoxylin and eosin. Metastatic foci containing >5 cancer cells were counted in the tissue section of the entire lung in both slides for each case.

**Effect of passage through circulatory system on CSC representation in tumor cell population**. Half a million unsorted MDA-MB-231 cells transduced with the SORE6 reporter and a constitutive volume marker were injected into the tail vein of 6- to 8-week-old female nude mice. The % SORE6+ cells at the time of injection ($t = 0$) was assessed by flow cytometry. At $t = 6$ h, lungs were collected and immediately imaged using a Zeiss LSM 880 microscope. The % SORE6+ cells in the lung at $t = 6$ h after tail vein injection was assessed by counting SORE6+ and SORE6− cells in 25 random ×40 high-power fields/lung for each of 3 mice (347, 450, and 271 total cells/mouse). SORE6+ cells were identified using the minCMV reporter as a gating (flow) or thresholding (microscopy) control.

**Human breast cancer tissue collection**. Breast cancer tissue was collected from 49 patients under the Albert Einstein College of Medicine/Montefiore Medical Center Institutional Review Board protocol approval. Patient consent was not required for this study (IRB# 2016-7193 & IRB# 2017-8158). Cancer cells were collected by FNA from unfixed mastectomies and lumpectomies and used for the assessment of stem cell marker expression as described below. FNA primarily collects loose tumor cells, with very few macrophages and no endothelial cells and incurs minimal tissue damage[114]. Briefly, five to ten FNA aspiration biopsies per tumor were performed on at least 3 areas of grossly visible tumor using 25-gauge needles. The adequacy of the sample was assessed by the standard Diff-Quick protocol[115]. Only samples composed of at least 90% malignant epithelial cells, as determined by standard pathologic characteristics, were used for the analysis[116]. The cells obtained by FNA were analyzed by flow cytometry for the expression of CD44+/CD24−. They were also analyzed for the expression of ALDH1 and CD133 by in situ hybridization and qRT-PCR. After cell collection by FNA, the remaining tissue was FFPE and sent for routine pathological analysis. FFPE tissue blocks with representative tumor were used for triple-IHC staining of TMEM and TMEM doorways score analysis. Thus, the TMEM analysis and the analyses for the expression of stem cell markers were blinded and done on the very same tumor. This approach allowed us to assess the percentage of cancer cells expressing stem cell markers and the density of TMEM doorways from the same samples (Fig. 8b).

**Tissue selection for TMEM staining and scoring**. At the time of routine microscopic examination of the lesions on which FNA biopsies had been performed, an appropriate area containing invasive cancer suitable for TMEM analysis was identified by low power scanning. The following criteria were used: high density of tumor, adequacy of tumor, lack of necrosis or inflammation, and lack of artifacts such as retraction or folds. TMEM stain is a triple immunostain for predicting metastatic risk in which three antibodies are applied sequentially and developed separately with different chromogens on a Bond Max Autostainer. We used the pan-Mena mouse monoclonal antibody at 1 : 1000 (BD# 610693), CD68 at 1 : 300 (Dako # M0876), and CD31 at 1 : 500 (Dako # M0823). The assessment of TMEM scores was performed using Adobe Photoshop Version: 2015.1.2 on 10 contiguous 400× digital images of the most representative areas of the tumor. The total TMEM for each image was tabulated and the scores from all ten images were

summed to give a final TMEM density for each patient sample, expressed as the number of TMEM per 10,400× fields (total magnification)[59,60]. Twenty-five randomly chosen cases were each independently scored by two pathologists. As the correlation between the scores read by 2 pathologists was excellent with the correlation coefficient $r = 0.97$, the remaining 24 cases were scored by one pathologist.

**Flow cytometry analysis of human breast cancer specimens for CD44+/24−**. The cells obtained by FNA were washed with PBS and filtered through 35 μm cell strainer to make it into a single cell suspension (4 °C was maintained all the time). The cells were then diluted to 500 cells/μl in PBS to 400 μl and divided into two eppendorf tubes at 200 μl each. In the first tube, 2 μl each of the isotype controls for CD44 and CD24 were added. Then, in the second tube each of two antibodies anti-human/Mouse CD44 PE-Cyanine5 and anti-human/Mouse CD24 PE were added. These solution mixtures were kept at room temperature for 20 min in the dark. The samples were then run through GUAVA flow cytometer. The first tube was run to set up the quadrants and antibody isotype controls were used to setup the gain in the machine. Then the samples with antibodies were analyzed. Cells present in lower right quadrant showed the % of CD44+/CD24− cell population.

**In situ hybridization for ALDH1 and CD133**. In situ hybridization was performed using Cy3 labeled ALDH1 and CD133 probes (Aanera Biotech, Albany, NY), the cell membrane was stained with 5-Hexadecanoylaminofluorescein and the nucleus with and DAPI (Thermo Fisher Scientific, Waltham, MA). The cells were fixed in 3% formaldehyde and permeabilized using 0.1% Triton-100 buffer. Fifty nano-grams of probe was mixed with 100 μl of hybridization buffer containing 1 mg of tRNA, 10 mg Dextran Sulfate, 20 μl of deionized formamide, 10 ng of N30 oligo-nucleotide in 2× saline-sodium citrate (SSC) and hybridized at 37 °C in dark in a shaker incubator at 50 r.p.m. The cells were resuspended in 10% deionized for-mamide in 2× SSC and incubated at 42 °C in dark in a shaker incubator at 50 r.p.m. Re-suspend cells in wash buffer containing 10 ng/μl DAPI for nuclear counter-staining and 200 ng of FITC dye an incubate for 5 min at 37 °C in dark. Recon-stitute in 50 μl of Anti-fade Mounting Buffer (Thermo Fisher Scientific, Waltham, MA) and mount in a glass-bottom petriplate (MatTek Corporation, Ashland, MA). Imaging was performed for DAPI, FITC, and Cy3 channels. Using ImageJ, the FITC area labeling the membrane and DAPI-stained nucleus area was identified. Cy3 signal was quantified in the donut-like cytoplasmic area and was compared to the TMEM values.

**qRT-PCR for ALDH1 and CD133**. Total RNA was amplified as previously described[117]. Briefly, 1× lysis buffer was added to the cancer cells. Primer mix (50 and 100 μg/μl) was added to the lysis solution. After incubation (80 °C for 10 min), RT master mix (5× First-strand buffer, dithiothreitol, 10 mM dNTP mix, RNase inhibitor, SS III enzyme) was added. This reaction mixture was incubated at 42 °C for 60 min for reverse transcription. cDNA was purified by ZYMO column and 1× wash buffer. Purified cDNA was eluted by nuclease free water. Tailing master mix (10× TdT Buffer, 10 mM dTTP, TdT enzyme) was added to the purified cDNA for TdT tailing. The reaction mixture was incubated for 37 °C for 2 min, 80 °C for 10 min, and 4 °C for 2 min. Then promoter synthesis master mix (10× Klenows buffer, 50 μg/μl T7 promoter oligo, dNTP mix, Klenow enzyme) was added to the poly A tailed cDNA solution. This reaction mixture was incubated at 22 °C for 30 min, hold at 4 °C for T7 promoter synthesis. Finally, in vitro transcription with T7 polymerase was done by T7 high-yield RNA synthesis kit (New England Bio-labs). Purification of mRNA was done by Qiagen mini column. After eluting the amplified mRNA, was quantified spectrophotometrically using the NanoDrop 2000 spectrophotometer (Thermo Scientific). After the first-round amplification, 100 ng RNA was taken for second-round amplification (same procedure described above). The first-strand cDNA synthesis was carried out by N9, dTVN, Superscript transcriptase II, dNTPs, RNAase, using this 2 μg of second-round amplified RNA. Quantitative PCR was performed with stem cell marker-specific primers. The expression of each gene was quantified by using the comparative Ct (ΔCt) method as described in the Assays-on-Demand User's Manual (Applied Biosystems). The fold values ($X$) were calculated using the formula: $X = 2(-\Delta Ct)$, where the data for the sample and sham-treated cell (here, the calibrator being the sham-treated cells) were first normalized against variations of sample quality and quantity. The ΔCt was determined using the formula: $\Delta C(t)$ sample = $C(t)$ target gene of sample − $C(t)$ reference gene. The expression of the target genes was normalized to GAPDH.

**Sox9 and triple-IHC TMEMs staining in human samples**. The FFPE human tumor sections with known TMEM scores[18] were stained with Sox9 antibody at 1 : 100 dilution (EMD Millipore, cat# ab5535). Sections were then scanned on 3D Histech Pannoramic P250 digital whole slide scanner and images were analyzed in Visiopharm program by first thresholding them based on the negative control (secondary antibody only) and then measuring the mean Sox9 fluorescence intensity. For Sox9Hi CSC distribution with respect to the TMEM doorways, sequential tissue sections were stained with triple-IHC (pan-Mena, CD68, and CD31 antibodies) and Sox9 antibody. TMEM-rich areas are defined as at least three TMEM doorways per ×40 magnification field of view, or 30 TMEM doorways per the ten ×40 fields (TMEM score). This cutoff point is chosen, because TMEM score of 23 is associated with increased risk of developing metastatic disease[59]. Sox9Hi

tumor cell population was defined by choosing approximately top 5% of the tumor cells with the highest Sox9 fluorescence signal, which corresponded to a fluorescence intensity threshold range of 60–255. All slides were scanned at the same excitation intensity and exposure times on the whole slide scanner.

**Statistical analysis**. GraphPad Prism 7 and Excel were used to generate graphs/plots and for statistical hypothesis testing. Statistical significance was determined by either *t*-test (normally distributed paired or unpaired dataset), Mann–Whitney test (non-normally distributed unpaired dataset), or Wilcoxon matched-pairs test (non-normally distributed paired dataset). Normality (Gaussian distribution) of each dataset was checked with D'Agostino and Pearson's normality test in GraphPad Prism[118]. Statistical significance was defined as $p$-value < 0.05. For the correlation of TMEM density with the proportion of cancer cells expressing stem cell markers, we used Pearson's correlation.

**Reporting summary**. Further information on research design is available in the Nature Research Reporting Summary linked to this article.

## Data availability
The data supporting the findings of this study are available within the paper and its Supplementary Information files. Any other relevant data including the original image files and the data behind all quantification plots are available from the authors upon request. Source data are provided with this paper.

## Code availability
ImageJ/Fiji macro for the "Shortest distance analysis" of CSCs from the nearest TMEM or blood vessels is provided as a supplementary software with this paper.

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

## Acknowledgements

We thank the Analytical Imaging Facility at Albert Einstein College of Medicine for microscopy help, particularly Dr. Peng Guo for help with Imaris 3D reconstructions; Yu Lin for help with TMEM staining; members of the Condeelis, Oktay, Segall, Cox, Entenberg, and Hodgson laboratories for helpful discussions, and Jen Mehalko for cloning expertise. This research was supported in part by CA150344, CA100324, CA216248, CA255153, F32 CA243350, an IRACDA fellowship, K12 GM102779, the Gruss Lipper Biophotonics Center and its associated Integrated Imaging Program, and SIG #1S10OD019961-01 and P30CA013330 (Flow Cytometry Core Facility). The research was supported in part by the Intramural Research Program of the NIH grant ZIA BC 005785 to L.M.W. This research was supported by Jane A. and Myles P. Dempsey.

## Author contributions

Conceptualization—V.P.S., B.T., Y.W., J.S.C., L.M.W., and M.H.O. Methodology—V.P.S., B.T., Y.W., G.S.K., D.E., C.L.D., R.J.E., and M.H.O. Formal analysis—V.P.S., B.T., Y.W., G.S.K., E.A.X., G.K., X.Y., J.E.S., L.M.W., J.S.C., and M.H.O. Software—V.P.S. and D.E. Investigation—V.P.S., B.T., Y.W., G.S.K., E.A.X., L.B., A.C., C.L.D., R.J.E., G.K., J.E.S., and M.H.O. Writing—V.P.S., B.T., Y.W., D.E., W.G., J.S.C., L.M.W., and M.H.O. Funding acquisition—J.S.C., L.M.W., M.H.O., and D.E. Resources—J.G.J., E.G., N.A., S.R., G.B., E.A., C.R.S., D.E., S.G., and J.E.S. Supervision—M.H.O., J.S.C., L.M.W., and D.E.

## Competing interests

The authors declare no competing interests.

## Additional information

**Peer review information** *Nature Communications* thanks Amaia Martinez-Usatorre, Lucio Miele and the other anonymous reviewer(s) for their contribution to the peer review this work. Peer reviewer reports are available.

