## [Peer Review File · Nature Communications]

Reviewers' Comments:

Reviewer #1:

Remarks to the Author:

The manuscript by Sharma and colleagues uses elegant intravital microscopy and cell tracing techniques to describe macrophage-induced phenotypic conversion of model breast cancer cells into cancer stem cells (CSC), the association of CSC at TMEMs and in CTC as well as their role in establishing lung metastases. A Jagged-Notch signal is proposed as non-cell autonomous pathway mediating the effects on macrophages on CSC generation. Examination of patient-derived FNA cells and corresponding surgical specimens yielded results consistent with the model proposed by the authors.

The manuscript is well-written, innovative and generally strong. Addressing the following issues will further strengthen the manuscript.

1. Figure 2Di and Dii: The rightmost panel in figure Di (01:34:30) is identical to the middle panel in figure Dii. Is this intentional? The figure legend is unclear.

2. Figure 4H: the experiments, as described in Materials and Methods, consisted of adding recombinant Jagged1 or a scrambled peptide to cell cultures. It is unclear whether Jagged1 was added in solution or used to coat the surface of wells prior to cell addition. This is important as Notch receptor activation requires subunit separation, which is achieved more easily when ligands are associated with the cell membrane of a signaling cell or bound to a solid surface.

3. Figure 4I: did the macrophages used for co-culture express Jagged1? What happens if Jagged1 is knocked down in the macrophages? Is Jagged1 the only Notch ligand capable of inducing stemness? This is a distinct possibility, because the kinetics of Notch activation by different ligands are different. However, it should be tested experimentally. Endothelial cells can also express Jagged1 and they do express DLL4. Since HUVEC do not induce stemness (Figure 4D), it is possible that another signal or a different Notch ligand (e.g. DLL1) is involved.

4. Figure 4J: the effectiveness and specificity of the Notch1 siRNA should be demonstrated by Western blotting.

5. Figure 4I: the use of DAPT is a good indication of Notch involvement but it is not conclusive proof. Gamma secretase has >95 substrates including the Notch receptors. An orthogonal method of Notch inhibition (e.g., dominant negative MAML1, a cell-penetrating MAML1 decoy peptide should be used

6. Similarly, for figures 4K and J, proof of target inhibition by DAPT should be presented (e.g., staining for cleaved Notch1 or decreased expression of Notch transcriptional targets such as NRARP, Myc, or Notch3 should be presented)

7. Figure 7G: Is Jagged1 detectable on the surface of macrophages in contact with MENA+ cells in the context of TMEMs in human tissue? Are the MENA+ cells in contact with macrophages Notch1-positive? This would be an important confirmation that the mechanism identified in the MDA-MB231 and PyMT models operates in human tumors. As investigational monoclonal antibodies that inhibit Jagged1 exist, data supporting the existence of the pathway discovered by the authors in human tumors would significantly increase the translational relevance of their manuscript.

Reviewer #2:

Remarks to the Author:

This is a timely and important subject with clear relevance to cancer pathogenesis. Using innovative intravital microscopy to obtain very high-quality in vivo images of the primary and metastatic tumour sites, the authors investigated the phenotype of individual CSCs during tumour dissemination and seeding. Using a cancer stem cell (CSC) reporter they propose that CSC intravasate more readily circulate and form metastases. They propose that the CSC phenotype is driven by macrophage induced Notch signalling.

The manuscript as a whole is fairly clear, although some important questions arise from the data presented that need to be addressed (see below).

Major Comments

1/ Based on imaging in fixed tissue and IVM, the authors propose that macrophage contact with cancer cells results in increased reporter signal, indicating development of a CSC phenotype. As it

is hard to control for the cells involved (and to some extent the z resolution probably doesn't help) they used a reductionist co-culture of tumour cells and macrophages in vitro to show this in more detail. However, the representative movie (Movie 11) doesn't seem to support the author's interpretation. At 00:40 the tumour cell on the left is already brightly green fluorescent, before the macrophage contact depicted in the stills Figure S6B at 2:50. Is this movie representative? How often was there a clear increase in GFP fluorescence after macrophage contact with a formerly GFP -ve tumour cell in this experiment?

2/ That the inhibition of Notch signalling presented in Figure 6 reduces the number of CSCs in the lung is intriguing. Importantly does this reduction in CSCs lead to a reduction in metastasis and an increase in survival?

Other comments

Figure 2.

The authors show that non-CSCs have a high-speed motile migration phenotype, whilst CSCs have a more slow-moving migratory phenotype. However, in Movie 7 the non-CSC cells barely move compared with Movie 6, is there a difference in their motility when they are close to the blood vessels compared with being further away?

2G: only shows the Tks5 and Cortactin staining in CSCs, what do these look like in non-CSCs?

Figure 3.

60-70% of the CSCs in live tumours are in contact with macrophages. How long do the CSCs and macrophages stay in contact? What is the percentage of non-CSC cancer cells in contact with macrophages?

3B: appears to show some macrophages which haven't been counted in the field of view

3D: how many mice underwent clodronate, how was the analysis was done?

3G: A supplemental figure showing the IHC for macrophage depletion would be helpful.

Figure 4.

4J: The authors show that CSCs which have inhibited Notch signalling incubated with macrophages have reduced CSC conversion compared with controls. This could be strengthened by timelapse imaging. Does Notch signalling affect the motility of the cancer cells and therefore their interaction with macrophages?

Figure 6

6F: How long after tumour implantation was the intravital imaging done?

It is understood that space is limited, however, a clearer statement of the tumour models being presented in each figure legend would be helpful in interpreting the data. Additionally, the number of mice is not clear in all of the figure legends.

Reviewer #3:

Remarks to the Author:

In this study, Sharma et al. employ intravital imaging to provide an important and interesting study on the nexus between cancer stem cells, macrophages, and key steps in metastatic dissemination. The manuscript is generally well written and the data are for the most part clearly presented. In essence, the authors use a biosensor for SOX2/OCT4 stem cell transcription factors in cancer cells and show that its activity correlates with tumor-initiating behaviors, and, furthermore, that this malignant cell state is enhanced when cancer cells are brought into close proximity with macrophages near endothelial networks in vivo, at sites referred to as the Tumor Microenvironment of Metastasis. A few experiments hint at a potential mechanism involving the induction of Notch signaling in cancer cells by way of Jagged1 release from macrophages, and several experiments also implicate MENA-invasive in this process as part of the body of literature that surrounds Dr. Oktay's research. The MDA-MB-231 cell line serves as the main workhorse of the study, although a few key findings are loosely corroborated in other cell lines and some correlative data is included from an analysis of patient samples.

Intravital imaging is a challenging technique to perform, and to recommend additional intravital imaging experiments seems excessive from the standpoint of this reviewer. However, it is important to recognize that claims which lean on intravital imaging can sometimes be disadvantaged by low sample sizes and, as such, many additional experiments could be performed in vitro to lend crucial support for the author's major claims, especially those that surround intravasation events. There is also notable concern about the specificity of the biosensor system used in the study; at the very least, this must be clearly addressed in the text. Therefore, in the opinion of this reviewer, the study by Sharm et al. could be appropriate for publication in Nature Communications only if the major issues outlined below are adequately addressed prior to resubmission. If a majority of the points below could be successfully tackled or convincingly refuted during a major revision, then this study would be worth reconsidering for publication in Nature Communications.

Major Issues Needing to be Addressed:

Figure 1:

There is some concern about the specificity of the SORE6 biosensor for SOX2/OCT4. A previous paper from the Wakefield group (Tang et al., Stem Cell Reports 2015) introduces the SORE6 sensor. The supplemental data for that paper provides the sequence for the SORE6 minigene and indicates the SOX2/OCT4 response elements embedded throughout. The sequence for the minigene is as follows:

```
5'atctatcgatcagctacttttgattacaatggccttggtgcagctacttttgattacaatggccttggtgcagctacttttgattaca  
tgcccttggtggaattccagctacttttgattacaatggccttggtgcagctacttttgattacaatggccttggtgcagctactttgca  
ttacaatggccttggtgactagttcta-3'
```

Entering this sequence into an online transcription factor binding site prediction database like PROMO reveals numerous putative binding sites for various other transcription factors including XBP-1, c-JUN, CEBPA. How confident can the reader be that the SORE6 sensor is specific to SOX2/OCT4? Some of these other transcription factors are associated with cell stress, so one wonders about the extent to which the SORE6 biosensor could be a cell stress sensor; many of the phenotypes presented herein could be explained by increased cancer cell stress, where the acquisition of stem-like (or the aggressive) features could accompany a stress response. The fact that the authors report SORE6 biosensor fluorescence without robust SOX2/OCT4 expression (Fig. S1E) lends credence to the idea that this system is promiscuous. To partially address this, for example, it could be useful to assess the expression of notable target genes for the aforementioned cell stress transcription factors in Figure S1E and in Figure S5B.

The Stem Cell Reports article in 2015 Tang et al. also appears to address the potential promiscuity of the SORE6 system by stating that the destabilized GFP boosts its specificity for CSCs, but these experiments are performed in the MCF-7 cell line and not in MDA-MB-231 cells. Conducting additional work in vitro to show sensor specificity for SOX2/OCT4 or, if that is not feasible, evaluating the status of additional stem cell markers is strongly recommended. It might also be important to acknowledge the potential caveats of this system clearly in the text. Performing additional supplemental experiments in vitro to establish whether SORE6 activation correlates with the acquisition of additional stem cell markers (e.g. wnt/notch pathway activation, CD44, CD133, ALDH1, etc.) or additional phenotypes (e.g. colony formation on soft-agar) is still another option.

Figure 1G from the Tang et al. paper also shows a baseline SORE6 positivity of around 40% for the MDA-MB-231 cell line in vitro, whereas the current Sharma et al. paper uses tumors in mice (e.g. Fig. 1D & S1F) to make the case that SORE6+ cells are a much smaller minority population generally consisting of less than 5% (Fig. 1D). Knowing the baseline of SORE6+ MDA-MB-231 cells in vitro is critical if the reader is to interpret key experiments presented later on, such as the co-culture data shown in Figure 4.

Figure 2:

In Figure 2 (Fig. 2c-d; S4c), two intravital imaging videos are provided that compare membrane protrusions emanating from SORE6+ vs SORE6- cells. Although the authors write that both cell types are located in close proximity to the blood vessel, the videos depict that the SORE6+ example cell is touching the blood vessel whereas the SORE6- is more distal. One wonders if a better controlled comparison could be provided for the reader? Perhaps at a minimum, a video for a SORE6+ cancer cell that is further away from a blood vessel would be useful?

Moreover, the invadopodia frequency data (Fig. 2c-f), while interesting, lacks robust quantification in its current form.

Figure 3:

Figure 3d-e purports to causally link a CSC phenotype to macrophages in vivo by administering clodronate to deplete macrophages in mice and by showing this correlates with a decrease in the amount of SORE6+ cells. Here it would be helpful to have access to the non-normalized data, so the reader can assess the total cell counts for each day. In addition, knowing what happens to SORE- cells during these timepoints would allow the reader to evaluate whether the SORE6+ cancer cells simply die, or if SORE6+ cells convert back into SORE6- phenotype during macrophage depletion. If the latter occurs, it would be highly interesting to the reader. If not, then some transparency is warranted.

Another important experiment may entail determining whether clodronate treatment impacts SORE6+ cancer cells in vitro, to lend support for the effect occurring through macrophages or even other microenvironmental cell types in vivo. Experiments like these would help one better judge whether the link is indeed casual like the authors claim.

Figure 4:

In Fig. 4E and S6D, the authors co-culture macrophages with MDA-MB-231-LM2 and 4T1 cells, and these cell lines are here first introduced into the story. The authors must better validate a link between SORE6 biosensor and stem-like features in these cell lines. At the very least, the authors should perform RT-qPCR to ascertain the expression of various stem cell genes for SORE6+/- cells in both lines in the manner depicted in Fig. S1E. Also, what is the baseline amount of SORE6+ cells in each cell line prior to the co-culture experiments?

The experiments conducted in Fig. 4H-J are crucial for establishing a mechanism. In its current form, however, this reviewer is not convinced that Jagged1 release by the macrophages activates a tumor-initiating phenotype in the cancer cells. Have the authors attempted to perturb the Jagged1 gene in the bac1.2f5 macrophage cell line, thereafter testing if this leads to a reduction in SORE6+ cancer cells after co-culture? If this is impractical, could the authors use Jagged1 antibodies to neutralize this factor in cell culture medium? Without key mechanistic experiments along this line, it is very difficult to accept that a Jag1-Notch axis explains the phenotype, given the small effect sizes depicted in Figure 4.

Figure 5/Figure 6:

The authors would greatly benefit from additional evidence indicating whether SORE6+ cells actually intravasate to a greater degree than SORE6- cells, as their enrichment in TMEM, their association with blood vessels, and their increased numbers in circulation are all circumstantial. At present, an intravital imaging movie of one intravasating SORE6+ cell is not enough to convince this reviewer that this cell state directly enhances this aspect of metastatic dissemination. Quantifying additional cells via intravital imaging may be too onerous, although the authors might consider further investigating the transendothelial migration potential of SORE6+ vs SORE6- cells in vitro using the Transwell platform or other systems, as a variety of assay types currently exist on the market. Microfluidic devices to assess vascular invasion represent yet another option. The manuscript uses the "doorway" and "portal" verbiage extensively, despite this aspect of the SORE6+ phenotype is not yet being convincingly shown. The large increase in SORE6+ cells in circulation (Fig. 6B) is truly fascinating but it may not necessarily be explained by TMEM, as it has not yet been established if SORE6+ cells intravasate to a higher degree of if SORE6- cells; SORE6-

cells may transform into SORE6+ in circulation.

The authors propose a mechanism whereby macrophages secrete Jagged1 to induce a CSC phenotype in metastasizing cancer cells. It seems important to address whether the administration of DAPT goes beyond reducing the number of SORE6+ cells (Fig. 6J) to also impact the degree of metastatic burden. An assessment of metastatic lesions in the mouse lung should be feasible for this experiment, given that mice were treated daily for 14 days prior to the collection of tissues for histology.

Figure 7:

In Figure 7D-E, the authors evaluate the following marker combinations to detect CSCs in human patient samples: CD44+/CD24-, ALDH+, and CD133+. They subsequently show that the expression of these markers correlates with the TMEM score. The current study would be significantly more cohesive if the authors assessed the status of some these markers (where applicable) in the cell lines being studied. For example, do SORE6+ MDA-MB-231 cells also harbor evaluated levels of CD44 or CD133?

Minor Issues to Consider:

In their introduction, the authors outline challenges associated with modeling stable CSC phenotypes in vitro, arguing that such phenotypes can "depend on signals from the tumor microenvironment." There is obviously validity to this notion and putting one's work in context is important, but these statements seem at odds with the many in vitro experiments embedded throughout the ensuing manuscript. At present, the introduction also reads like technical justification for using intravital imaging and the SORE6 biosensor, as opposed to covering topics that will aid its readers in understanding the biology. It is recommended that the introduction be expanded or edited to incorporate more content on macrophages and Notch signaling, and to explain how they relate to the phenomenon being studied.

For Fig. S1E: It would be useful to sort out GFP+ cells from the minCMV>GFP control to assess whether GFP expression by itself leads to the increased expression of stem cell genes. This would further boost confidence about the link between SORE6+ and stem-like features in cancer cells.

In Fig. 3C, the labels for "Mac contact" do not align perfectly with the bars.

In Fig. S5B, can the authors please expand their assessment of gene expression in Met-1 cells to also include the genes also examined in the MDA-MB-231 cell line? This would allow the reader to determine the degree of concordance between the models and whether the sensor functioning as described in that line. Of particular interest is whether Met-1 cells express Oct-4.

For Fig. 3G-H, can the graph or figure legend depict at what timepoint these data were collected? This would help the reader relate the results back to Figure 3D-E.

In Fig. 4D, what is the purpose of "Mac ◊ TC ◊ Notch ◊ CSC" above the graph? It is assumed this is a typo on the figure.

In Fig. S1G (or S1E), what is the baseline expression of NOTCH1 in SORE6+ vs SORE6- cells before the introduction of macrophages? This information seems important if the reader is to attribute its induction in cancer cells to macrophages, as the narrative for Figure 4 suggests.

In Fig. 5a: Should the figure legend on the image indicate labels for all colors? What do the authors make of the green SORE6+ cells that are ostensibly negative for tdTomato?

The description for Fig. 5i has a potentially confusing hyphen "We observed intravasation of CSC, identified by expression of the SORE6-GFP biosensor..." When landing on the margin, the text appears to read "SORE6- [i.e. negative] GFP "biosensor." Perhaps the authors can write SORE6+ biosensor instead?

In Figure 7F, does either SOX2 or OCT4 co

Reviewer #1 (Remarks to the Author):

The manuscript by Sharma and colleagues uses elegant intravital microscopy and cell tracing techniques to describe macrophage-induced phenotypic conversion of model breast cancer cells into cancer stem cells (CSC), the association of CSC at TMEMs and in CTC as well as their role in establishing lung metastases. A Jagged-Notch signal is proposed as non-cell autonomous pathway mediating the effects of macrophages on CSC generation. Examination of patient-derived FNA cells and corresponding surgical specimens yielded results consistent with the model proposed by the authors.

The manuscript is well-written, innovative and generally strong. Addressing the following issues will further strengthen the manuscript.

1. Figure 2Di and Dii: the rightmost panel in Figure Di (01:34:30) is identical to the middle panel in Figure Dii. Is this intentional? The Figure legend is unclear.

RE: Our intention to show the protrusion of invadopodia in all directions (2Di) and towards blood vessels (2Dii) separately may be confusing for some readers. Therefore, we combined Fig. 2Di and 2Dii into a new single Fig. panel (2D) in the revised manuscript. This Figure now summarizes our finding that invadopodia protrude in all directions. We modified the legend to reflect this change (page 44, lines 1405-1410).

2. Figure 4H: the experiments, as described in Materials and Methods, consisted of adding recombinant Jagged1 or a scrambled peptide to cell cultures. It is unclear whether Jagged1 was added in solution or used to coat the surface of wells prior to cell addition. This is important as Notch receptor activation requires subunit separation, which is achieved more easily when ligands are associated with the cell membrane of a signaling cell or bound to a solid surface.

RE: Recombinant Jagged1 or scrambled peptides were added in solution. We have added this information in the Materials and Methods on page 36, line 1112-1114 in the revised manuscript.

3. Figure 4I: did the macrophages used for co-culture express Jagged1? What happens if Jagged1 is knocked down in the macrophages? Is Jagged1 the only Notch ligand capable of inducing stemness? This is a distinct possibility because the kinetics of Notch activation by different ligands are different. However, it should be tested experimentally. Endothelial cells can also express Jagged1 and they do express DLL4. Since HUVEC do not induce stemness (Figure 4D), it is possible that another signal or a different Notch ligand (e.g. DLL1) is involved.

RE: We evaluated the macrophages (BAC1.2F5), which we used in our co-culture experiments, for Notch ligands expression and found that these macrophages primarily express Jag1 and Jag2, and small amounts of DLL1, DLL3, and DLL4 (New data shown in supplemental Fig S6H in the revised manuscript). The HUVEC express Jag1, DLL1, DLL4 as well as Notch1, 3 and 4, but the co-culture of HUVEC with cancer cells did not induce stemness in cancer cells. Therefore, it is unlikely that either DLL4 or DLL1 are involved in the induction of stemness in cancer cells via Notch. Since BAC1.2F5 express very low levels of DLL3 we reasoned that the most likely Notch ligands on macrophages involved in the induction of stemness are Jag1 and Jag2. Since Jag1KO BAC1.2F5 macrophages were available (1), we performed tumor cell-macrophage co-culture experiments with Jag1KO macrophages and found that Jag1 deletion from macrophages leads to complete inhibition of CSC induction caused by macrophages. This new data has been included as Fig 5F in the revised manuscript and described on page 13, lines 389-391. Moreover, we tested the effect of Jag1 and Jag2 blocking antibodies on macrophage-mediated induction of stemness in macrophage cancer cell co-culture experiments. These additional data show that both Jag1 and Jag2 are involved in macrophage-mediated induction of stemness in cancer cells and that blocking either Jag1 or Jag2, or Jag1 and Jag2 together leads to complete inhibition of CSC induction caused by macrophages, suggesting that DLL ligands play no role in macrophage-contact mediated CSC induction. These new data are presented in Fig 5G and described on page 13, lines 391-396, in the revised manuscript.

4. Figure 4J: the effectiveness and specificity of the Notch1 siRNA should be demonstrated by Western blotting.

RE: The western blot data are included in the revised manuscript as Fig 5C to demonstrate the effectiveness and specificity of Notch1 siRNA.

5. Figure 4I: the use of DAPT is a good indication of Notch involvement but it is not conclusive proof. Gamma secretase has >95 substrates including the Notch receptors. An orthogonal method of Notch inhibition (e.g., dominant negative MAML1, a cell-penetrating MAML1 decoy peptide should be used.

RE: As suggested by the reviewer, we used a small molecule inhibitor, SAHM1, which binds Notch transcription factor complex competitively with MAML1, and inhibits Notch signaling (2). We repeated our tumor cell – macrophage co-culture experiments in the presence of SAHM1 or DMSO control and found that Notch transcription factor complex inhibition leads to decrease in SORE6+ CSCs. These new data are added as Fig. 5B and described on page 12, lines 370-373 in the revised manuscript.

6. Similarly, for Figures 4K and J, proof of target inhibition by DAPT should be presented (e.g., staining for cleaved Notch1 or decreased expression of Notch transcriptional targets such as NRARP, Myc, or Notch3 should be presented)

RE: As suggested by the reviewer, we stained sections of breast tumors from DAPT or control treated PyMT mice for cleaved Notch1 (NICD1) and Hes1, a downstream Notch1 transcriptional target gene. We found dramatic decreases in both NICD1 and Hes1 staining in DAPT treated tissues compared to the control. Likewise, significant reductions in nuclear NICD1 and Hes1 were observed in DAPT treated tissues compared to the control. These new data are presented in Fig S6 panels J-M and described on page 13, lines 398-400 and page 37, lines 1163-1166 in the revised manuscript. Together with the previous data on the observed goblet cell hyperplasia in the intestinal crypts (Fig S6I), this new data confirms successful Notch signaling inhibition in vivo after DAPT treatment in mice.

7. Figure 7G: Is Jagged1 detectable on the surface of macrophages in contact with MENA+ cells in the context of TMEMs in human tissue? Are the MENA+ cells in contact with macrophages Notch1-positive? This would be an important confirmation that the mechanism identified in the MDA-MB231 and PyMT models operates in human tumors. As investigational monoclonal antibodies that inhibit Jagged1 exist, data supporting the existence of the pathway discovered by the authors in human tumors would significantly increase the translational relevance of their manuscript.

RE: Since human breast cancers are very heterogeneous (**Fig. 8A & C**), the meaningful evaluation of the spatial relationship of Notch ligands in macrophages and Mena isoforms in cancer cells in would require extensive evaluation in many patient samples, many of which are already exhausted. Therefore, these relationships are being studied in detail in another cohort and in a context of another manuscript.

Reviewer #2 (Remarks to the Author):

This is a timely and important subject with clear relevance to cancer pathogenesis. Using innovative intravital microscopy to obtain very high-quality in vivo images of the primary and metastatic tumour sites, the authors investigated the phenotype of individual CSCs during tumour dissemination and seeding. Using a cancer stem cell (CSC) reporter they propose that CSC intravasate more readily circulate and form metastases. They propose that the CSC phenotype is driven by macrophage induced Notch signaling.

The manuscript as a whole is fairly clear, although some important questions arise from the data presented that need to be addressed (see below).

Major Comments

1/ Based on imaging in fixed tissue and IVM, the authors propose that macrophage contact with cancer cells results in increased reporter signal, indicating development of a CSC phenotype. As it is hard to control for the cells involved (and to some extent the z resolution probably doesn't help) they used a reductionist co-culture of tumour cells and macrophages in vitro to show this in more detail. However, the representative movie (Movie 11) doesn't seem to support the author's interpretation. At 00:40 the tumour cell on the left is already brightly green fluorescent, before the macrophage contact depicted in the stills Figure S6B at 2:50. Is this movie representative? How often was there a clear increase in GFP fluorescence after macrophage contact with a formerly GFP-ve tumour cell in this experiment?

RE: We understand reviewer's confusion and have now replaced this movie with a better representative movie 12, showing the induction of stem cell phenotype upon macrophage cancer cell contact. The time-lapse panels from the new movie 12 were added as new Fig S6B in the revised manuscript. The detailed evaluation of the conversion of cancer non-stem (SORE6-) to cancer stem (SORE6+) cells is shown in Figure 4F.

2/ That the inhibition of Notch signaling presented in Figure 6 reduces the number of CSCs in the lung is intriguing. Importantly does this reduction in CSCs lead to a reduction in metastasis and an increase in survival?

RE: We evaluated the number of metastases in the lungs of control or DAPT treated SCID mice bearing SORE6>GFP MDA-MB-231 tumors. We found significant decrease in the number of lung metastases in DAPT treated mice compared to the control treated mice. This new data is shown in Fig 7K and described on page 17, lines 520-521 and page 40, lines 1241-1246 in the revised manuscript. An example of lung metastasis is shown in Fig S7 panel C of the revised manuscript. Survival studies in mice require extensive amount of time and funds to complete which prohibits us from including them in the current manuscript.

Other comments

Figure 2.

The authors show that non-CSCs have a high-speed motile migration phenotype, whilst CSCs have a more slow-moving migratory phenotype. However, in Movie 7 the non-CSC cells barely move compared with Movie 6, is there a difference in their motility when they are close to the blood vessels compared with being further away?

RE: Tumor cells are heterogeneous in their motility, with the majority of tumor cells being non-motile (3-5). Among the motile cells, we saw high-speed migration phenotype in non-CSCs. We know from previous work that migratory tumor cells use aligned collagen fibers for high-speed migration toward the blood vessel (3, 5-9). Since the tumor microenvironment near blood vessels is different than further away (including lower collagen fiber alignment and density), we have observed that fast migratory non-CSCs slow down as they approach the blood vessel. The effect of tumor microenvironment on tumor cell speed switch from fast to slow as cells approach blood vessels was also described in (5). We have added this information on page 7, lines 204-205 and page 20, lines 608-613.

2G: only shows the Tks5 and Cortactin staining in CSCs, what do these look like in non-CSCs?

RE: As requested by the reviewer, we have included an example of Tks5 and Cortactin staining in non-CSCs

in new supplemental Fig. S4J in the revised manuscript. This does not affect the invadopodia quantification shown in Fig. 2J.

Figure 3.

60-70% of the CSCs in live tumours are in contact with macrophages. How long do the CSCs and macrophages stay in contact? What is the percentage of non-CSC cancer cells in contact with macrophages?

RE: The contact between macrophage and cancer cell lasts approximately 0.5-1.5 hours before cancer cell activate SORE6 reporter (Figs. 4A, 4B). As per reviewer's request, we performed quantification of non-CSCs in contact with macrophage and find that only about 30% of non-CSCs are in contact with a macrophage. This new data is added to the CSC data in Fig 3C and described on page 9, lines 262-263, in the revised manuscript. We have moved the Fixed tissue CSC quantification into supplemental Fig S5A.

3B: appears to show some macrophages which haven't been counted in the field of view.

RE: We did not count all the macrophages in the field of view in Fig 3B. Rather, we evaluated all the CSCs (SORE6+ green cells) in the field of view for direct macrophage contact (white oval) or not (white square). In Fig 3B, we did notice that the square outline of one SORE6+ cell was missing, even though this cell was included in the quantification in Fig S5A. We fixed it in the updated Fig 3B in the revised manuscript.

3D: how many mice underwent clodronate, how was the analysis was done?

RE: Three mice each for control and clodronate treatments were evaluated. We have added this information in the legends for Figs. 3D, page 45, lines 1463-1464 and 3E, page 46, lines 1470-1471. Analysis for macrophage and SORE6+ cells is described in the materials and methods section under heading "Macrophage depletion using Clodronate" on page 33. We are also providing the non-normalized data for SORE6+ cell counts in these experiments in new supplemental Fig S5B in the revised manuscript.

3G: A supplemental Figure showing the IHC for macrophage depletion would be helpful.

RE: As requested, we added a supplementary Figure S5F showing IHC visualization of macrophages in the revised manuscript.

Figure 4.

4J: The authors show that CSCs which have inhibited Notch signaling incubated with macrophages have reduced CSC conversion compared with controls. This could be strengthened by timelapse imaging. Does Notch signaling affect the motility of the cancer cells and therefore their interaction with macrophages?

RE: Inhibiting Notch signaling does not affect the motility of cancer cells in the absence of macrophages as shown previously (Fig 6D in (10)). We have added this information on pages 12-13, lines 376-379 in the revised manuscript.

Figure 6

6F: How long after tumour implantation was the intravital imaging done?

RE: The stem reporter cells tdTomato MDA-MB-231 SORE6>GFP or the reporter control cells tdTomato MDA-MB-231 minCMV>GFP were injected into the mammary fat pad of SCID mice (NCI). In approximately 6 weeks, these mice developed primary tumors of size ~1 cm in diameter. After which, mice underwent placement of permanent lung imaging window and Intravital imaging was performed the following two days. We have added this information into Materials and Methods section under Live Lung Imaging, on page 39, lines 1219-1223 in the revised manuscript.

It is understood that space is limited, however, a clearer statement of the tumour models being presented in each Figure legend would be helpful in interpreting the data. Additionally, the number of mice is not clear in all of the Figure legends.

RE: We added the description of tumor models and number of mice into each Figure legend.

Reviewer #3 (Remarks to the Author):

In this study, Sharma et al. employ intravital imaging to provide an important and interesting study on the nexus between cancer stem cells, macrophages, and key steps in metastatic dissemination. The manuscript is generally well written, and the data are for the most part clearly presented. In essence, the authors use a biosensor for SOX2/OCT4 stem cell transcription factors in cancer cells and show that its activity correlates with tumor-initiating behaviors, and, furthermore, that this malignant cell state is enhanced when cancer cells are brought into close proximity with macrophages near endothelial networks in vivo, at sites referred to as the Tumor Microenvironment of Metastasis. A few experiments hint at a potential mechanism involving the induction of Notch signaling in cancer cells by way of Jagged1 release from macrophages, and several experiments also implicate MENA-invasive in this process as part of the body of literature that surrounds Dr. Oktay's research. The MDA-MB-231 cell line serves as the main workhorse of the study, although a few key findings are loosely corroborated in other cell lines and some correlative data is included from an analysis of patient samples.

Intravital imaging is a challenging technique to perform, and to recommend additional intravital imaging experiments seems excessive from the standpoint of this reviewer. However, it is important to recognize that claims which lean on intravital imaging can sometimes be disadvantaged by low sample sizes and, as such, many additional experiments could be performed in vitro to lend crucial support for the author's major claims, especially those that surround intravasation events. There is also notable concern about the specificity of the biosensor system used in the study; at the very least, this must be clearly addressed in the text. Therefore, in the opinion of this reviewer, the study by Sharm et al. could be appropriate for publication in Nature Communications only if the major issues outlined below are adequately addressed prior to resubmission. If a majority of the points below could be successfully tackled or convincingly refuted during a major revision, then this study would be worth reconsidering for publication in Nature Communications.

Major Issues Needing to be Addressed:

Figure 1:

There is some concern about the specificity of the SORE6 biosensor for SOX2/OCT4. A previous paper from the Wakefield group (Tang et al., Stem Cell Reports 2015) introduces the SORE6 sensor. The supplemental data for that paper provides the sequence for the SORE6 minigene and indicates the SOX2/OCT4 response elements embedded throughout. The sequence for the minigene is as follows:

```
5'atctatcgatcagctacttttgcattacaatggccttggtgcagctacttttgcattacaatggccttggtgcagctacttttgcattaca  
tggccttggtggaattccagctacttttgcattacaatggccttggtgcagctacttttgcattacaatggccttggtgcagctacttttgc  
ttacaatggccttggtgactagttcta-3'
```

Entering this sequence into an online transcription factor binding site prediction database like PROMO reveals numerous putative binding sites for various other transcription factors including XBP-1, c-JUN, CEBPA. How confident can the reader be that the SORE6 sensor is specific to SOX2/OCT4? Some of these other transcription factors are associated with cell stress, so one wonders about the extent to which the SORE6 biosensor could be a cell stress sensor; many of the phenotypes presented herein could be explained by increased cancer cell stress, where the acquisition of stem-like (or the aggressive) features could accompany a stress response. The fact that the authors report SORE6 biosensor fluorescence without robust SOX2/OCT4 expression (Fig. S1E) lends credence to the idea that this system is promiscuous. To partially address this, for example, it could be useful to assess the expression of notable target genes for the aforementioned cell stress transcription factors in Figure S1E and in Figure S5B.

RE: Different prediction algorithms return different results. For example, the HOMER tool returns a set of predicted TF binding sites that is essentially non-overlapping with those identified by PROMO. However, to

address the reviewer's legitimate concerns that the SORE6 sensor might be responding to transcription factors other than SOX2 and OCT4, we performed siRNA knockdown in the MDA-MB-231 cells and showed that knockdown of Oct4 reduces the number of SORE6+ cells by 95%, while knockdown of SOX2 or SOX9 reduces them by 28%. Thus, Oct4 is the major driver of the reporter in this cell line, with other transcription factors contributing less than 5% of the signal. This data has been added as new Fig S1F and described in the revised manuscript on page 6, lines 164-166. We further showed that expression of stress-related transcription factors is not different (CEBPa, cJUN) or is reduced (XBP1) in SORE6+ cells making it highly unlikely that these transcription factors contribute significantly to SORE6 reporter expression. These data are provided below for reviewer's use only. Nanog is the positive control for this experiment.

The Stem Cell Reports article in 2015 Tang et al. also appears to address the potential promiscuity of the SORE6 system by stating that the destabilized GFP boosts its specificity for CSCs, but these experiments are performed in the MCF-7 cell line and not in MDA-MB-231 cells. Conducting additional work in vitro to show sensor specificity for SOX2/OCT4 or, if that is not feasible, evaluating the status of additional stem cell markers is strongly recommended. It might also be important to acknowledge the potential caveats of this system clearly in the text. Performing additional supplemental experiments in vitro to establish whether SORE6 activation correlates with the acquisition of additional stem cell markers (e.g. wnt/notch pathway activation, CD44, CD133, ALDH1, etc.) or additional phenotypes (e.g. colony formation on soft-agar) is still another option.

RE: We have included new data showing a strong overlap between CD133 positivity and SORE6 positivity in the MDA-MB-231 model (new supplementary Fig S1G). Additionally, using the Aldefluor assay we showed highly significant overlap between ALDH1 activity and SORE6 positivity in the MDA-MB-231 (new supplementary Fig S1H). These new results are described on page 6, lines 166-169 and page 31, lines 947-961. We also show that SORE6+ cell form significantly more tumorspheres than SORE6- cells (new supplementary Fig S1D). These new results are described on page 6, line 158. Taken together with the demonstration (new Suppl Fig S1F) that the SORE6 reporter expression is absolutely dependent on the presence of Oct4 in the MDA-MB-231 cell line, and all the additional validation data elsewhere in Suppl. Fig S1, we believe that we have very rigorously re-validated the sensor for this cell line using a wide range of orthogonal approaches.

Figure 1G from the Tang et al. paper also shows a baseline SORE6 positivity of around 40% for the MDA-MB-231 cell line in vitro, whereas the current Sharma et al. paper uses tumors in mice (e.g. Fig. 1D & S1F) to make the case that SORE6+ cells are a much smaller minority population generally consisting of less than 5% (Fig. 1D). Knowing the baseline of SORE6+ MDA-MB-231 cells in vitro is critical if the reader is to interpret key experiments presented later on, such as the co-culture data shown in Figure 4.

The baseline SORE6 positivity for the MDA-MB-231 subline used in the current study varies from ~15-40% in vitro, depending on the culture conditions. We have added this information in the "In vitro tumor cell-macrophage co-culture assay" methods section on page 35 lines 1101-1102. It is always our experience that the % SORE6+ cells in an *in vitro* culture is much higher than the % SORE6+ cells in a tumor in vivo. We believe that this is because the tumor cells follow a more steeply hierarchical model in the primary tumor in vivo (generally less plasticity) than in vitro. Below is a graph comparing the % SORE6+ cells in vitro (assessed

by flow cytometry immediately prior to implantation) and in vivo (assessed by flow cytometry on digested primary tumors once they reached ~1cm diameter, after staining with LIN-APC antibody to distinguish stromal from tumor cells) across the 3 breast cancer cell lines (MDA-MB-231, MCF7, MCF10Ca1h) to illustrate this point. However, this graph will be published as a part of another study and is presented here for reviewers only.

Figure 2:

In Figure 2 (Fig. 2c-d; S4c), two intravital imaging videos are provided that compare membrane protrusions emanating from SORE6+ vs SORE6- cells. Although the authors write that both cell types are located in close proximity to the blood vessel, the videos depict that the SORE6+ example cell is touching the blood vessel whereas the SORE6- is more distal. One wonders if a better controlled comparison could be provided for the reader? Perhaps at a minimum, a video for a SORE6+ cancer cell that is further away from a blood vessel would be useful?

RE: As suggested by the reviewer, we have now added a new movie #6 for a SORE6+ CSC that is further away from the blood vessel. Similar to SORE6+ CSC touching the blood vessel, the SORE6+ CSC away from the blood vessel also displays dynamic invadopodia protrusions coming out in all directions, possibly involved in degrading the surrounding ECM as shown in Fig 2C and Movie 3. Time-lapse panels from the new Movie 6 were added in Fig S4B and described on page 8, lines 219-222 in the revised manuscript.

Moreover, the invadopodia frequency data (Fig. 2c-f), while interesting, lacks robust quantification in its current form.

RE: We quantified invadopodia protrusions in vivo in SORE6- and SORE6+ cells and found that compared to non-CSCs, CSCs display on average 6 protrusions per hour (i.e. protrusion period of ~10 min), in agreement with previously published values (11, 12). This new data was included as Fig 2G and described on page 8, lines 226-228 in the revised manuscript.

Figure 3:

Figure 3d-e purports to causally link a CSC phenotype to macrophages in vivo by administering clodronate to deplete macrophages in mice and by showing this correlates with a decrease in the amount of SORE6+ cells. Here it would be helpful to have access to the non-normalized data, so the reader can assess the total cell counts for each day.

RE: As requested, we have now provided the non-normalized data for the CSC counts for each day in the Supplementary Fig. S5B in the revised manuscript.

In addition, knowing what happens to SORE- cells during these timepoints would allow the reader to evaluate whether the SORE6+ cancer cells simply die, or if SORE6+ cells convert back into SORE6- phenotype during macrophage depletion. If the latter occurs, it would be highly interesting to the reader. If not, then some transparency is warranted.

RE: We have no evidence from the short-term (30 hours) fate mapping experiments in vitro that the SORE6+ cells die following loss of macrophage contact (Fig. 4F).

Another important experiment may entail determining whether clodronate treatment impacts SORE6+ cancer cells in vitro, to lend support for the effect occurring through macrophages or even other microenvironmental cell types in vivo. Experiments like these would help one better judge whether the link is indeed casual like the authors claim.

RE: We cultured SORE6+ MDA-MB-231 cells with and without clodronate in vitro and saw no direct effect of clodronate on the number of SORE6+ cells. This new data has now been added as Suppl. Fig. S5C and described on page 9, lines 271-274 and page 33, lines 1021-1024 in the revised manuscript.

Figure 4:

In Fig. 4E and S6D, the authors co-culture macrophages with MDA-MB-231-LM2 and 4T1 cells, and these cell lines are here first introduced into the story. The authors must better validate a link between SORE6 biosensor and stem-like features in these cell lines. At the very least, the authors should perform RT-qPCR to ascertain the expression of various stem cell genes for SORE6+/- cells in both lines in the manner depicted in Fig. S1E. Also, what is the baseline amount of SORE6+ cells in each cell line prior to the co-culture experiments?

RE: The baseline % SORE6+ cells for 4T1 and MDA-MB-231-LM2 cell lines are 3.3 +/- 0.4% and 21.2 +/- 0.5%. We have added this information in the methods section on page 36, lines 1119-1120, 1125. To validate the link between SORE6 biosensor activity and stemness for the MDAMB231-LM2 and 4T1 cells, we performed in vivo limiting dilution data (the gold-standard functional CSC assay) and the results are presented below. Since these data are part of another manuscript in preparation, we can only use them in this rebuttal as an answer to this reviewer's question.

MDA-MB-231-LM2

Initial cell inoculum (# cells/site)		Tumor incidence at day 64 (#tumors/# sites)			CSC frequency	Fold enrichment of CSCs in SORE6+ population	Chisq p-value for difference in frequency
		2000	500	100			
Cells implanted	SORE6+	1/6	4/6	2/4	1 in 1868	7.8	0.017
	SORE6-	1/6	0/6	0/6	1 in 14577		

4T1

Initial cell inoculum (# cells/site)		Tumor incidence at day 21 (#tumors/# sites)			CSC frequency	Fold enrichment of CSCs in SORE6+ population	Chisq p-value for difference in frequency
		10,000	1000	200			
Cells implanted	SORE6+	10/10	10/10	7/10	1 in 163	16.8	8.70E-09
	SORE6-	9/10	6/10	0/10	1 in 2737		

The experiments conducted in Fig. 4H-J are crucial for establishing a mechanism. In its current form, however, this reviewer is not convinced that Jagged1 release by the macrophages activates a tumor-initiating phenotype in the cancer cells. Have the authors attempted to perturb the Jagged1 gene in the bac1.2f5 macrophage cell line, thereafter testing if this leads to a reduction in SORE6+ cancer cells after co-culture? If this is impractical, could the authors use Jagged1 antibodies to neutralize this factor in cell culture medium? Without key mechanistic experiments along this line, it is very difficult to accept that a Jag1-Notch axis explains the phenotype, given the small effect sizes depicted in Figure 4.

RE: As suggested by the reviewer, we ran tumor cell-macrophage co-culture experiments with Jag1KO BAC1.2F5 cells (1) and found a significant reduction in SORE6+ CSCs after co-culturing with Jag1KO macrophages compared to the control macrophages. This new data has been included as Fig 5F and described on page 13, lines 389-391 in the revised manuscript. In addition, we followed reviewer's advice and performed co-culture experiments with Jag1 and Jag2 blocking antibodies. These additional data show that both Jag1 and Jag2 are involved in macrophage-mediated induction of stemness in cancer cells. These new data are presented in Fig 5G and described on page 13, lines 391-396 in the revised manuscript.

Figure 5/Figure 6:

The authors would greatly benefit from additional evidence indicating whether SORE6+ cells actually intravasate to a greater degree than SORE6- cells, as their enrichment in TMEM, their association with blood vessels, and their increased numbers in circulation are all circumstantial. At present, an intravital imaging movie of one intravasating SORE6+ cell is not enough to convince this reviewer that this cell state directly enhances this aspect of metastatic dissemination. Quantifying additional cells via intravital imaging may be too onerous, although the authors might consider further investigating the transendothelial migration potential of SORE6+ vs SORE6- cells in vitro using the Transwell platform or other systems, as a variety of assay types currently exist on the market. Microfluidic devices to assess vascular invasion represent yet another option. The manuscript uses the "doorway" and "portal" verbiage extensively, despite this aspect of the SORE6+ phenotype is not yet being convincingly shown. The large increase in SORE6+ cells in circulation (Fig. 6B) is truly fascinating but it may not necessarily be explained by TMEM, as it has not yet been established if SORE6+ cells intravasate to a higher degree of if SORE6- cells; SORE6- cells may transform into SORE6+ in circulation.

RE: We provided new data which indicate that the % of SORE6+ cells within unsorted MDA-MB-231 cells injected into the tail-vein remains unchanged after we recovered them 6 hours later from the lungs, indicating that there has been neither die-off of SORE6- cells while in the circulation, nor significant conversion of SORE6- to SORE6+. These new data are provided in supplementary Figure S7B and described on page 15, lines 461-467 and page 40, lines 1248-1257.

The authors propose a mechanism whereby macrophages secrete Jagged1 to induce a CSC phenotype in metastasizing cancer cells. It seems important to address whether the administration of DAPT goes beyond reducing the number of SORE6+ cells (Fig. 6J) to also impact the degree of metastatic burden. An assessment of metastatic lesions in the mouse lung should be feasible for this experiment, given that mice were treated daily for 14 days prior to the collection of tissues for histology.

RE: As suggested by the reviewer, we assessed the number of metastatic foci in the mouse lung tissues treated with control or DAPT and found that the metastatic burden (number of lung metastases) is significantly decreased after DAPT treated mice. This new data is presented in Fig 7K and described on page 17, lines 520-521 and page 40, lines 1241-1246 in the revised manuscript.

Figure 7:

In Figure 7D-E, the authors evaluate the following marker combinations to detect CSCs in human patient samples: CD44+/CD24-, ALDH+, and CD133+. They subsequently show that the expression of

these markers correlates with the TMEM score. The current study would be significantly more cohesive if the authors assessed the status of some these markers (where applicable) in the cell lines being studied. For example, do SORE6+ MDA-MB-231 cells also harbor evaluated levels of CD44 or CD133?

RE: As requested by the reviewer, we have included new data showing a strong overlap between CD133+ and SORE6+ in the MDA-MB-231 model. Additionally, using Aldefluor assay we find significant overlap between ALDH1 and SORE6+ in the MDA-MB-231. These new data are presented in supplementary Figs S1G and S1H and described in the revised manuscript on page 6, lines 166-169 and page 31, lines 947-961. As has also been shown by others (13), we find the CD44 to be present on >95% of MDA-MB-231 cells, so it is not a good marker of stemness in this cell line.

Minor Issues to Consider:

In their introduction, the authors outline challenges associated with modeling stable CSC phenotypes in vitro, arguing that such phenotypes can “depend on signals from the tumor microenvironment.” There is obviously validity to this notion and putting one’s work in context is important, but these statements seem at odds with the many in vitro experiments embedded throughout the ensuing manuscript. At present, the introduction also reads like technical justification for using intravital imaging and the SORE6 biosensor, as opposed to covering topics that will aid its readers in understanding the biology. It is recommended that that the introduction be expanded or edited to incorporate more content on macrophages and Notch signaling, and to explain how they relate to the phenomenon being studied.

RE: As per reviewer’s suggestion, we have re-written the Introduction on pages 3-4 in the revised manuscript.

For Fig. S1E: It would be useful to sort out GFP+ cells from the minCMV>GFP control to assess whether GFP expression by itself leads to the increased expression of stem cell genes. This would further boost confidence about the link between SORE6+ and stem-like features in cancer cells.

RE: By definition, the number of GFP positive cells in the minCMV>GFP control is extremely small (<0.1%) so the question cannot readily be addressed by this approach as sorting sufficient cells for analysis would be technically extremely challenging. To approach the question in a different way, we asked whether overexpression of GFP in the cell line led to increased expression of stem cell transcription factors, and we find that it does not. The data are given for the reviewer below. Thus we do not believe that GFP expression induces a stem phenotype.

MDA-MB-231-tdTom cells were transduced with control lentivirus (CON) or EF1>GFP (GFP) from Sartorius Inc, (Cat #4624) and briefly selected with puromycin for constitutive high expression of GFP. Nanog and Oct4 mRNA expression were assessed by QRT-PCR and normalized to the control condition. Results are mean +/- SD (n=3), t-test

In Fig. 3C, the labels for “Mac contact” do not align perfectly with the bars.

RE: CSC Data in Fig 3C has been replotted with additional new data for non-CSCs. The fixed tissue data in the

original Fig 3C was moved to Fig S5A and the labels for “Mac contact” have been aligned there in Fig S5A in the revised manuscript.

In Fig. S5B, can the authors please expand their assessment of gene expression in Met-1 cells to also include the genes also examined in the MDA-MB-231 cell line? This would allow the reader to determine the degree of concordance between the models and whether the sensor functioning as described in that line. Of particular interest is whether Met-1 cells express Oct-4.

RE: As requested by the reviewer, in addition to Sox9, we have now expanded the assessment of gene expression in Met-1 cells to Oct4 and Nanog. This new data is presented in supplemental Fig S5E and described on page 10, lines 283-286 and pages 29-30, lines 874-900.

For Fig. 3G-H, can the graph or Figure legend depict at what timepoint these data were collected? This would help the reader relate the results back to Figure 3D-E.

RE: The timepoint of data collection was included in the Figure legends 3G-H and in the Materials and Methods section on page 33, lines 1013-1014 in the revised manuscript.

In Fig. 4D, what is the purpose of “Mac ◊ TC ◊ Notch ◊ CSC” above the graph? It is assumed this is a typo on the Figure.

RE: The purpose of the Mac → TC → Notch → CSC above the graph was to emphasize the sequence of cell contact and pathway induction event preceding induction of cancer stem cells. Since this seems to be confusing, we decided to take it out.

In Fig. S1G (or S1E), what is the baseline expression of NOTCH1 in SORE6+ vs SORE6- cells before the introduction of macrophages? This information seems important if the reader is to attribute its induction in cancer cells to macrophages, as the narrative for Figure 4 suggests.

RE: As requested by the reviewer, we evaluated the baseline expression of NOTCH1 in SORE6+ and SORE6- cells and found no difference in NOTCH1 expression between SORE6+ and SORE6- cells. These new data were added in Fig S6G and described on page 13, line 379-382 in the revised manuscript.

In Fig. 5A: Should the Figure legend on the image indicate labels for all colors? What do the authors make of the green SORE6+ cells that are ostensibly negative for tdTomato?

RE: We modified Figure 6A to indicate that CSC can appear as green or yellow depending on the expression level of volume marker tdTomato.

The description for Fig. 5i has a potentially confusing hyphen “We observed intravasation of CSC, identified by expression of the SORE6-GFP biosensor...” When landing on the margin, the text appears to read “SORE6- [i.e. negative] GFP “biosensor.” Perhaps the authors can write SORE6+ biosensor instead?

RE: As requested, we modified SORE6-GFP to SORE6+ to avoid the confusion.

In Figure 7F, does either SOX2 or OCT4 correlate with the TMEM score in human patient samples?”

RE: This analysis was not performed because many samples were exhausted after extensive analysis for other stem cell markers.

REFERENCES

1. Cabrera RM, Mao SPH, Surve CR, Condeelis JS, Segall JE. (2018). A novel neuregulin - jagged1 paracrine loop in breast cancer transendothelial migration. **Breast cancer research : BCR**. 20(1):24. PMID: 29636067 / PMCID: PMC5894135.
2. Moellering RE, Cornejo M, Davis TN, Del Bianco C, Aster JC, Blacklow SC, Kung AL, Gilliland DG, Verdine GL, Bradner JE. (2009). Direct inhibition of the NOTCH transcription factor complex. **Nature**. 462(7270):182-8. PMID: 19907488 / PMCID: PMCPMC2951323.
3. Sidani M, Wyckoff J, Xue C, Segall JE, Condeelis J. (2006). Probing the microenvironment of mammary tumors using multiphoton microscopy. **J Mammary Gland Biol Neoplasia**. 11(2):151-63. PMID: 17106644.
4. Patsialou A, Bravo-Cordero JJ, Wang Y, Entenberg D, Liu H, Clarke M, Condeelis JS. (2013). Intravital multiphoton imaging reveals multicellular streaming as a crucial component of in vivo cell migration in human breast tumors. **Intravital**. 2(2):e25294. PMID: 25013744 / PMCID: PMC3908591.
5. Gligorijevic B, Bergman A, Condeelis J. (2014). Multiparametric classification links tumor microenvironments with tumor cell phenotype. **PLoS Biol**. 12(11):e1001995. PMID: 25386698 / PMCID: PMC4227649
6. Wyckoff JB, Wang Y, Lin EY, Li JF, Goswami S, Stanley ER, Segall JE, Pollard JW, Condeelis J. (2007). Direct visualization of macrophage-assisted tumor cell intravasation in mammary tumors. **Cancer Res**. 67(6):2649-56. PMID: 17363585.
7. Xue C, Wyckoff J, Liang F, Sidani M, Violini S, Tsai KL, Zhang ZY, Sahai E, Condeelis J, Segall JE. (2006). Epidermal growth factor receptor overexpression results in increased tumor cell motility in vivo coordinately with enhanced intravasation and metastasis. **Cancer Res**. 66(1):192-7. PMID: 16397232.
8. Roussos ET, Condeelis JS, Patsialou A. (2011). Chemotaxis in cancer. **Nature reviews Cancer**. 11(8):573-87. PMID: 21779009 / PMCID: PMC4030706.
9. Leung E, Xue A, Wang Y, Rougerie P, Sharma VP, Eddy R, Cox D, Condeelis J. (2017). Blood vessel endothelium-directed tumor cell streaming in breast tumors requires the HGF/C-Met signaling pathway. **Oncogene**. 36(19):2680-92. PMID: 27893712 / PMCID: PMCPMC5426963.
10. Pignatelli J, Bravo-Cordero JJ, Roh-Johnson M, Gandhi SJ, Wang Y, Chen X, Eddy RJ, Xue A, Singer RH, Hodgson L, Oktay MH, Condeelis JS. (2016). Macrophage-dependent tumor cell transendothelial migration is mediated by Notch1/MenaINV-initiated invadopodium formation. **Sci Rep**. 6:37874. PMID: 27901093 / PMCID: PMC5129016
11. Sharma VP, Eddy R, Entenberg D, Kai M, Gertler FB, Condeelis J. (2013). Tks5 and SHIP2 regulate invadopodium maturation, but not initiation, in breast carcinoma cells. **Curr Biol**. 23(21):2079-89. PMID: 24206842 / PMCID: PMC3882144.
12. Magalhaes MA, Larson DR, Mader CC, Bravo-Cordero JJ, Gil-Henn H, Oser M, Chen X, Koleske AJ, Condeelis J. (2011). Cortactin phosphorylation regulates cell invasion through a pH-dependent pathway. **J Cell Biol**. 195(5):903-20. PMID: 22105349 / PMCID: PMC3257566.
13. Fillmore CM, Kuperwasser C. (2008). Human breast cancer cell lines contain stem-like cells that self-renew, give rise to phenotypically diverse progeny and survive chemotherapy. **Breast cancer research : BCR**. 10(2):R25. PMID: 18366788 / PMCID: PMCPMC2397524.

Reviewers' Comments:

Reviewer #1:

Remarks to the Author:

The revised manuscript does address my most important concerns and been strengthened significantly. The revised manuscript would be of considerable interest for the readers of Nature Communications.

Reviewer #2:

Remarks to the Author:

In general, the authors have done a very good job addressing the concerns raised in the previous review. Their point re-survival data is well taken, and the metastasis data they provide add a lot to the manuscript.

However, the imaging data supporting the induction of the stem cell phenotype after macrophage contact in co-culture are still not clear.

The new movie (Movie 12) appears to show a consistently low level of GFP-expression in the reporter tumour cells prior to interaction with the macrophage, with what seems a very modest change in fluorescence intensity after interaction. How does this compare with the MinCMV-GFP ctrl used for thresholding, could these be shown?

The stills used in the figure (S6B) are slightly clearer than the movie, so some of this may be due movie compression and could benefit from the use of a 'glow' type look up table? However, as the graph in Fig. 4F depicts that nearly 1 in 4 macrophage:tumour cell interactions lead to the conversion to GFP+ it is surprising that a better example in terms of the clarity of signal change could not be found. There are two other, extremely bright, GFP+ tumour cells in this video alone that unfortunately we do not see the interaction history of.

Reviewer #4:

Remarks to the Author:

Employing a previously described Sox2/Oct4 reporter (SORE6) and intravital imaging authors track breast cancer cells with cancer stem cell (CSC) properties. They show that SORE6+ cells are enriched in stemness markers, have higher tumor-initiation activity, slow migratory phenotype and form more invadopodia compared to SORE6- cells. Moreover, SORE6+ cells are predominant in tumor microenvironment of metastasis (TMEM) doorways, among circulating tumor cells and at distant lung metastatic sites at the initiation of the metastatic growth. Importantly, induction of SORE6 reporter expression in breast cancer cells is, at least in part, attributed to Notch1 signaling upon interaction with Jag1/Jag2 ligands on macrophages. Finally, they describe that cancer cells expressing markers of stemness are also enriched in TMEMs of breast cancer patients where macrophages, endothelial cells and cancer cells co-localize. Therefore, suggesting that TMEMs are not only metastasis doorways where cancer cells intravasate but also where stemness properties are acquired by cancer cells by interacting with macrophages.

The manuscript is well written and generally convincing. While intravital imaging only provides descriptive observations ex vivo readouts and in vitro experiments support their nice observations. Nonetheless, there are a couple of statements that are not robustly proven by the data and some technical/data representation aspects that may need to be addressed:

- While authors observe increase stemness markers (CD133 and ALDH1) expression in SORE6+ vs SORE6- cells (as requested by Reviewer 3) there is still a big proportion of SORE6+ cells that do not express CD133 (Figure S1G) and it is not clear whether all SORE6+ cells have the capacity to form tumorspheres (Figure S1D). Moreover, in Figure S1B only 1 in 2974 SORE6+ cells have the capacity to initiate tumors in immunocompromised SCID mice and there is still a proportion of cells among SORE6- cells with CSC properties. Therefore, it would be more appropriate to say that SORE6+ cells are enriched in CSCs rather than stating that SORE6+ cells are bona-fide CSCs.
- Figure S1C: As SORE6+ and SORE6- cells were mixed from the initiation of the treatment it is not possible to distinguish whether SORE6+ cells are indeed resistant to paclitaxel or whether SORE6 reporter expression is enhanced overtime due to other factors (cell-cell contacts etc). In

fact, SORE6+ cell numbers decrease or barely grow in the first 20h whereas SORE6- cells grow in the same time frame. Cell growth +/- paclitaxel of sorted SORE6+ vs SORE6- cells should be assessed to confirm increased chemoresistance of SORE6+ cells.

- Figure 1. Authors provide new evidence to exclude that that stress related TFs (CEBPA, XBP-1 and c-Jun) are responsible for the SORE6 reporter expression as requested by Reviewer 3.

Moreover, they provide new data indicating that Oct4 is the main SORE6 reporter expression inducer (Figure S1F). Thus, it is not clear why authors use Sox9 instead of Oct4 to assess cancer cells stemness as an alternative to SORE6 reporter in the PyMT mammary primary tumor model (Figure 3H, 5H). Are Oct4+ cancer cells also decreased upon macrophage depletion and DAPT treatment? Moreover, authors define Sox9Hi cells as CSCs without an explanation of the criteria used to define high Sox9 expression, which is essential in immunofluorescence staining where technical factors can contribute to changes in staining intensity.

- As Reviewer 3 pointed out it is important to state that the frequency of SORE6+ cells in vitro is much higher than in vivo to properly interpret Figure 4 co-culture experiments. Thus, the initial frequencies of SORE6+ cells in vitro should be described in the results part in addition to material and methods section. Moreover, while normalizing SORE6+ cell frequencies to control samples makes the comparison with the experimental condition easier in Figures 4D and E, it does not show whether SORE6+ cell numbers change over time in culture, which is important to validate that macrophages are strictly necessary for SORE6 reporter induction. Could authors represent the frequencies of SORE6+ cells at the initiation of the co-culture and at the end-timepoint?

- To confirm that macrophages are inducing cancer cell stemness, the tumor initiation activity of cancer cells previously in contact with macrophages compared to those cultured alone could be assessed.

- The in vitro data clearly points out to the role of Notch1 signaling on cancer cells in the induction of SORE6 reporter expression by interaction with Jag1/Jag2 ligands on macrophages. However, the tumor microenvironment greatly differs from in vitro conditions and additional experiments would be needed to confirm that the same Notch1/Jagged signaling predominates the stemness acquisition of breast cancer cells in TMEMs in vivo. The current macrophage depletion with clodronate liposomes and Notch1 inhibition does not completely proof the cross-talk:

o Figure 3D-E. Even if in co-culture experiments (Fig 4F) SORE6+ cells do not die 30h following loss of macrophage contact, it is still not clear whether SORE6+ cells die, SORE6- cells cannot acquired SORE6+ phenotype or SORE6+ cells convert into SORE6- upon macrophage depletion in vivo. In vitro an in vivo conditions are too different to be able to extrapolate what happens to SORE6+ cells in vivo from in vitro observations. Could authors provide the non-normalized cell counts/frequencies of SORE6- cells upon clodronate treatment (as for SORE6+ cells in Figure S5B) to assess whether a SORE6- phenotype enrichment occurs or only SORE6+ cells die upon macrophage depletion in vivo?

o While authors attempt to address Reviewer 3's concern about the possible direct impact of clodronate on SORE6+ cancer cells, the provided data in Figure S5C is limited. Whether clodronate impacts the general cancer cell viability and growth (in SORE6+ and SORE6- cells) is missing. Would the same in vitro results be obtained when liposome clodronate instead of clodronate was administered?

o Figure 5H shows that Notch signaling is important for Sox9 expression on cancer cells but it does not confirm whether Notch interaction with Jag1 or Jag2 on macrophages is required. Could authors check whether in vivo blocking of Jag1 and Jag2 decreases the Sox9 expressing cancer cell frequencies?

o If authors find that the in vivo validation of the Notch1/Jagged signaling mediated macrophage-cancer cell crosstalk is out of the scope of the manuscript they may state it in the discussion.

- Figure 6I, while the images and video nicely show a SORE6+ cell intravasating in a TMEM, how do authors know that intravasation does not occur in other non TMEM vessels and by SORE6- cells? Intravasation events in non-TMEM vessels and by SORE6- cells would need to be monitored and quantified.

- Figure 7D-E: To confirm that MenaINV expression is found exclusively or predominantly in SORE6+ cells, quantification of MenaINV+ cell frequencies among SORE6- cells would be necessary.

- Figure 7F-G. To confirm that SORE6+ cells have increased capacity to metastasize into distant organs and exclude that the increased SORE6+ cell frequencies in the lung was not due to in situ acquisition of SORE6 expression by SORE6- CTCs after extravasation, i.v injection of SORE6+ vs SORE6- cells and analysis of SORE6 expression in the lungs could be performed (as in Figure S7B).

- Figure 8G: Which non-biased parameter was used to define a TMEM rich areas? Could authors exclude that the Sox9hi enrichment did not occur in non-TMEM vessel rich areas? Could be possible to assess the distance of Sox9hi cells from TMEMs vs non-TMEM vessels in patients tissues as performed in Figure 6D-H?
- As only 50% of SORE6+ cells express MenaINV and not all SORE6+ cells intravasate one wonders whether MenaINV expression is necessary for SORE6+ cells to intravasate. Could authors quantify MenaINV expression in SORE6+ vs SORE6- cells in the primary tumor, CTCs and distant lungs?

Minor comments

- In general, it is more appropriate to plot individual values instead of bars representing the mean + SEM. Alternatively, bars representing mean + SD give more information about the variability of the data than mean + SEM.
- Figure 4G: It seems that the error bar of the % of S6+ cells among cells cultured with macrophages is missing.
- Line 524: There is a reference to a non-existing Figure 6L.
- In Figure S5B "GFP-SORE6+" could be replaced by "SORE6+" to avoid confusion.
- Material and methods: Could authors precise the concentration/amount of clodronate liposomes given for in vivo treatment?
- Figure S6F: The y-axis should be modified so that the maximum is around the highest SORE6+ proliferation events observed. In this graph it does look like there is decrease proliferation among cells in contact with macrophages. Although they report non-statistically significant differences, in this graph, it is particularly important to show the actual individual values for an accurate evaluation of the data. The lack of significance could be driven by few "outlayers".

Reviewer comments are reproduced below in the regular font. Our responses to each of the reviewer comments are in **bold**.

The response to independent comments from Reviewer #4 are now addressed and are shown in cyan highlight here in the rebuttal as well as in the manuscript text.

Previous changes in the manuscript text are still shown in yellow highlight.

Reviewer #1 (Remarks to the Author):

The revised manuscript does address my most important concerns and been strengthened significantly. The revised manuscript would be of considerable interest for the readers of Nature Communications.

We are glad that we addressed all the concerns of this reviewer. We thank him/her for his comments which helped us significantly strengthen the manuscript and made it of considerable interest for the Nature Communications readership.

Reviewer #2 (Remarks to the Author):

In general, the authors have done a very good job addressing the concerns raised in the previous review. Their point re-survival data is well taken, and the metastasis data they provide add a lot to the manuscript.

However, the imaging data supporting the induction of the stem cell phenotype after macrophage contact in co-culture are still not clear.

The new movie (Movie 12) appears to show a consistently low level of GFP-expression in the reporter tumour cells prior to interaction with the macrophage, with what seems a very modest change in fluorescence intensity after interaction. How does this compare with the MinCMV-GFP ctrl used for thresholding, could these be shown?

The stills used in the figure (S6B) are slightly clearer than the movie, so some of this may be due movie compression and could benefit from the use of a 'glow' type look up table? However, as the graph in Fig. 4F depicts that nearly 1 in 4 macrophage:tumour cell interactions lead to the conversion to GFP+ it is surprising that a better example in terms of the clarity of signal change could not be found. There are two other, extremely bright, GFP+ tumour cells in this video alone that unfortunately we do not see the interaction history of.

We thank the reviewer for the suggestion about using lookup table (LUT). We redid Movie 12 and the corresponding image panels in Fig S6B by adding GFP channel shown in the "fire" LUT colors. The panels in Fig S6B and movie 12 now clearly show an increase in GFP signal (stemness induction) in a tumor cell after macrophage contact. As requested, we are also providing the mCMV-GFP control in Fig S6B, which was used to set the GFP threshold.

Reviewer #4 (Remarks to the Author):

Employing a previously described Sox2/Oct4 reporter (SORE6) and intravital imaging authors track breast cancer cells with cancer stem cell (CSC) properties. They show that SORE6+ cells are enriched in

stemness markers, have higher tumor-initiation activity, slow migratory phenotype and form more invadopodia compared to SORE6⁻ cells. Moreover, SORE6⁺ cells are predominant in tumor microenvironment of metastasis (TMEM) doorways, among circulating tumor cells and at distant lung metastatic sites at the initiation of the metastatic growth. Importantly, induction of SORE6 reporter expression in breast cancer cells is, at least in part, attributed to Notch1 signaling upon interaction with Jag1/Jag2 ligands on macrophages. Finally, they describe that cancer cells expressing markers of stemness are also enriched in TMEMs of breast cancer patients where macrophages, endothelial cells and cancer cells co-localize. Therefore, suggesting that TMEMs are not only metastasis doorways where cancer cells intravasate but also where stemness properties are acquired by cancer cells by interacting with macrophages.

The manuscript is well written and generally convincing. While intravital imaging only provides descriptive observations *ex vivo* readouts and *in vitro* experiments support their nice observations. Nonetheless, there are a couple of statements that are not robustly proven by the data and some technical/data representation aspects that may need to be addressed:

(1) While authors observe increase stemness markers (CD133 and ALDH1) expression in SORE6⁺ vs SORE6⁻ cells (as requested by Reviewer 3) there is still a big proportion of SORE6⁺ cells that do not express CD133 (Figure S1G) and it is not clear whether all SORE6⁺ cells have the capacity to form tumorspheres (Figure S1D). Moreover, in Figure S1B only 1 in 2974 SORE6⁺ cells have the capacity to initiate tumors in immunocompromised SCID mice and there is still a proportion of cells among SORE6⁻ cells with CSC properties. Therefore, it would be more appropriate to say that SORE6⁺ cells are enriched in CSCs rather than stating that SORE6⁺ cells are bona-fide CSCs.

This is related to Reviewer #3's comments on Figure 1, Figure S1 and Figure 7.

We have modified the text to emphasize that SORE6⁺ cells are “enriched for CSC properties” (page 6, lines 172-173). No stemness marker definitively identifies CSCs as opposed to enriching for the phenotype. The enrichment of SORE6⁺ cells within the CD133⁺ and AldeRed⁺ populations and vice versa is highly significant (Fig. S1G, H), strongly supporting the contention that SORE6 positivity identifies a cell population that is enriched for the stem phenotype as identified by multiple orthogonal approaches (cell surface marker, enzyme activity). The efficiency of the SORE6⁺ cells in tumorsphere formation and tumor initiating activity is fully in line with that observed by others for breast cancer models (e.g. Ginestier et al. PMID: 18371393, Bieri et al., PMID 28270621).

(2) Figure S1C: As SORE6⁺ and SORE6⁻ cells were mixed from the initiation of the treatment it is not possible to distinguish whether SORE6⁺ cells are indeed resistant to paclitaxel or whether SORE6 reporter expression is enhanced overtime due to other factors (cell-cell contacts etc). In fact, SORE6⁺ cell numbers decrease or barely grow in the first 20h whereas SORE6⁻ cells grow in the same time frame. Cell growth +/- paclitaxel of sorted SORE6⁺ vs SORE6⁻ cells should be assessed to confirm increased chemoresistance of SORE6⁺ cells.

Since chemotherapy acts in the real world on a mixture of stem and non-stem tumor cells, we felt it was more meaningful to assess the paclitaxel effect in this context. Fig S1C (right panel) shows that the expansion of the CSC compartment occurs with essentially similar kinetics in the presence or

absence of paclitaxel, despite the very different numbers of non-stem cells, and hence total cells, at the later times (Fig S1C, left panel). The selective loss of SORE6- non-stem cells from the mixed tumor cell population in response to paclitaxel is exactly as predicted by the CSC hypothesis – cancer stem cells will not be affected by chemotherapy while cancer non-stem cells will. We note that in our original paper validating the SORE6 reporter system, we used flow cytometry to show resistance of the SORE6+ cells to chemotherapy in an independent breast cancer model both in vitro and in vivo, (Tang et al., PMID: 25497455). Thus by multiple approaches, in multiple models, and in vitro and in vivo, we have demonstrated enhanced resistance of SORE6+ cells to chemotherapy, as expected if the reporter is marking cancer stem cells.

(3) Figure 1. Authors provide new evidence to exclude that that stress related TFs (CEBPA, XBP-1 and c-Jun) are responsible for the SORE6 reporter expression as requested by Reviewer 3. Moreover, they provide new data indicating that Oct4 is the main SORE6 reporter expression inducer (Figure S1F). Thus, it is not clear why authors use Sox9 instead of Oct4 to assess cancer cells stemness as an alternative to SORE6 reporter in the PyMT mammary primary tumor model (Figure 3H, 5H). Are Oct4+ cancer cells also decreased upon macrophage depletion and DAPT treatment? Moreover, authors define Sox9Hi cells as CSCs without an explanation of the criteria used to define high Sox9 expression, which is essential in immunofluorescence staining where technical factors can contribute to changes in staining intensity.

This comment is related to Reviewer #3's comment on Figure 1.

There are no rigorously validated antibodies for detection of Oct4 in tumor tissues. Thus we had to go with the Sox rather than the Oct arm in analyzing the PyMT model. Our data (Fig. S5E) identified Sox9 as the Sox paralog that was enriched in SORE6+ CSCs in the PyMT model.

To improve the technical quality of the data in the manuscript, we have addressed the Sox9Hi comment from reviewer #4. The criteria for defining Sox9Hi cells has already been outlined in the materials and methods section on page 33. In the newly revised version we have expanded this part further as follows: "For Sox9Hi quantification, FFPE tumor sections were stained with Sox9 antibody (EMD Millipore, cat# ab5535) and Sox9Hi tumor cell population was defined by choosing ~ top 5% of the tumor cells with the highest Sox9 fluorescence signal in the control tissues, which corresponded to a fluorescence intensity threshold range of 80-255. Same intensity threshold was applied to the clodronate tissue. Both control and clodronate tissues were imaged at the same excitation intensity and exposure times on 3D Histech Panoramic P250 digital whole slide scanner." This expansion can be found on page 33, lines 1037-1043 of the revised manuscript.

On page 10, line 295, in the Results section where we first mention Sox9Hi cells, we have added a sentence - "see materials and methods for Sox9Hi definition", to let the reader know where this information could be found.

(4) As Reviewer 3 pointed out it is important to state that the frequency of SORE6+ cells in vitro is much higher than in vivo to properly interpret Figure 4 co-culture experiments. Thus, the initial frequencies of SORE6+ cells in vitro should be described in the results part in addition to material and methods section. Moreover, while normalizing SORE6+ cell frequencies to control samples makes the comparison with the experimental condition easier in Figures 4D and E, it does not show whether SORE6+ cell numbers change over time in culture, which is important to validate that macrophages are strictly necessary for

SORE6 reporter induction. Could authors represent the frequencies of SORE6+ cells at the initiation of the co-culture and at the end-timepoint?

This comment is related to Reviewer #3's comment on Figure 4.

As requested, in addition to the materials and methods section, we have now added the initial frequencies of SORE6+ cells for the three tumor cell lines (MDA-MB-231, MDA-MB-231-LM2 and 4T1) in the results section on page 11, lines 328-329 and 336-337. The data in 4D, E represent relative SORE6+ frequencies at the end of the experiments (with all conditions having the same initial SORE6+ frequencies), so the increase in frequency seen at endpoint in the co-culture does support a role for macrophages in increasing CSC number. However, the most convincing demonstration of the effect of macrophage contact on tumor cell acquisition of the stem phenotype comes from the fate and origin mapping data in Fig 4F, G. In these experiments the induction events were observed and quantitated directly by monitoring the behavior of individual cells over time, rather than relying on endpoint analyses of the bulk culture. As seen in this analysis, there is a low level of direct (macrophage-independent) conversion of SORE6- to SORE6+ cells in the cultures, but the conversion rates are greatly increased by macrophage contact. This mapping approach represents an extremely rigorous and quantitative validation of the role of the macrophage in this process.

(5) To confirm that macrophages are inducing cancer cell stemness, the tumor initiation activity of cancer cells previously in contact with macrophages compared to those cultured alone could be assessed.

We agree that it is important to demonstrate the functionality of the cancer stem cells that are induced by macrophage contact. To do this, we showed that tumor cells that had previously been in contact with macrophages had an increased efficiency of tumorsphere formation (Fig 4C), which is a widely used in vitro surrogate of tumor initiating/cancer stem cell activity.

(6) The in vitro data clearly points out to the role of Notch1 signaling on cancer cells in the induction of SORE6 reporter expression by interaction with Jag1/Jag2 ligands on macrophages. However, the tumor microenvironment greatly differs from in vitro conditions and additional experiments would be needed to confirm that the same Notch1/Jagged signaling predominates the stemness acquisition of breast cancer cells in TMEMs in vivo. The current macrophage depletion with clodronate liposomes and Notch1 inhibition does not completely proof the cross-talk:

Clearly the mechanistic pathway underpinning the induction of the stem phenotype by macrophages could be fleshed out in more detail in future studies. For the current manuscript, we feel that we have convincingly demonstrated the involvement of the Notch pathway in this interaction both in vitro and in vivo, and that this is the broad concept that we want to put across. Our in vitro work nominates Jag1,2 as plausible candidate ligands, but extensive (and expensive) additional work will be required to rigorously identify the specific ligand in the in vivo setting. We feel this undertaking is beyond the scope of the current manuscript. As per reviewer's request, we have clarified this on lines 410-411 and 654-657

(6A) Figure 3D-E. Even if in co-culture experiments (Fig 4F) SORE6+ cells do not die 30h following loss of macrophage contact, it is still not clear whether SORE6+ cells die, SORE6- cells cannot acquire SORE6+ phenotype or SORE6+ cells convert into SORE6- upon macrophage depletion in vivo. In vitro and in vivo conditions are too different to be able to extrapolate what happens to SORE6+ cells in vivo from in vitro observations. Could authors provide the non-normalized cell counts/frequencies of SORE6- cells upon clodronate treatment (as for SORE6+ cells in Figure S5B) to assess whether a SORE6- phenotype enrichment occurs or only SORE6+ cells die upon macrophage depletion in vivo?

It is true that in vitro and in vivo conditions are different. However, having observed a macrophage contact-dependent induction of stemness in non-stem tumor cells in vivo, we were then able to reproduce this phenomenon in vitro, suggesting that the key features of this interaction are independent of any major modulatory effect of the tumor microenvironment. The reviewer will appreciate that cancer stem cells are a minor subpopulation of tumor cells in vivo, so it is difficult to capture a sufficient number of conversion events and post-conversion fates by the intravital imaging approach to draw statistically meaningful conclusions. This does not undermine the biologically meaningful and statistically solid observation that macrophage depletion for two days in vivo results in a reduction in the cancer stem cell population in the tumor. The in vitro data suggest that this phenomenon results primarily from blockade of de novo CSC generation by macrophages, but it is conceivable that additional mechanisms (such as death of pre-existing CSCs) may also contribute in vivo. As requested, we have added these caveats in the discussion section on lines 657-661.

(6B) While authors attempt to address Reviewer 3's concern about the possible direct impact of clodronate on SORE6+ cancer cells, the provided data in Figure S5C is limited. Whether clodronate impacts the general cancer cell viability and growth (in SORE6+ and SORE6- cells) is missing. Would the same in vitro results be obtained when liposome clodronate instead of clodronate was administered?

This comment is related to Reviewer #3's comment on Figure 3.

New data showing no direct effect of clodronate on total tumor cells or on the SORE6+ subpopulation in vitro have been added to replace the original Fig. S5C. In vivo, the purpose of the liposomal encapsulation is to make uptake of the clodronate specific to phagocytic cells such as macrophages. In the in vitro culture, since the tumor cells are not phagocytic, liposomal clodronate will have less of an effect on these cells than free clodronate.

(6C) Figure 5H shows that Notch signaling is important for Sox9 expression on cancer cells but it does not confirm whether Notch interaction with Jag1 or Jag2 on macrophages is required. Could authors check whether in vivo blocking of Jag1 and Jag2 decreases the Sox9 expressing cancer cell frequencies? If authors find that the in vivo validation of the Notch1/Jagged signaling mediated macrophage-cancer cell crosstalk is out of the scope of the manuscript they may state it in the discussion.

The question of the in vivo Notch1/Jagged macrophage-cancer cell crosstalk is an interesting one but beyond the scope of the current manuscript. As mentioned above in point 6, we have clarified this on lines 410-411 and 654-657 as per reviewer's request.

(7) Figure 6I, while the images and video nicely show a SORE6+ cell intravasating in a TMEM, how do authors know that intravasation does not occur in other non TMEM vessels and by SORE6- cells? Intravasation events in non-TMEM vessels and by SORE6- cells would need to be monitored and quantified.

The assessment of exclusive SORE6+ cell intravasation in TMEMs is beyond of the scope of current manuscript, therefore we have removed the word “preferentially” on line 631 in the Discussion.

Using intravital microscopy our group has established TMEM as the only site for cancer cell intravasation (Harney et al. Cancer Disc. 2015, PMID: 26269515; Harney et al. Mol Cancer Ther. 2017, PMID: 28838996), therefore the capacity of cancer cells with stem cell properties to intravasate in other areas outside TMEMs were not evaluated. We have added this information on lines 458-460.

(8) Figure 7D-E: To confirm that MenaINV expression is found exclusively or predominantly in SORE6+ cells, quantification of MenaINV+ cell frequencies among SORE6- cells would be necessary.

Evaluation of MenaINV expression in SORE6- cells is beyond of the scope of current manuscript. We have added a sentence in the manuscript on line 489 to clarify this.

(9) Figure 7F-G. To confirm that SORE6+ cells have increased capacity to metastasize into distant organs and exclude that the increased SORE6+ cell frequencies in the lung was not due to in situ acquisition of SORE6 expression by SORE6- CTCs after extravasation, i.v injection of SORE6+ vs SORE6- cells and analysis of SORE6 expression in the lungs could be performed (as in Figure S7B).

We have previously published that the MDA-MB-231 SORE6+ cells have an increased metastatic efficiency compared with SORE6- cells (Tang et al; PMID: 25497455, Fig 3F, reproduced below).

Lung metastases formed following tail-vein injection of sorted SORE6+ and SORE6- and sham-sorted cells from MDA-MB-231 cultures. Results are shown as median \pm interquartile range for n = 5 mice/group. *p < 0.05, two-way ANOVA.

(10) Figure 8G: Which non-biased parameter was used to define a TMEM rich areas? Could authors exclude that the Sox9hi enrichment did not occur in non-TMEM vessel rich areas? Could be possible to assess the distance of Sox9hi cells from TMEMs vs non-TMEM vessels in patients tissues as performed in Figure 6D-H?

To improve the technical quality of the manuscript, we have now added the definition of TMEM-rich area as follows. “TMEM-rich areas are defined as at least 3 TMEM doorways per 40X magnification field of view, or 30 TMEM doorways per the ten 40X fields (TMEM score). This cutoff point was chosen

because TMEM score of 23 is associated with increased risk of developing metastatic disease (Rohan et al JNCI 2014)". We have added this information in a new subsection "Sox9 and triple-IHC TMEM staining in human samples" under materials and methods on page 43, lines 1369-1382, in the revised manuscript.

The additional analyses requests are beyond the scope of current manuscript.

(11) As only 50% of SORE6+ cells express MenaINV and not all SORE6+ cells intravasate one wonders whether MenaINV expression is necessary for SORE6+ cells to intravasate. Could authors quantify MenaINV expression in SORE6+ vs SORE6- cells in the primary tumor, CTCs and distant lungs?

This is an interesting question but goes beyond the scope of the current manuscript. We have added the points raised here by the reviewer as a possible future study in the discussion section on lines 635-639

Minor comments

(12) In general, it is more appropriate to plot individual values instead of bars representing the mean + SEM. Alternatively, bars representing mean + SD give more information about the variability of the data than mean + SEM.

To improve the technical quality of the data in the manuscript, we have replotted all our graphs as mean +/- SD, as suggested by reviewer #4.

(13) Figure 4G: It seems that the error bar of the % of S6+ cells among cells cultured with macrophages is missing.

There is no error bar because all four independent wells analyzed gave the same result (see raw data below). We have added this information in the legend to figure 4G.

Tumor cells alone		S6- to S6+ conversion events/total S+ cells traced			
		Well #1	Well #2	Well #3	Well #4
	Field #1	0/8	0/8	1/8	0/8
	Field #2	0/8	1/8	0/8	1/8
	Field #3	0/8	1/8	1/8	2/8
	Field #4	0/8	1/8	0/8	0/8
Total # S6- to S6+ conversion events		0	3	2	3
Total # S6+ cells origin-traced		32	32	32	32
% S6+ cells generated by direct conversion from S6-		0	9.37	6.25	9.37

Tumor cells + BACs		S6- to S6+ conversion events/total S+ cells traced			
		Well #1	Well #2	Well #3	Well #4
	Field #1	3/8	2/8	2/8	2/8
	Field #2	2/8	2/8	4/8	2/8
	Field #3	2/8	2/8	1/8	2/8
	Field #4	2/8	3/8	2/8	3/8

Total # S6- to S6+ conversion events		9	9	9	9
Total # S6+ cells origin-traced		32	32	32	32
% S6+ cells generated by direct conversion from S6-		28	28	28	28

(14) Line 524: There is a reference to a non-existing Figure 6L.

Thank you for pointing out this typo. It has been corrected to Figure 7L (page 17, line 532)

(15) In Figure S5B “GFP-SORE6+” could be replaced by “SORE6+” to avoid confusion.

We have made this change in Fig S5B.

(16) Material and methods: Could authors precise the concentration/amount of clodronate liposomes given for in vivo treatment?

Clodronate liposome concentration given for the in vivo treatment was 18.4 mM. We have now added this information in the Materials and Methods section on page 33, line 1033.

(17) Figure S6F: The y-axis should be modified so that the maximum is around the highest SORE6+ proliferation events observed. In this graph it does look like there is decrease proliferation among cells in contact with macrophages. Although they report non-statistically significant differences, in this graph, it is particularly important to show the actual individual values for an accurate evaluation of the data. The lack of significance could be driven by few “outliers”.

For tumor cells alone, summing across 4 wells analyzed, a total of 3/128 (2.3%) SORE6+ cells were derived from proliferation of pre-existing SORE6+ cells, while for tumor cells + macrophages, a total of 1/128 (0.8%) SORE6+ cells came from proliferation (see raw data below). Treating each well independently (32 cells mapped/well), and applying the t-test, the difference between the two groups is not statistically significant ($p=0.207$). The plot in Fig S6F and its figure legend have been updated as requested to make this clearer.

Tumor cells alone	Well #1	Well #2	Well #3	Well #4
---------	---------	---------	---------

# S6- to S6+ conversion events	0	3	2	3
# S6+ to S6+ proliferation events	1	1	1	0
Total # S6+ cells origin-traced	32	32	32	32
% S6+ cells generated by direct conversion from S6-	0	9.37	6.25	9.37
% S6+ cells generated by proliferation of pre-existing S6+ cells	3.1	3.1	3.1	0
Tumor cells + BACs	Well #1	Well #2	Well #3	Well #4
# S6- to S6+ conversion events	9	9	9	9
# S6+ to S6+ proliferation events	0	1	0	0
Total # S6+ cells origin-traced	32	32	32	32
% S6+ cells generated by direct conversion from S6-	28	28	28	28
% S6+ cells generated by proliferation of pre-existing S6+ cells	0	3.1	0	0

Reviewers' Comments:

Reviewer #2:

Remarks to the Author:

Thanks for making those changes to the movie and figure, this is now much easier to interpret.

Reviewer #4:

Remarks to the Author:

While authors address all reviewer 3's comments. Many of my independent comments have not been addressed.

Some of my concerns may be minor on the overall context: my concern on the validity of the experiment of Figure S1C to proof the chemoresistance of SORE6+ vs SORE6- cells (point 2) may be minor due to the many parameters sustaining the enrichment of CSC among SORE6+ cells. The assessment of exclusive SORE6+ cell intravasation in TMEMs (point 7) and co-occurrence of MenaINV expression and acquisition of stemness properties (points 8 and 11) at TMEMs may be out of the scope of the manuscript. However, it should be mentioned in the discussion, as limitation of the study, that the capacity of cancer cells with stem cell properties to intravasate in other areas outside TMEMs were not evaluated (relative to points 7,9 and 10).

Moreover, they don't provide sufficient evidence to proof that contact dependent -Notch-Jagged 1/2 signaling between cancer cells and macrophages are responsible in vivo for the acquisition of stemness properties by cancer cells. Macrophage specific Jagged 1/2 blockade in vivo would be needed to confirm their hypothesis (relative to point 6). Thus, lines 408-410 and 645-646 should be rephrased to note that further experiments are needed for its confirmation in vivo. Indirect effects of macrophage depletion may have also driven loss of SORE6+ cells in vivo.